# Decoding Generalization from Memorization in Deep Neural Networks

**Simran Ketha**                                                             *p20200021@hyderabad.bits-pilani.ac.in*
*Anuradha and Prashanth Palakurthi Centre for Artificial Intelligence Research,*
*Department of Computer Science & Information Systems,*
*Birla Institute of Technology & Science Pilani, Hyderabad 500078, India.*

**Venkatakrishnan Ramaswamy**                                                *venkat@hyderabad.bits-pilani.ac.in*
*Anuradha and Prashanth Palakurthi Centre for Artificial Intelligence Research,*
*Department of Computer Science & Information Systems,*
*Birla Institute of Technology & Science Pilani, Hyderabad 500078, India.*

**Reviewed on OpenReview:** *https://openreview.net/forum?id=BeT6jaD6ao*

## Abstract

Overparameterized deep networks that generalize well have been key to the dramatic success of deep learning in recent years. The reasons for their remarkable ability to generalize are not well understood yet. When class labels in the training set are shuffled to varying degrees, it is known that deep networks can still reach perfect training accuracy at the detriment of generalization to true labels – a phenomenon that has been called *memorization*. It has, however, been unclear why the poor generalization to true labels that accompanies such memorization, comes about. One possibility is that during training, all layers of the network irretrievably re-organize their representations in a manner that makes generalization to true labels difficult. The other possibility is that one or more layers of the trained network retain significantly more latent ability to generalize to true labels, but the network somehow "chooses" to readout in a manner that is detrimental to generalization to true labels. Here, we provide evidence for the latter possibility by demonstrating, empirically, that such models possess information in their representations for substantially-improved generalization to true labels. Furthermore, such abilities can be easily decoded from the internals of the trained model, and we build a technique to do so. We demonstrate results on multiple models trained with standard datasets. Our code is available at: `https://github.com/simranketha/MASC_DNN`.

## 1 Introduction

Prior to the advent of deep learning, the conventional wisdom for long[1], was that in building a predictive model, the model should have as few parameters as possible and this number should certainly be less than the number of training samples that one was fitting. The dogma was that, otherwise, the model would exactly fit the training points, but invariably generalize poorly to unseen data, i.e. overfit. This intuition was also largely borne out by the models of the day. Modern deep learning, however, has gone on to show the opposite, namely that overparameterized models not only don't necessarily overfit, but that they can generalize remarkably well to unseen data. However, over a decade later, we still do not satisfactorily understand why this is so. Interestingly, it has been shown (Zhang et al., 2017; 2021) that when one randomly shuffles class labels of data points from standard training datasets to varying degrees, deep networks can still have high/perfect training accuracy when trained on such corrupted training data; however, this appears to typically be accompanied by poor performance on unseen test data (that have true labels). This phenomenon

---

[1]von Neumann famously said, "With four parameters I can fit an elephant, and with five I can make him wiggle his trunk." (Dyson et al., 2004)

has been called *memorization*[2], since it is thought that the model rote-learned the training data without acquiring the ability to generalize to true labels. It has been suggested that progress on understanding memorization could enable a better understanding of generalization to true labels in deep networks trained on real-world data (Zhang et al., 2017; 2021) and indeed that a detailed understanding of mechanisms of generalization to true labels should also be able to explain the phenomenon of memorization.

An open question arising in this context is about the detailed mechanisms that lead to poor generalization to true labels in models trained with shuffled labels, i.e. models that memorize. A natural hypothesis governing such mechanisms, stated informally, is that, during training, the network organizes its internal representations in all layers, in a manner suited to doing well on the (corrupted) training data. Since this data is significantly noisy, on being given unseen data with true labels, it fundamentally lacks the ability to have good prediction performance, leading to poor generalization to true labels. An alternative hypothesis is that layerwise representations on a subset[3] of the layers in the network retain significantly more ability to generalize to true labels, than the model, but that the network somehow chooses to readout in favor of high training accuracy in a manner that incidentally causes poor generalization to true labels. A consequence of this alternative hypothesis is that one ought to be able to construct a decoder (i.e. a probe) for the outputs of such layers that has better generalization performance on true labels.

Here, surprisingly, we show evidence for this alternative hypothesis. In particular, we study the organization of subspaces of class-conditioned training data on layerwise outputs, in deep networks. We estimate these subspaces using Principal Components Analysis (PCA). In order to remain oblivious to the information decoded by subsequent layers, we build a simple probe that leverages the geometry of the present layer's output of an incoming datapoint, relative to these class-conditioned subspaces. Specifically, we measure the angle between this output vector and its projection on each of these class-conditioned subspaces and the probe predicts the datapoint's class to be the class whose subspace has the minimum such angle. We call this probe the Minimum Angle Subspace Classifier (MASC). Notably, unlike probes used conventionally (e.g. in (Alain & Bengio, 2018)) whose parameters are determined by iteratively minimizing a crossentropy loss, the parameters of MASC are directly determined from the subspace geometry of the training data. A schematic illustrating the geometry of MASC is presented in Figure 2.

We train a number of deep networks with standard datasets in the memorization setting. Here, a randomly-chosen fraction of training data points have their labels changed to a randomly-chosen label from the available labels in the dataset. We do so for differing fractions of the training dataset and – consistent with previous work (Zhang et al., 2017; 2021; Arpit et al., 2017) – see that training with such corrupted training datasets causes correspondingly poor test accuracies in the model. However, MASC – which uses the internals of the network to predict the class label – tends to do significantly better[4] than the model on the test set. A schematic illustration of the memorization setting with MASC is shown in Figure 1.

We outline a more detailed summary of our main contributions below.

1. For models trained with standard methods & datasets with training data corrupted by label noise to varying degrees, we demonstrate (with one exception) that MASC applied on at least one layer, when using subspaces corresponding to such corrupted training data, has significantly better test accuracy than the model. For example, MASC outperforms the model test accuracy by upto 159.93%, 189.00%, 64.86% and 119.16% on MLP-MNIST, CNN-Fashion-MNIST, AlexNet-CIFAR-100 and ResNet-18-CIFAR-10 respectively. A more detailed account of these numbers is in Table 1.

2. For the models discussed above, we perform a comparison study evaluating five probes namely Logistic regression , K-Nearest Neighbor (KNN), Linear Discriminant Analysis(LDA), Quadratic Discriminant Analysis (QDA), and Nearest Class Mean (NCM) over the layers of multiple deep networks under varying corruption degrees. While many of these probes exhibit similar overall accuracies and show no consistent trends across corruption degrees, their computational costs differ considerably, which we have empirically analyzed.

---

[2]We direct the reader to Section 3.1 for a formal description of our setting.

[3]It is indeed possible that certain layers of the network have representations that are significantly more generalizable to true labels than others and these layers may be early or later layers.

[4]We find that a few other probes do comparably well.

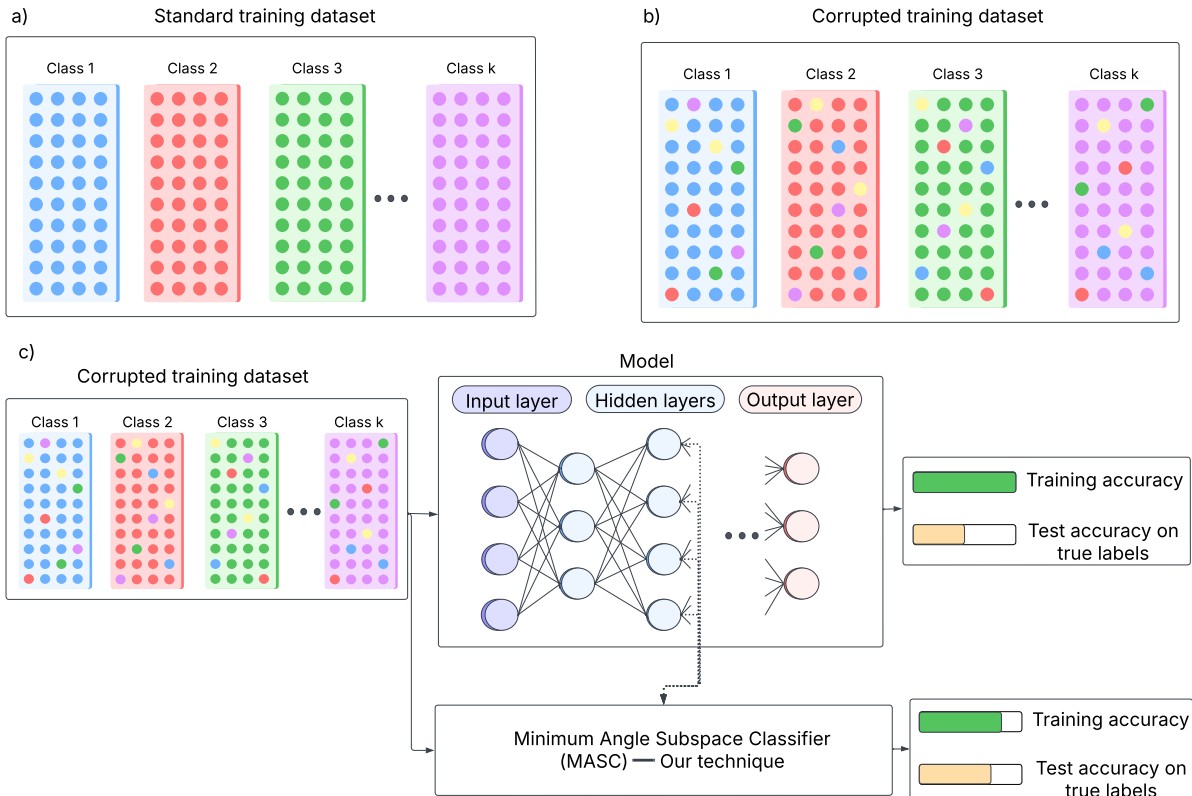

Figure 1: A schematic of the memorization setting used in our work and the application of the MASC classifier in it. a) Illustration of a standard training dataset. b) A corrupted training dataset is created by changing the labels of the standard training dataset with a specific probability (due to which a few of the colors are changed, representing the changed labels). Changing the labels happens uniformly at random for the whole dataset. c) A deep network is trained with this corrupted training dataset to achieve ~100% training accuracy, which is usually accompanied by poor test accuracy (as measured on true labels from the test set). We have shown that the Minimum Angle Subspace Classifier (MASC) – our technique – which uses the internals of the deep network, tends to have significantly better generalization to true labels (test accuracy) than the deep network itself.

3. For the aforementioned models, if the true training class labels are known post hoc, i.e. after the model is trained, we can build MASC using subspaces corresponding to true class labels. These MASC classifiers usually have better generalization to true labels than in (1). For example, MASC using true labels outperforms the model by upto 198.43%, 212.42%, 337.51% and 228.64% on MLP-MNIST, CNN-Fashion-MNIST , AlexNet-CIFAR-100 and ResNet-18-CIFAR-10 respectively. A more detailed account of these numbers is in Table 7. This demonstrates that the layers of the memorized network maintain representations in a manner that is amenable to straightforward generalization to true labels to a degree not previously recognized.

4. Conversely, we asked if a model trained on true training labels similarly retained internal representations that have the capability to memorize easily, as manifested by MASC. Adapting our technique to this setting, we create corrupted training sets which we use to build MASC. In this setting, we find that we can extract a high degree of memorization, in some cases. The results are presented in Section H.

5. Finally, leveraging the MASC classifiers built in (1) and (3), we ask, if we can retrain the memorized model for a few epochs to achieve better model generalization to true labels. We find that indeed, in many cases, there is an improvement in the generalization to true labels of the model.

## 2 Related work

The idea of probing intermediate layers of deep networks isn't new. For example, kernel-PCA (Montavon et al., 2011) with RBF kernels has been used to analyze layerwise evolution of representations of deep networks. In that work, they quantify the quality of layerwise representations and find that the last layers of the network tend to have representations that are more simple and accurate than previous layers. Likewise, linear classifier probes (Alain & Bengio, 2018) have been used to study the roles and dynamics of intermediate layers in deep networks. There, they show that the degree of linear separability increases over the layers of the network. However, they explicitly avoid examining memorized networks (Zhang et al., 2017) because they thought such probes would inevitably overfit. Our results are therefore especially surprising in this context, because we demonstrate, on the contrary, that intermediate representations, in fact, tend to resist overfitting, to a degree not previously recognized.

The authors in (Zhang et al., 2017) investigate the question of whether deep networks can perfectly learn noise by training on randomly relabeled data, showing that although such models achieve near-perfect training accuracy, they exhibit poor generalization to the true labels. Subsequent work by (Arpit et al., 2017) shows that early on in training, memorized networks (Zhang et al., 2017) start off by having better generalization to true labels; however generalization to true labels worsens as training accuracy increases across epochs of training. Despite these observations, there have been limited efforts to analyze the internal representations of models in the memorization regime (i.e. one where the labels of a subset of training data points are shuffled randomly) using probing methods. It was assumed that the cause of poor generalization to true labels in memorization settings directly manifests from poor representations. For example, in a probe-based analysis (Alain & Bengio, 2018) argue that probe measurements would be "entirely meaningless" in memorization settings. They say

> "It was recently demonstrated that very large models can fit random labels on ImageNet (Zhang et al., 2016). This is a situation that we want to avoid because the probe measurements would be entirely meaningless in that situation."

This view aligns with our null hypothesis that, during training, all layers of the network irretrievably reorganize their representations in a manner that makes generalization to true labels difficult. However, the alternative hypothesis that we articulate here has not been explicitly examined in past work, presumably because it was not considered plausible: that one or more layers of the trained network retain significantly more latent ability to generalize to true labels, but the network somehow "chooses" to readout in a manner that is detrimental to generalization to true labels.

There have been efforts to build training algorithms that are designed to extract better generalization to true labels in the case where the data is known to be noisy (Jiang et al., 2018; Han et al., 2018; Liu et al., 2020). Stephenson et al (Stephenson et al., 2021) investigate memorized models, suggesting that memorization predominantly occurs in the later layers. This is based, in part, on the observation that rewinding early-layer weights to their early-stopping values can recover generalization to true labels, whereas rewinding later-layer weights does not yield the same effect. In general, the thinking in the field has been that while there is an initial peak in generalization to true labels, it is lost during further training, although one can mitigate some of this loss by modifying training (Jiang et al., 2018) or by rewinding a subset of weights to their early values. On the contrary, our results suggest that layerwise outputs of deep networks retain significant ability to generalize after training and we demonstrate that this generalization to true labels can be extracted without modifying the weights of the trained network that are obtained via standard training methods.

An important line of theoretical research on deep linear models has explored the question of generalization to true labels (Saxe et al., 2013). Here, a theoretical explanation for the phenomenon of memorization in networks trained with noisy labels has been proposed (Lampinen & Ganguli, 2018).

Studies have investigated training dynamics across layers using various forms of Canonical Correlation Analysis (Raghu et al., 2017), including analyses in both generalized and memorized networks (Morcos et al., 2018). Centered Kernel Alignment has been employed to examine the effects of different random initializations (Kornblith et al., 2019), as well as to study network similarity between models trained on the same data with different initializations (Kornblith et al., 2019). Additionally, experiments have explored the use of representational geometry measures to understand the dynamics of layerwise outputs (Chung et al., 2016; Cohen et al., 2020), along with other structural measures such as curvature dimensionality (Hénaff et al., 2019), which aim to capture underlying properties of learned representations (Sussillo & Abbott, 2009; Farrell et al., 2019; Gao & Ganguli, 2015; Litwin-Kumar et al., 2017; Bakry et al., 2015; Cayco-Gajic & Silver, 2019; Yosinski et al., 2014; Stringer et al., 2019).

To address label noise, various heuristic approaches have been proposed (Khetan et al., 2017; Scott et al., 2013; Reed et al., 2014; Zhang & Sabuncu, 2018; Malach & Shalev-Shwartz, 2017), particularly in the context of classification tasks (Frénay et al., 2014; Ren et al., 2018; Menon et al., 2018; Shen & Sanghavi, 2019). In the case of overparameterized models, Li et al (Li et al., 2020) demonstrate that memorization requires the network weights to deviate significantly from their initial random state in order to overfit noisy labels. Additionally, in a theoretical model of epochwise double descent (Stephenson & Lee, 2021), it has been suggested that for smaller models, moderate levels of label noise can lead to a reduction in generalization error at later stages of training.

## 3 Methods

### 3.1 Preliminaries

In this subsection, we state precisely the setting that is treated in this paper.

We study a classification task defined over a data distribution $D$, with an i.i.d. training dataset $T$ drawn from $D$. For a given corruption degree $p$[5], we generate a modified training set $\hat{T}_p$ by randomly relabeling a certain expected fraction of the training samples uniformly at random. The resulting label corrupted training data distribution is denoted by $\hat{D}_p$.

A series of prior studies (e.g., Zhang et al. (2017); Arpit et al. (2017)) have demonstrated that deep networks can successfully fit such corrupted datasets $\hat{T}_p$, even when a substantial fraction of the labels are randomized. These findings highlight the remarkable expressive power of the hypothesis class $H$ associated with contemporary deep models. In particular, they show that $H$ is sufficiently rich to learn training data arising from highly perturbed label distributions such as $D_p$, thereby enabling the model to achieve near perfect training accuracy despite the presence of significant label noise.

As the corruption degree $p$ increases, the distribution of the corrupted data $\hat{D}_p$ diverges progressively from the true distribution $D$. Consequently, one would expect that any network $h$ trained to fit samples $\hat{T}_p$ drawn from $\hat{D}_p$ becomes increasingly misaligned with the task defined by $D$, and therefore performs poorly on test dataset $T'$ drawn from distribution $D$. This phenomenon has been called[6] *memorization* in past work (e.g. (Zhang et al., 2017; 2021; Arpit et al., 2017)).

---

[5]When we say the training dataset has corruption degree $p$, we mean that with probability $p$, we attempt changing the label for each training datapoint. Changing the labels happens uniformly at random to any of the class labels. Note that this may result in the label remaining the same; therefore the expected fraction of datapoints whose labels changed are $p - p/K$, where $K$ is the number of class labels. So, e.g. for a dataset with 10 classes, this would mean that for corruption degrees of 20%, 40%, 60%, 80% and 100%, the expected percentage of training datapoints with changed labels is 18%, 36%, 54%, 72% and 90% respectively. We have run experiments for values of $p$ being 0% (generalized model), and memorized models with $p$ being 20%, 40%, 60%, 80% and 100% .

[6]This phenomenon could also be viewed as learning under label noise. However, given the usage of the term memorization to refer to this phenomenon in past work, we choose to continue to do so here.

For a network trained on $\hat{T}_p$ drawn from $\hat{D}_p$, generalization would, usually, refer to network's performance on $\hat{D}_p$ after learning from $\hat{T}_p$. In this work, however, we examine whether a network trained on $\hat{T}_p$ (using the network's internal representations) can perform well on test data drawn i.i.d. from the true distribution $D$. Accordingly, we define good generalization in terms of high performance on a test set $T'$ sampled from $D$. Throughout the rest of the paper, we refer to this notion simply as generalization to true labels.

### 3.2 Experimental setup

We have used multiple models and datasets, namely Multi-layer Perceptron (MLP) trained on MNIST (Deng, 2012) and CIFAR-10 (Krizhevsky, 2009) datasets, Convolutional Neural Networks (CNN) [7] trained on MNIST, Fashion-MNIST (Xiao et al., 2017), and CIFAR-10, AlexNet (Krizhevsky et al., 2012) trained on CIFAR-100 (Krizhevsky, 2009) and Tiny ImageNet (Moustafa, 2017) and ResNet-18(He et al., 2016) trained on CIFAR-10. We have trained these models with training data having true labels ("generalized models") as well as separately using training data with labels shuffled to varing degrees ("memorized models") (Zhang et al., 2017; 2021).

A summary of the models, datasets, training set sizes, and number of parameters is provided in Table 2. Tables 3 and 4 report the average training and test accuracies of all models over three runs. Additional details on the models, hyperparameters, and training procedures are also included in the Appendix. The general terminology used in this work is also explained in the Appendix.

Following standard practice in studying memorized models (e.g. Stephenson et al. (2021)), we do not use explicit regularizers such as dropout or batchnorm, or early stopping, unless otherwise mentioned, as a result of which our baseline test accuracy numbers are often much lower than what is usually found with standard training of these models. All the models are trained to either reach very high training accuracies (i.e. $99\% - 100\%$) or trained until 500 epochs. Some models did not reach such high accuracies, in which case, results have been shown on the model obtained at epoch 500. We trained 3 instances of each model and results displayed are averaged over these instances with the shaded region indicating the range of results also indicated in the plots.

### 3.3 Minimum Angle Subspace Classifier Algorithm (MASC)

For a given data point $\boldsymbol{x}$ from the training or test set, a layer output data point $\boldsymbol{x_l}$ from layer $l$ when input $\boldsymbol{x}$ is passed through the network and its corresponding training subspaces $\{S_k\}_{k=1}^K$ with $K$ classes, we use Minimum Angle Subspace Classifier (MASC) Algorithm 1 for predicting class labels $y(\boldsymbol{x_l})$.

---

**Algorithm 1 Minimum Angle Subspace Classifier (MASC)**

---

**Input:** Training subspaces $\{S_k\}_{k=1}^K$, layer output data point $\boldsymbol{x_l}$ from layer $l$ when input $\boldsymbol{x}$ is passed through the network and classes $\{C_k\}_{k=1}^K$.
**Output:** MASC prediction class label $y(\boldsymbol{x_l})$ according to layer $l$ .

1: **for** each class $C_k$ **do**
2:     $\boldsymbol{x_{lk}} \longleftarrow$ compute the projection of $\boldsymbol{x_l}$ onto subspace $S_k$.
3:     Compute the angle $\theta(\boldsymbol{x_l}, \boldsymbol{x_{lk}})$ between $\boldsymbol{x_l}$ and $\boldsymbol{x_{lk}}$
4: **end for**
5: Assign the label $y(\boldsymbol{x_l}) = C_k$ where $k = \arg\min_k \theta(\boldsymbol{x_l}, \boldsymbol{x_{lk}})$
6: **Return:** label $y(\boldsymbol{x_l})$

---

Given training dataset $\mathcal{D}\{(\boldsymbol{x_i}, y_i)\}_{i=1}^m$, where each $\boldsymbol{x_i} \in \mathbb{R}^d$ and $y_i \in \{C_k\}_{k=1}^K$ are input-label pairs, we estimate training subspaces $\{S_k\}_{k=1}^K$ for all classes $K$, for a given layer $l$ of the neural network using Algorithm 2 and 3. In practice, $S_k$ is represented via its principal components, which form a basis for the subspace.

We have used 99% as the percentage of variance explained by the principal components, unless otherwise mentioned. While the subspaces are estimated using the training data alone, accuracy of the MASC is

---

[7]The CNN models were built along the lines of (Tran et al., 2022).

---

**Algorithm 2 Subspaces Estimator for MASC**

---

**Input:** Training dataset $\mathcal{D}\{(\boldsymbol{x_i}, y_i)\}_{i=1}^m$, where each $\boldsymbol{x_i} \in \mathbb{R}^d$ and $y_i \in \{C_k\}_{k=1}^K$ are input-label pairs, neural network, and layer $l$.
**Output:** Subspaces $\{S_k\}_{k=1}^K$ for classes $K$, for given layer $l$.

  1: $\mathcal{D}_l = \phi$
  2: **for** each input pair $(\boldsymbol{x_i}, y_i)$ in $\mathcal{D}$ **do**
  3:     Pass $\boldsymbol{x_i}$ through the network layers to obtain the output of layer $l$, denoted as $\boldsymbol{x_l} \in \mathbb{R}^{ld}$.
  4:     $\mathcal{D}_l = \mathcal{D}_l \cup \{(\boldsymbol{x_l}, \boldsymbol{y_i})\}$
  5: **end for**
  6: Estimated subspaces $\{S_k\}_{k=1}^K \longleftarrow$ **PCA-Based Subspace Estimation**$(\mathcal{D}_l)$
  7: **Return:** Subspaces $\{S_k\}_{k=1}^K$

---

---

**Algorithm 3 PCA-Based Subspace Estimation**

---

**Input:** Layer output $\mathcal{D}_l = \{(\boldsymbol{x_i}, y_i)\}_{i=1}^m$, where $\boldsymbol{x_l} \in \mathbb{R}^{ld}$ and $y_i \in \{C_k\}_{k=1}^K$.
**Output:** Subspaces $\{S_k\}_{k=1}^K$ for classes $K$.

  1: $\mathcal{D}_{\text{new}} \leftarrow \mathcal{D}_l$
  2: **for** each $(\boldsymbol{x_i}, \boldsymbol{y_i}) \in \mathcal{D}_l$ **do**
  3:     $\mathcal{D}_{\text{new}} \leftarrow \mathcal{D}_{\text{new}} \cup \{(-\boldsymbol{x_i}, \boldsymbol{y_i})\}$
  4: **end for**
  5: **for** each $k \in \{1, \ldots, K\}$ **do**
  6:     Extract the subset of data $\mathcal{D}_{\text{new},k} = \{\boldsymbol{x_i} \mid y_i = C_k\}$
  7:     $S_k = \text{PCA}(\mathcal{D}_{\text{new},k})$
  8: **end for**
  9: **Return:** Subspaces $\{S_k\}_{k=1}^K$

---

determined for the training data and the test data separately. This process is followed for all the layers in the network independently. MASC is using labels of the dataset while creating the class-specific subspaces. The process of creation and use of subspaces with MASC for a new data point are shown schematically in Figure 2.

We apply MASC on each layer of the network with respect to different subspaces. For MLP models, all the MASC experiments were performed for all the layers in the network including on the input (after it is pre-processed). For CNN models and AlexNet models, the experiments were performed on flatten layer (Flat) and fully connected layers (FC). For ResNet-18 model, we evaluated nine layers – L0, L0-1, L0-2, L0-3, L1, L2, L3, L4, and the average-pool (avg_pool) layer – containing 16,384; 16,384; 16,384, 16,384; 16,384; 8,192; 4,096; 2,048; and 512 neurons respectively. Here, L0 denotes the layer immediately before the L1 block; L0-1, L0-2, and L0-3 are intermediate outputs within the L1 block; and L1–L4 correspond to the outputs of successive residual blocks. All layer outputs were flattened prior to analysis. While we ran the experiments on the input layer for CNNs, we did not do so for AlexNet or ResNet-18.

### 3.4 Leveraging MASC to retrain the model

Here, the idea is to use one of the layerwise MASC classifiers in order to relabel the corrupted training set. This relabeled training set is then used to retrain the existing model. To determine the layer whose MASC classifier we will use, we find the layer whose MASC classifier generalizes best. To this end, we first split the test data set into 80%-20%. We use the MASC accuracy on the corrupted subspaces in the validation set (created from 20% of the test dataset) to identify the model's best-layer. Then, using the best-layer MASC predictions, we relabel the corrupted training dataset. We train with the relabeled corrupted training dataset for upto 30 epochs and perform early stopping with patience of 3 by considering the 20% test dataset as a validation dataset; this validation dataset was not used in reporting test accuracy. A similar process was followed while working with subspaces corresponding to true labels. The test accuracy on the models is

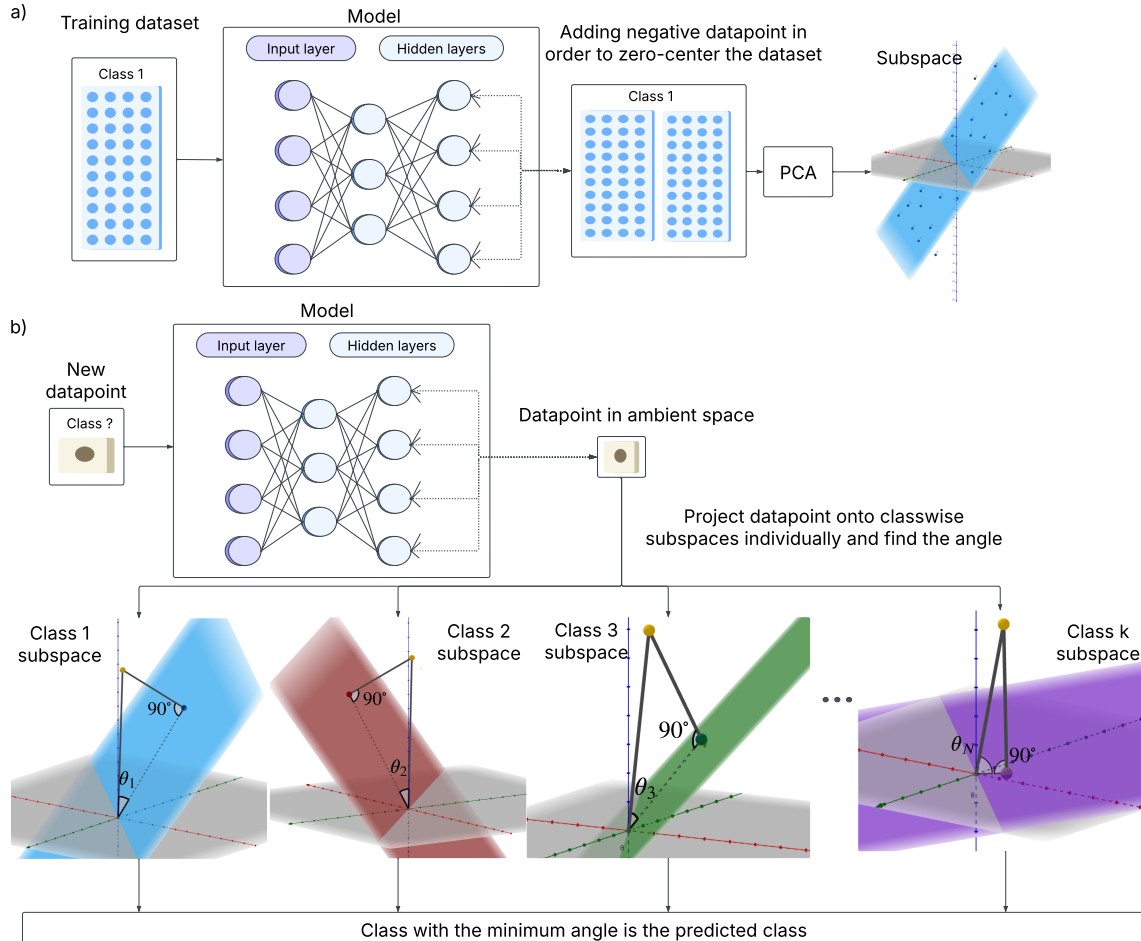

Figure 2: A schematic of the Minimum Angle Subspace Classifier (MASC) constructed for a specific hidden layer. a) Illustration of the process of fitting a subspace (i.e. a linear space that passes through the origin) corresponding to a single class, for the outputs of a specific hidden layer. For a specific layer, class-wise training dataset (Class 1) is passed through the model till the hidden layer in question. For every output datapoint (activation values) of the layer, a negative data point is added to the point set in order to zero-center the outputs / dataset before performing Principal Components Analysis (PCA).
b) Such subspaces for the hidden layer are constructed for every class. When a new (e.g. unseen) datapoint needs to be classified by MASC, its output from the hidden layer is computed, which is a datapoint in the ambient space of the hidden layer. This datapoint is projected onto the individual class-conditional subspaces and the angle between the data point and its respective projections, $\theta_1, \ldots \theta_k$, are determined. MASC predicts the datapoint's class to be the one whose subspace the datapoint has the smallest such angle with, i.e $\arg\min_i\{\theta_i\}$.

calculated with respect to the 80% test dataset, obtained in the aforementioned split. A schematic of the retraining process using MASC is shown in Figure 9.

## 4 Enhanced innate generalization to true labels in memorized models

Models trained with corrupted labels have high training accuracy (on corrupted labels) while also having low accuracy on the test set with true labels (Zhang et al., 2017; 2021). We ask if we can decode the representations of the hidden layers of these memorized models to obtain better generalization to true labels.

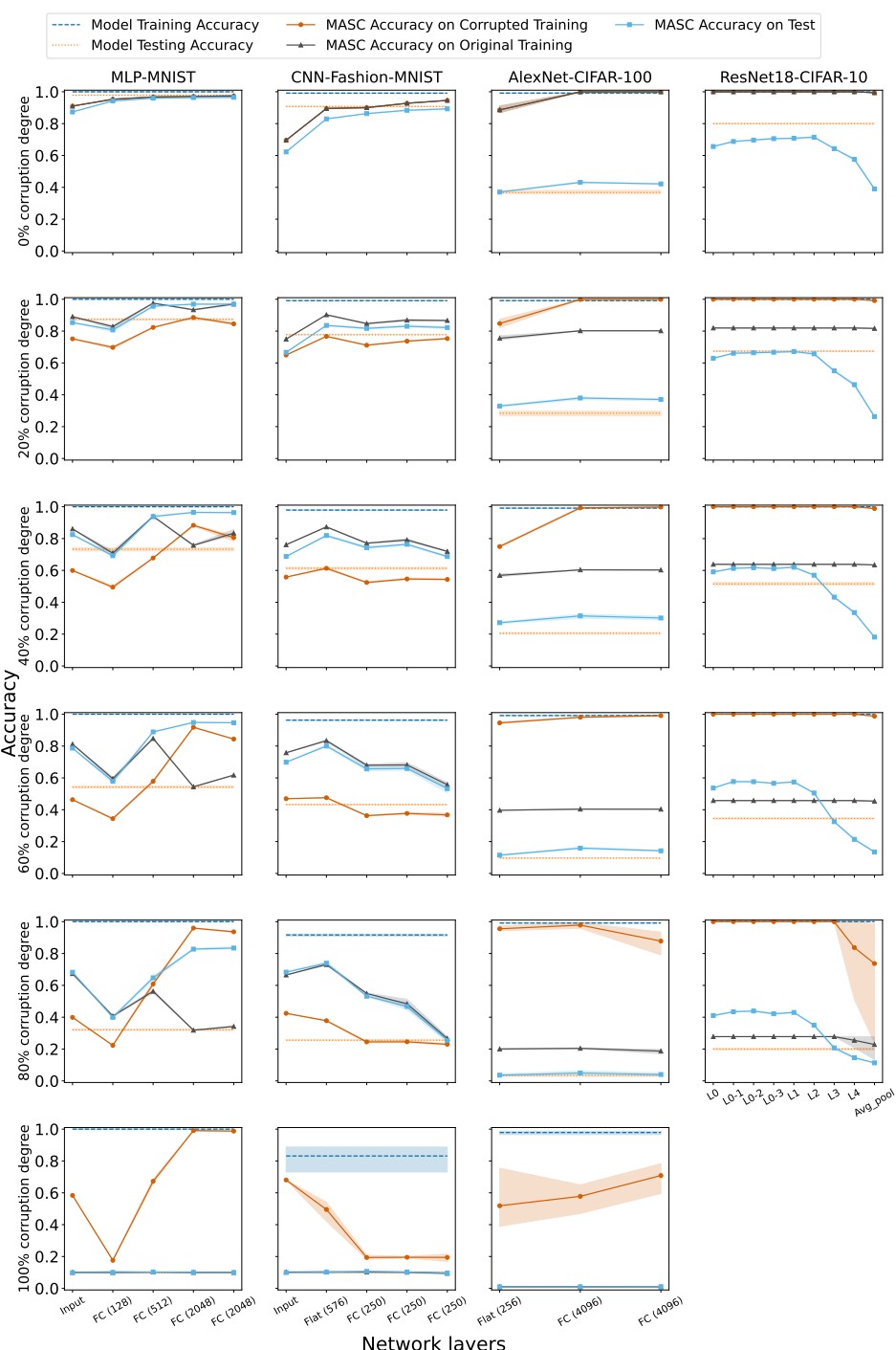

Figure 3: Minimum Angle Subspace Classifier (MASC) accuracy over the layers of the network when the data is projected onto corrupted training subspaces with the indicated corruption degree, for multiple models/datasets. Rows corresponds to plots with the same corruption degree & the columns correspond to the models, as noted. Training accuracy (dashed line) & test accuracy (dotted line) of the model is shown. FC corresponds to fully connected layer with *ReLU* activation whereas Flat corresponds to flatten layer without *ReLU* activation. The number of class-wise PCA components of these models are shown in Figure 30. *SGD* optimizer (Qian, 1999) was used for training MLP models, whereas *Adam* optimizer (Kingma, 2014) was used for other models. ResNet-18 has layer outputs of size 16,384 for L0-L1, followed by 8,192 (L2), 4,096 (L3), 2,048 (L4), and 512 (avg_pool) layer.

To do so, we build a probe that we call Minimum Angle Subspace Classifier (MASC) using class-conditioned corrupted training subspaces obtained from the memorized models' hidden layer outputs. MASC is performed layer-wise for the layers of the network independently. More details on MASC are available in the Methods section. MASC accuracy on corrupted training data, MASC accuracy on original training data (with true labels), and MASC accuracy on test data (with true labels) over the layers of MLP trained on MNIST, CNN trained on Fashion-MNIST, AlexNet trained on CIFAR-100, and ResNet-18 trained on CIFAR-10 for various randomly-chosen fractions of label corruption in training data (i.e. corruption degrees) are shown in Figure 3. Likewise, results for MLP trained on CIFAR-10, CNN trained on MNIST & CIFAR-10 and AlexNet trained on Tiny ImageNet are presented in Figure 27.

Importantly, for every corrupted model we have (with non-zero corruption degree), except those with 100% corruption degree, we find that our Minimum Angle Subspace Classifier (MASC) in at least one layer (with one exception[8]) has better test accuracy than the corresponding model itself. Table 1 reports by what percentage the MASC classifier outperformed the model for the best such layer, for each model. In Table 6, we also report the accuracy difference between the MASC classifier and the model for the best such layer, for each model. In many cases, the MASC test accuracy is dramatically better than that of the model. This is remarkable, because, in addition to the layerwise outputs, MASC used precisely the same information (including the same corrupted training dataset) that was available to the model itself, and yet is able to extract better generalization to true labels. This suggests that the model retains significant latent generalization to true labels, which is not captured in its own test-set performance. In many models, the same MASC, especially on the later layers, also approaches perfect accuracy on the corrupted training set, indicating that this improved generalization to true labels can happen concurrently with memorization of training data points with shuffled labels. Below, we make more specific observations on the performance of the models.

Table 1: Percentage by which the MASC classifier (run on the best layer) outperformed the model's test accuracy when the data is projected onto corrupted training subspaces. The best layer corresponds to the one that has the highest measured MASC test accuracy among the layers for the said model/dataset. The accuracies in each case are averaged over three runs and are rounded to the second decimal place. Some of the detailed results are available in Appendix, as indicated.

| Corruption degree | 20% | 40% | 60% | 80% |
|---|---|---|---|---|
| MLP-MNIST | 10.93% | 31.63% | 75.04% | 159.93% |
| MLP-CIFAR-10 (Appendix) | 9.90% | 24.42% | 46.97% | 64.75% |
| CNN-MNIST (Appendix) | 9.81% | 37.03% | 98.69% | 201.06% |
| CNN-Fashion-MNIST | 7.49% | 33.50% | 84.93% | 189.00% |
| CNN-CIFAR-10 (Appendix) | 2.29% | 6.26% | 27.03% | 60.17% |
| AlexNet-CIFAR-100 | 33.58% | 53.10% | 64.86% | 45.00% |
| AlexNet-Tiny ImageNet (Appendix) | 27.50% | 53.46% | 45.38% | 14.16% |
| ResNet-18-CIFAR-10 | -0.41% | 20.34% | 67.28% | 119.16% |

With generalized models i.e. those with 0% corruption degree, at the later layers of the network, it is observed that in most of the cases MASC accuracy on training data approaches the models training accuracy. Similarly, MASC accuracy on test dataset is comparable to or performed better than the models' test accuracy, with the exception of the ResNet-18 model.

Even for high corruption degrees, we find that MASC performs well. For example, with 80% corruption degree, which implies that approximately 72% of the training labels have been changed, we observed good MASC test accuracy in many cases. Notably, the MASC test accuracy on the later layers is over 80% on MLP-MNIST, in comparison to 34% test accuracy by the model. Similarly, MASC test accuracy on one of the layers is about 75% for CNN-Fashion-MNIST, in contrast to 25% model test accuracy. Even for larger

---

[8]ResNet-18 trained on CIFAR-10 with 20% corruption degree is the lone exception. See Figure 3 and Table 1 for the corresponding results.

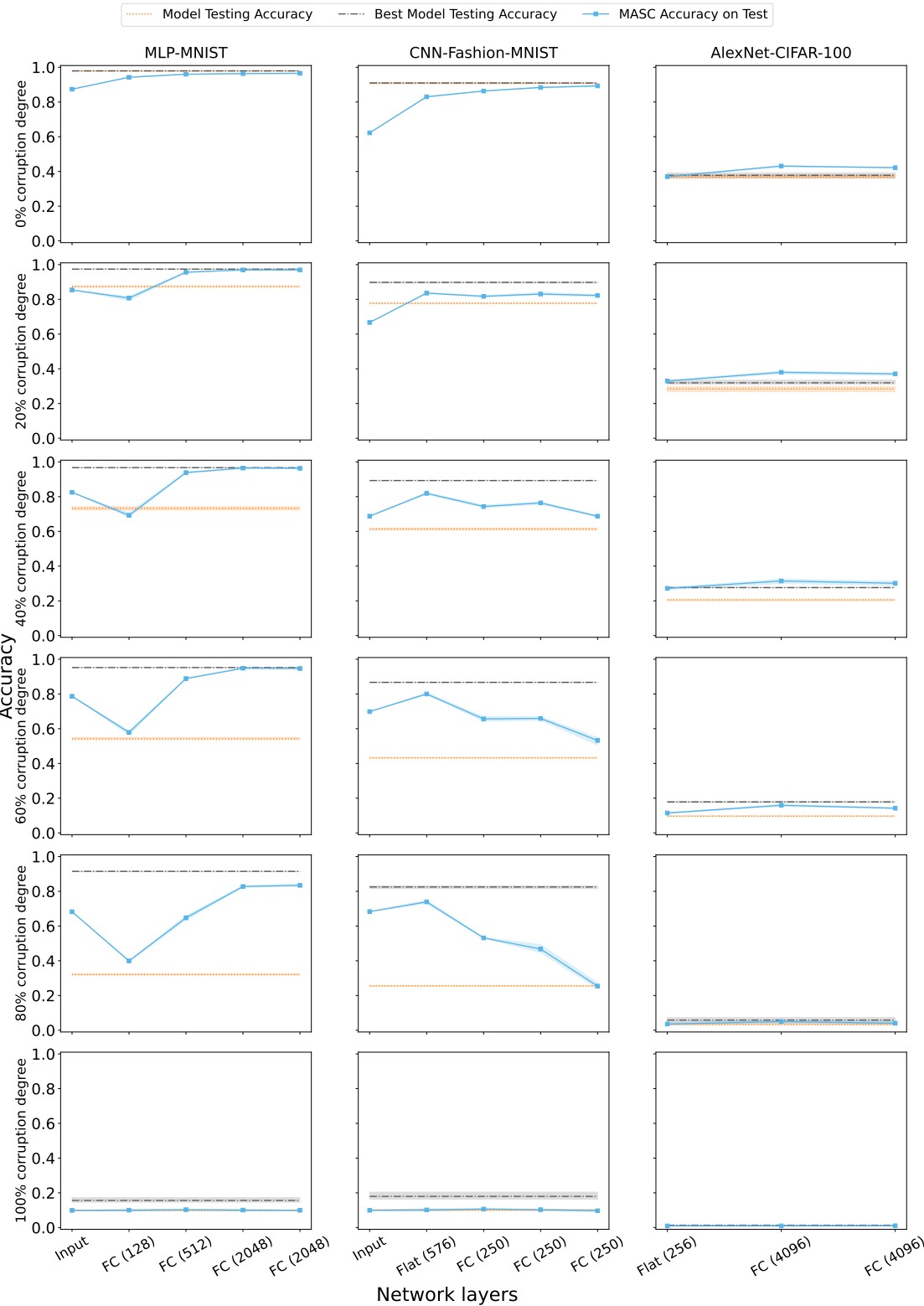

Figure 4: MASC test accuracy over the layers of the network when the data is projected onto corrupted training subspaces with the indicated corruption degree. Best model test accuracy corresponds accuracy of the test data of the model if early stopping was used.

models/datasets such as AlexNet-CIFAR-100, MASC test accuracy outperforms the model test accuracy by 45%, for training sets with 80% corruption degree. Likewise, for ResNet-18-CIFAR-10, several layers exhibit MASC test accuracies that exceed the model's test accuracy.

Not only does MASC have better accuracy than the model on the test data but, when applied to some layers, it also does well on the training data with the true labels. Although the model has memorized the training data with corrupted labels, outputs from certain layers have the ability to predict the trained true labels. For example, in MLP-MNIST, for low to moderate degrees of corruption, MASC on the middle layer (FC (512)) has good accuracy on the true training labels, while also retaining good accuracy on the test set. With 40% corruption degree, approximately 36% are changed labels and yet the model has good accuracy on the true training labels in at least one layer of the network. e.g. MLP-MNIST has over 90% true training accuracy at layer FC(512), CNN-Fashion-MNIST has approximately 85% in Flat (576) layer & AlexNet-CIFAR-100 has approximately 60% in FC (4096) layer. This means that almost 20% of those labels are predicted correctly even though the model was trained for 500 epochs or has reached high training accuracy on corrupted labels. In the process of doing this, the model does not have any direct information about the true labels and neither does MASC.

For a subset of models, we also compare the results of test accuracy of MASC on trained model with early stopping model accuracy. MASC accuracy over the layers of the network when the data is projected onto corrupted training subspaces is shown in Figure 4. Best model test accuracy and trained model test accuracy are shown for reference. Best model test accuracy corresponds to the accuracy of the test data of the model if early stopping was used.

For MASC when the data is projected onto corrupted training subspace, in AlexNet-CIFAR-100, the MASC in at least one layer shows better performance than the best model test accuracy for less than 60% corruption degree. For MLP-MNIST, the best model (early stopping) maintains over 90% accuracy even when the data is corrupted up to 80%. Despite the increase in corruptions (except 100% corruption), the accuracy of the last layer remains close to that of the best model accuracy. For CNN-Fashion-MNIST (except 100% corruption), in at least one layer MASC performance is near to that of best model test accuracy.

One way to think about a deep network, is as one that successively transforms input representations in a manner that aids in good prediction performance. Therefore, performance of MASC on the input is a good baseline measure to assess if subsequent layers have favorable accuracies. Naively, for models trained with corrupted data, one would expect layered representations that enable the model to do well on the corrupted training data, but not do well on the test/training data that have true labels. While this expectation seems to hold with respect to the model itself, we find that the layer-wise representations do not necessarily follow this expectation. That is, MASC applied to subsequent layers, often have better true training accuracy and test accuracy than MASC applied to the input, suggesting that the deep network does indeed transform the data in a manner amenable to better generalization to true labels, even if its labels are dominated by noise.

## 5 Generalization to true labels comparison: MASC versus other probes

Given that MASC is a probe on layerwise outputs, it is natural to ask how a few other probes might perform in the memorization setting. Accordingly, we have used five different probes[9] on the layer of the deep neural networks namely, Logistic Regression (LR) , K-Nearest Neighbor (KNN), Linear Discriminant Analysis (LDA), Quadratic Discriminant Analysis (QDA), and Nearest Class Mean (NCM). Figure 5 reports the test accuracies of all probes across the layers of MLP-MNIST, CNN-Fashion-MNIST, and AlexNet-CIFAR-100. Model and MASC test accuracies are overlaid for comparison. Results for additional models are provided in Section C.2. We also report computational cost (GFLOPS) for different probes over the layer of the network in Section C.3.

We find that, although all probes achieve broadly comparable test accuracies – with no consistent pattern regarding which probe performs best at different corruption levels – their computational costs (GFLOPs) differ substantially. For AlexNet, the probes ranked from highest to lowest computational cost are: QDA, LDA, KNN, MASC, LR, and NCM. For the CNN models, the ordering is KNN, followed by QDA and LDA

---

[9]Experimental setup is provided in Section C.1

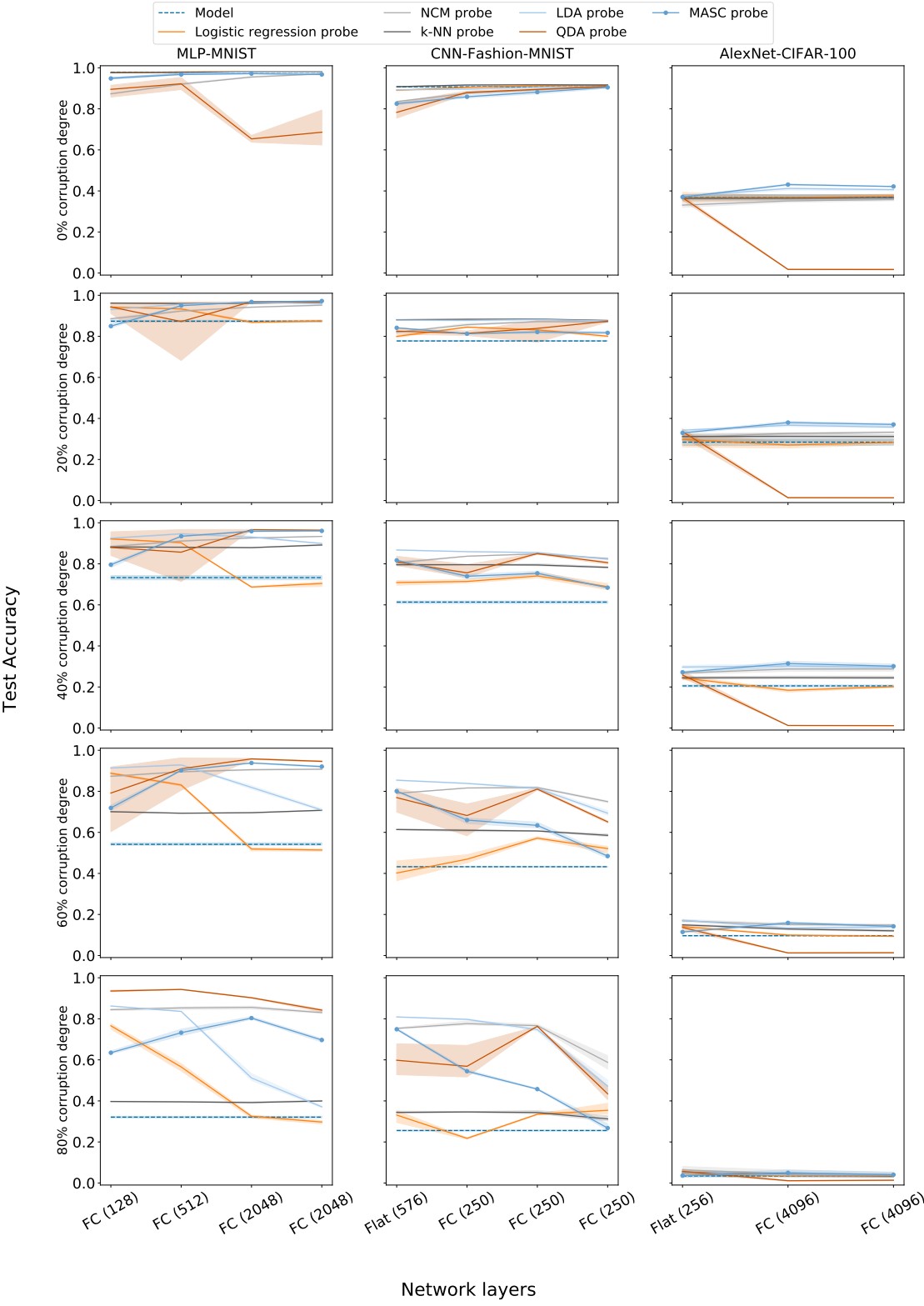

Figure 5: Test accuracies for different probes over the layers of the network. Rows corresponds to plots which have the same corruption degree and the columns correspond to the models as noted. Test accuracy of the model and MASC are shown for comparison. FC corresponds to fully connected layer with ReLU activation whereas Flat corresponds to flatten layer without ReLU activation.

(which have identical cost), MASC, LR and NCM. For the MLP models, the sequence is KNN, QDA, LDA, MASC, LR, and NCM. Across all models, MASC consistently exhibits lower computational cost compared to the KNN, QDA, and LDA probes, although its cost remains higher than that of LR and NCM.

Given this computational profile, we focus on results comparing the test accuracy of MASC with LR and NCM; the corresponding results are detailed in Section C.4. The three probes display distinct trends across layers and corruption degrees. On AlexNet–CIFAR-100 and MLP–MNIST at 20%–40% corruption, MASC surpasses both NCM and LR on most layers. For CNN-based models, NCM generally performs best, with MASC typically ranking above LR. Notably, MASC matches or exceeds NCM in specific cases such as CNN–MNIST and CNN–Fashion-MNIST at 20%, 40%, and 60% corruption, at the Flat(576) layer.

While our initial focus had been on MASC, it is interesting that other probes also have comparable performance, and indeed, this performance isn't always correlated with that of MASC. That is, in some cases, these probes perform better than MASC in some layers and worse than MASC in other layers. This suggests that latent representations that contain information useful for generalization to true labels may manifest in different forms, which result in different probes decoding them with differing accuracies. This phenomenon requires deeper investigation, which has been beyond the scope of the present paper.

## 6 Generalization to true labels via true training label subspaces in memorized models

While Section 4 demonstrated improved generalization to true labels by MASC, here, we investigate if there exist subspaces that can offer even better generalization to true labels. To this end, we consider the setting where the true label identities of the training set are known, after training with corrupted labels is complete. Can we extract significantly high training as well as test performance in this case from the layerwise outputs of the network? To do so, we build MASC using subspaces obtained from training data with true labels. It is a priori unclear if MASCs trained in this manner will have high accuracy. Since the network trained assuming different labels for many of the datapoints, it is conceivable that class-wise subspaces corresponding to true labels lack structure and predictive power. We find, however, that these possibilities do not bear out.

MASC accuracy on original training data and on test data projected on true training label subspace over the layers of the same networks is shown in Figure 6 and Figure 28. For comparison, MASC accuracy on corrupted training data and test data projected on corrupted training subspace is also shown. We find that, in many cases, accuracies on the true training labels, as well as the test set are dramatically better here than with the experiments where subspaces were determined for the corrupted training data. In Table 7, we show by what percentage the MASC classifier outperformed the model for the best layer for corruption degrees 20%, 40%, 60% and 80%. In Table 8, we also report the accuracy difference between the MASC classifier and the model for the best such layer, for each model. In fact, the MASC test accuracies for the corrupted models (with non-zero corruption degree) are sometimes fairly close to the test accuracy of the uncorrupted model.

Notably, even for models trained with 100% corruption degree, in most cases, MASC retains significant accuracy on the true training labels as well as the test set. This is in spite of the fact that the model itself has chance-level test-set accuracy. For example, MASC classifier has 95% test accuracy in the last FC(2048) layer for MLP-MNIST, 69% test accuracy for Flat(576) layer in CNN-Fashion-MNIST, and 4% test accuracy for Flat(256) layer in AlexNet-CIFAR-100.

The results here are proof of principle that suggest the existence of subspaces which allow one to extract significantly high generalization to true labels on models trained with datapoints whose labels are shuffled to a remarkably high degree. This has two implications. On the one hand, it demonstrates that models trained with very high label noise, surprisingly, retain the latent ability to generalize very well. On the other hand, it suggests that development of new techniques to identify favorable subspaces could help markedly boost generalization to true labels of models, whose training data is known to have label noise.

The results comparing MASC test accuracy over the layers of the network when the data is projected onto true training subspaces with early stopping model accuracy is shown in Figure 7.

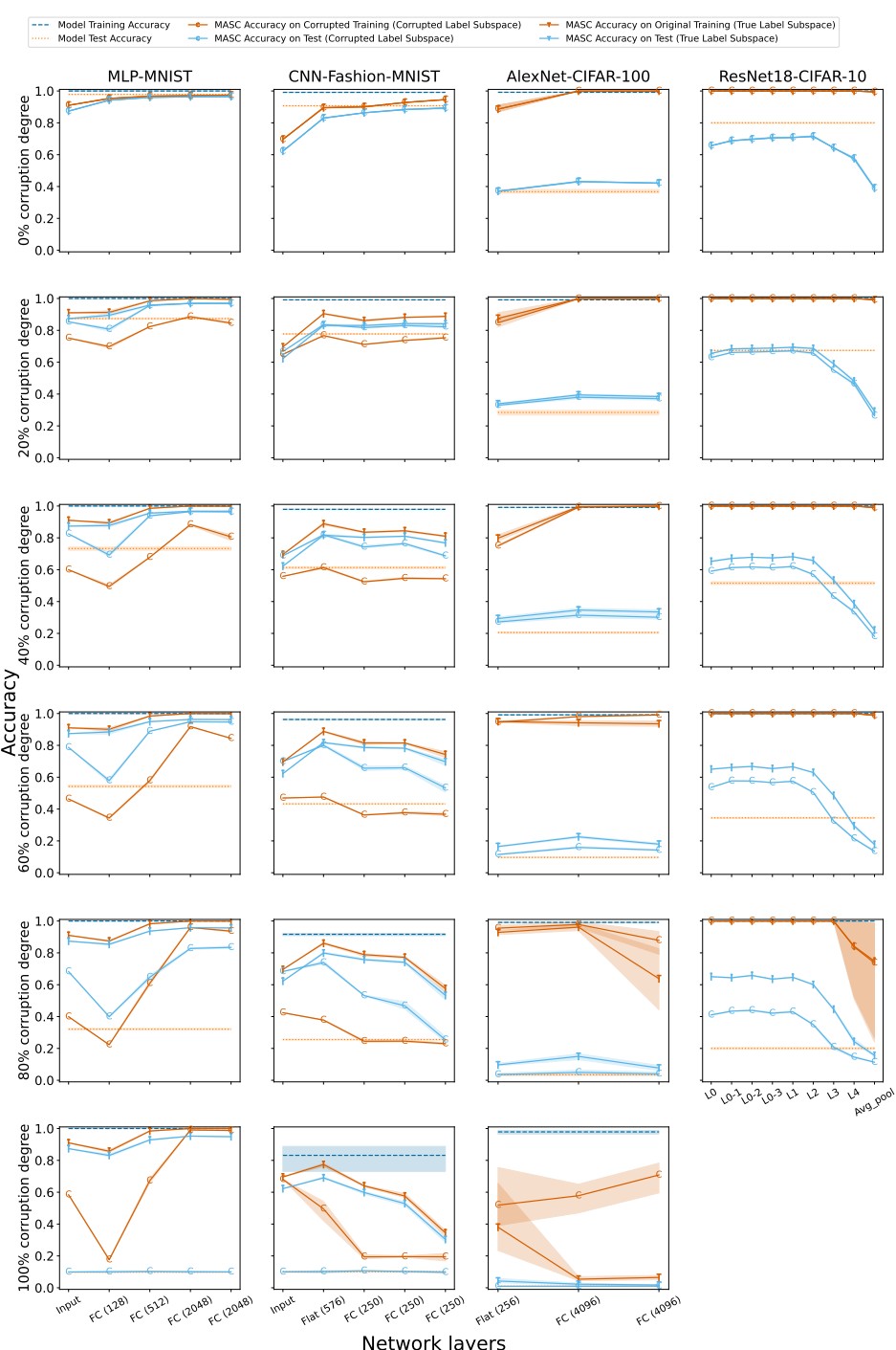

Figure 6: Minimum Angle Subspace Classifier (MASC) accuracy over the layers of the network when the data set is projected onto corrupted subspace and subspace corresponding to true training labels. Rows corresponds to plots which have the same corruption degree and the columns correspond to the models as noted. Training and test accuracy of the model is shown. FC corresponds to fully connected layer with ReLU activation whereas Flat corresponds to flatten layer without ReLU activation. The respective number of class-wise PCA components for true training label subspaces of the models is shown in Figure 32. *SGD* optimizer was used for training MLP models, whereas *Adam* optimizer was used for other models. ResNet-18 has layer outputs of size 16,384 for L0-L1, followed by 8,192 (L2), 4,096 (L3), 2,048 (L4), and 512 (avg_pool) layer.

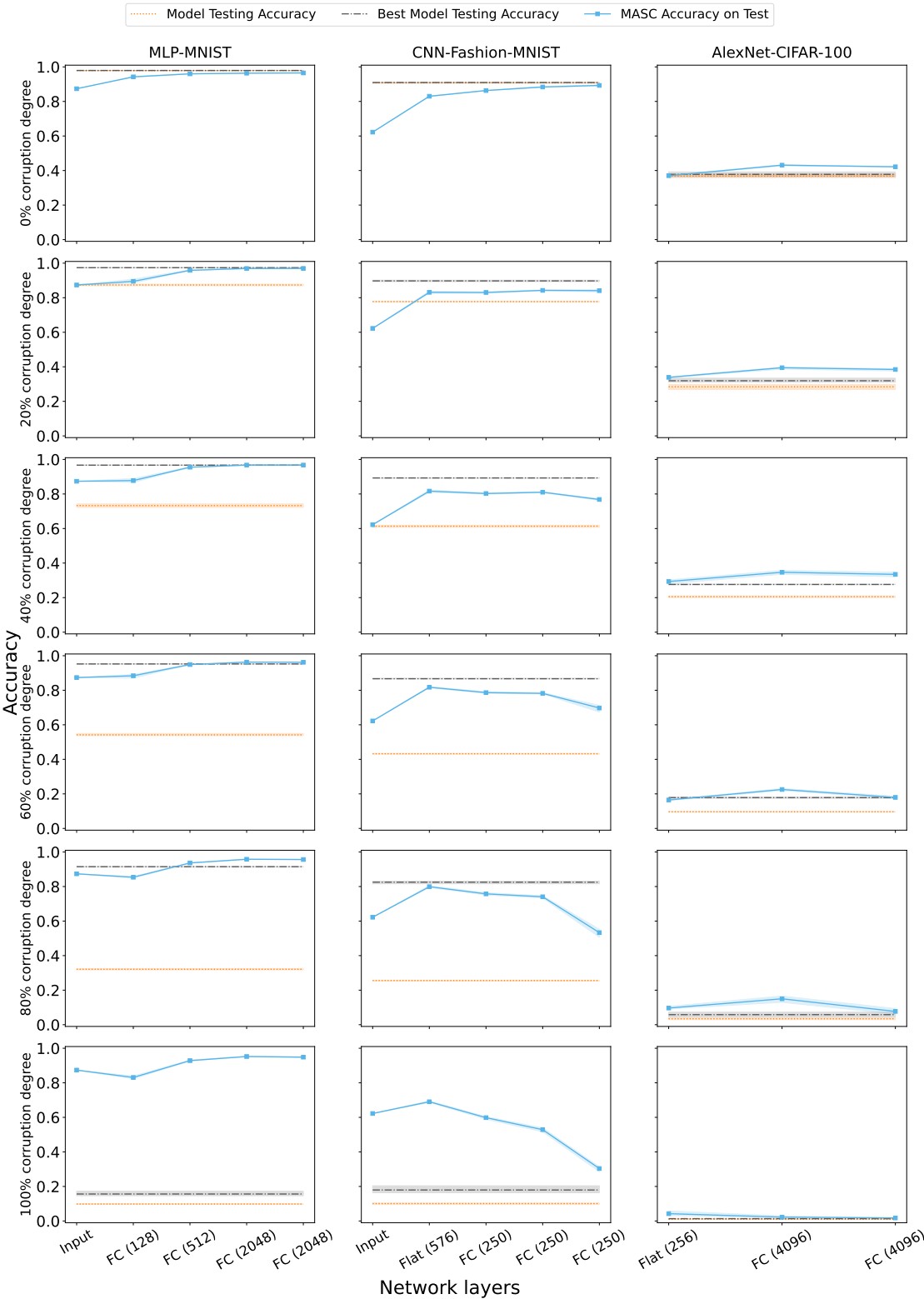

Figure 7: MASC test accuracy over the layers of the network when the data set is projected onto subspace corresponding to true training labels. Best model test accuracy corresponds accuracy of the test data of the model if early stopping was used.

The Appendix presents MASC comparison results across different PCA variance thresholds, along with an analysis of MASC's computational and time complexity. The Appendix also describes a control experiment with MASC accuracies on a random initialization of the network. We have results corresponding to MLP trained on CIFAR-10, CNN trained on MNIST and CIFAR-10, and AlexNet trained on Tiny ImageNet in the Appendix for all the experiments. We also have a section comparing MLP models trained on MNIST and CIFAR-10 with SGD and Adam optimizer.

In Section H, we also investigating the latent memorization capabilities of uncorrupted models. Here, conversely, we ask how well a network trained on true labels can manifest memorization of an arbitrary relabeling of its training data. More specifically, we built a MASC classifier on a model trained on true training labels, with the goal of memorizing training data whose labels are corrupted to varying degrees post hoc. Interestingly, we find a dichotomy in model behavior here, with some models trained on specific datasets having the propensity to memorize to a high degree, whereas others not demonstrating such ability.

## 7 Leveraging MASC to retrain the base memorized model for improved generalization to true labels

Taking into account the better generalization to true labels using MASC on memorized models, in this section, we ask the following question. Can we use the MASC classifier to retrain the existing model to achieve better generalization to true labels?

Details of the pipeline for retraining existing models, leveraging MASC are already presented before. For different corruption degrees, test accuracy[10] before and after retraining with relabeled data using MASC (corrupted and true subspaces) for MLP-MNIST, MLP-CIFAR-10, CNN-MNIST, CNN-Fashion-MNIST, CNN-CIFAR-10 models are shown in Figure 8. Similar results for AlexNet-CIFAR-100, AlexNet-Tiny ImageNet models are shown in Figure 42.

In order to study the dynamics of accuracy during retraining, unencumbered by the early stopping criterion, we also performed a similar experiment without using early stopping, for 10 epochs. The results for model before training, model after retraining on MASC corrupted subspace, and model after retraining on MASC subspace with true labels over the 10 epochs for all model-dataset pairs with various corruption degrees are shown in Figure 46.

In Figure 8 and 42, we find that for some models (MLP-MNIST, CNN-MNIST, CNN-Fashion-MNIST, CNN-CIFAR-10) with non-zero corruption degrees, there is an improvement in the test accuracy of models retrained using relabelling with MASC on corrupted subspaces, in comparison to the models' test accuracy before retraining (existing models). Indeed, in some cases, the improvement is quite significant, especially for larger corruption degrees (that are below 100% corruption degree).

However, for some models (MLP-CIFAR-10, AlexNet-CIFAR100), the accuracy gains due to such retraining appear marginal. Indeed, in some cases (MLP-CIFAR10, AlexNet-Tiny ImageNet for 20% corruption degree), there is a decrease in the test accuracy with such retraining. In order to study why, we checked the fraction of incorrect labels in the relabeled training dataset and compared it with the same measure for the existing corrupted training dataset. For the corruption degrees 20%, 40%, 60%, and 80%, these results are plotted in Figure 43. Table 9 lists the exact values of the same.

In particular, it turns out that for MLP-CIFAR-10 and AlexNet-CIFAR-100 this fraction is almost equal to the fraction on the existing corrupted dataset; for MLP-CIFAR-10 the fraction is marginally higher for the relabeled dataset and for AlexNet-CIFAR-100, it is marginally lower. This simply implies that the MASC classifier on the best layer that uses corrupted subspaces does roughly as well on the training set with true labels as the existing model, while surprisingly, the same MASC classifier is able to perform significantly better than the existing model on the test-set with true labels (See light orange bar in Figure 8 and 42). With regard to retraining, it would seem that the relabeled training isn't more effective than the existing corrupted training set in training the model, which possibly reflects in the lack of significant improvement in test accuracy. More broadly, this suggests that MASC's better generalization to true labels isn't necessarily

---

[10]Note that the test accuracy on the models is calculated with respect to the 80% test dataset.

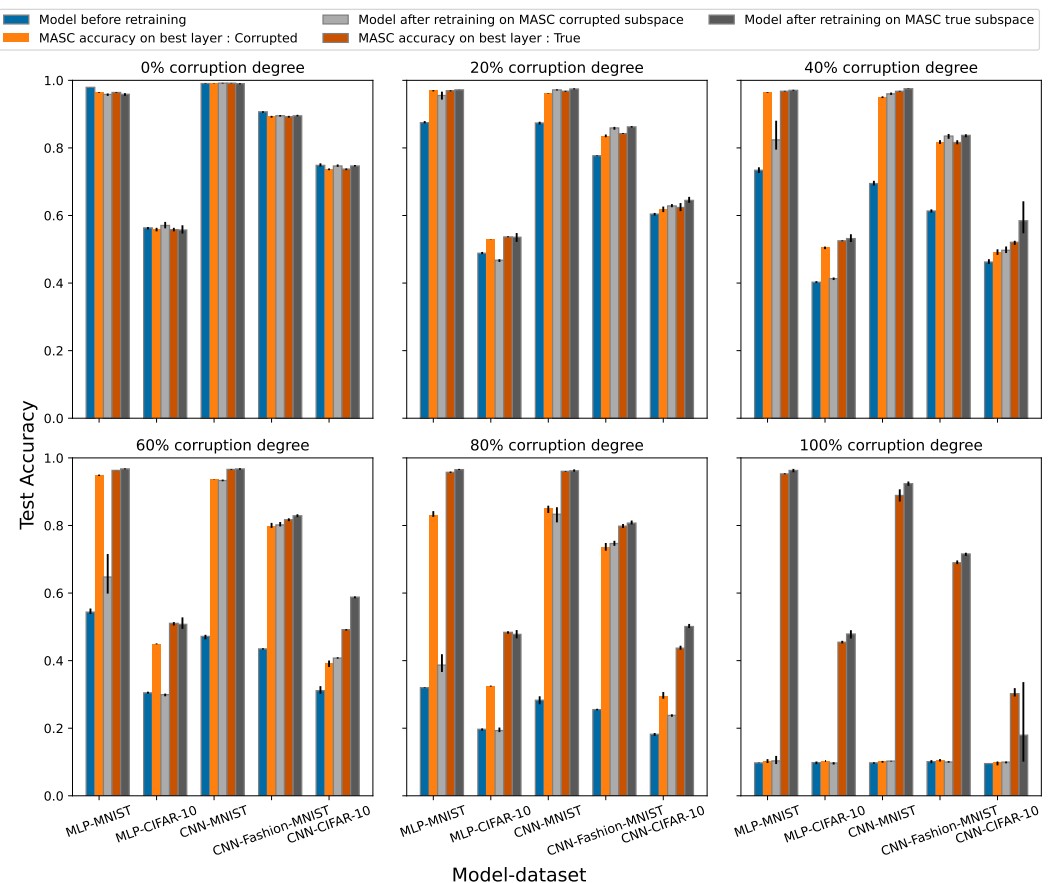

Figure 8: Test accuracies averaged over three runs on the 80% test dataset is plotted for different model-dataset pairs for various corruption degrees and for various models/MASC classifiers. Model before retraining corresponds to the existing memorized model. Model after retraining on MASC corrupted subspace corresponds to model trained with training dataset relabeled using MASC corrupted subspace predictions on the best layer. Model after retraining on MASC true subspace corresponds to model trained with training dataset relabeled using MASC subspaces corresponding to true label predictions on the best layer. MASC test accuracy on the best layers for corrupted and true label subspaces on existing corrupted models (before retraining) are shown for comparison. The best layer was identified using a validation set that was carved out of the test dataset, for this experiment. Error bar represents the range on three different runs.

accompanied by better training set performance on true labels. This phenomenon requires a more detailed future investigation.

Secondly, for AlexNet-Tiny ImageNet with 20% corruption degree, the relabeled training set has significantly fewer fraction of correct labels (92.46% lower) than in the existing corrupted training set. We wanted to determine to what extent this lower fraction is driven by previously incorrect label predictions (in the existing corrupted training dataset) being predicted correctly (in the relabeled set), versus previously correct predictions being relabeled incorrectly. For all models, these numbers are visualized in Figures 44 & 45 and Tables 10 & 11 list corresponding numbers. We find for the case of AlexNet-Tiny ImageNet with 20% corruption degree that there is a small fraction (5.94%) of previously incorrect labels that are correctly relabeled and a large fraction (24.46%) of correctly labeled points that are incorrectly labeled by the MASC classifier trained on the corrupted label subspaces. Notably, even though this MASC classifier while doing significantly worse on the training data than the model happens to do markedly better than the model in

test accuracy. As before, we think that the poor performance of the retrained model is driven by the fact that relabeling results in a dataset with larger fraction of incorrect labels.

Thirdly, in some cases (e.g., AlexNet-Tiny ImageNet with 40%, 60%, 80% corruption degree), we observe that even though relabeling results in a somewhat larger fraction of incorrect labels, the test accuracy of the retrained model is slightly better. This suggests that retraining performance is not simply driven by fraction of correct labels, but that specifics of which points are relabeled can drive retraining performance in ways that remain to be investigated.

With respect to models retrained on MASC true label subspaces, it was observed that the test accuracy of such models usually performed significantly better than the models before retraining.

By-and-large, we find that the retrained models that use MASC corrupted subspace relabeling don't have better test performance than the corresponding MASC classifiers. However, it would be interesting to apply MASC on the retrained models to see if that would further improve generalization to true labels performance.

## 8 Discussion

In this work, we investigated the phenomenon of memorized networks not generalizing well, asking why the ability to generalize is apparently diminished due to the act of memorizing. We find, surprisingly, that the intrinsic ability to generalize remains present to a degree not previously recognized, and this ability can be decoded from the internals of the network by straightforward means. On the one hand, we design probes that use the subspace geometry of the corrupted training data to decode such better generalization to true labels. We also demonstrate using true labels post hoc that there exist subspaces that allow for an even more improved decoding. Furthermore, we show that such decoding can be leveraged to retrain the models to have better generalization to true labels. We also show (Appendix) that the internal representations of some deep networks trained on true labels, possess the ability to substantially memorize relabelings of its training data. Moreover, the new type of probe – MASC – that we use here, is relatively lightweight computationally while being easy to implement and lending itself to a straightforward geometric interpretation.

In building MASC, we were motivated by the manifold hypothesis in machine learning Goodfellow et al. (2016) that posits that high-dimensional data typically reside on a low-dimensional manifold. It has also been suggested (Brahma et al., 2015) that such manifolds in layerwise representations flatten across layers of deep networks. Fitting manifolds can be computationally expensive, so we were interested in examining the organization of classwise data in subspaces, even if such subspaces might be somewhat higher dimensional than the corresponding manifolds. Indeed, this view leads to the natural idea of classifying unseen data points by determining which class manifold it is closest to. MASC is simply a formalization of this idea. In particular, this classifier lends strong geometric motivation and intuition, in contrast to e.g. training a standard linear probe that iteratively minimizes a crossentropy loss. However, the reasons for the success of MASC in this setting are still largely unclear to us. The difficulty is that the principles that underlie the nature of layerwise representations in deep networks trained with standard techniques are not well understood at this time and it appears that such representations play a significant role in the success of MASC in the memorization setting. Indeed, it is even a bit surprising that the deep network does not directly leverage this structure to obtain better generalization to true labels, although that may also be because its loss function aims to maximize training accuracy which might run counter to the act of bettering generalization to true labels.

An interesting question is about why this phenomenon even occurs; naïvely one would expect that deep networks, on being trained with noisy data, discard the ability to generalize in favor of learning noise. Are there specific inductive biases that promote such generalization to true labels? And, do such mechanisms also promote generalization to true labels in networks whose training data isn't corrupted significantly by such noise? It would also be instructive to study the dynamics of this form of generalization to true labels during training. It is known (Arpit et al., 2017) that the model's test accuracy transiently peaks in the early epochs of training with corrupted data, before dropping while training accuracy of the corrupted training data rises. It is unclear whether this transient rise in model generalization to true labels is caused by the subspace organization seen here, & if so, why such subspace organization isn't degraded as much as the model's test

accuracy over further epochs of training. Additionally, certain deep networks trained on specific uncorrupted datasets seem to possess internal representations that are amenable to significant memorization, whereas others aren't. The mechanistic basis of this ability is unclear & its possible connections to generalization to true labels in the such models merit further investigation.

It is interesting that probes other than MASC also often have generalization to true labels comparable to MASC, and this performance often manifests in layers different from those that have best accuracies of MASC. The reasons for this are unclear and remain to be investigated.

The work has a number of implications. On the one-hand, it suggests that the ability to memorize and generalize may not be antithetical. Indeed, in multiple cases, we are able to construct single MASC classifiers that perform well both on the shuffled training labels as well as on the held-out test data that has true labels. Secondly, theories proposed to explain generalization to true labels in deep networks have traditionally argued for the setting where the data distribution is well-behaved, i.e. corresponding to real-world data, but not for data with shuffled labels. We suggest, in light of the present results, that such theories also ought to be able to explain why networks retain the ability to generalize even in the face of noisy training data. That is, a satisfactory understanding of generalization to true labels in deep networks should also cover settings where the training data is noisy and its distribution is not well-behaved. Thirdly and more pragmatically, techniques such as the MASC classifier might suggest a way of boosting generalization to true labels in trained deep networks, whose training data intrinsically contains varying degrees of label noise. While this has been beyond the scope of the present paper, possibilities of designing new techniques for learning subspaces that have good generalization to true labels could be explored. Indeed, it is possible that significantly better subspaces exist than the ones uncovered here, & it would be interesting to see how much the generalization to true labels accuracy can be improved by pursuing this direction. Relatedly, it is possible that other classifiers operating on layerwise outputs have better performance than MASC – a possibility that merits further exploration. Fourthly, it would be interesting to formulate a measure to study representational similarity between memorized & generalized networks to see if they use similar mechanisms. Does the answer depend on the particular class of networks (e.g. MLPs vs. CNNs)?

Finally, the results here are reminiscent of a puzzling phenomenon observed in Neuroscience. In multiple settings (Miura et al., 2012; Stringer et al., 2021), in the rat olfactory system and the mouse visual system, it has been shown that a decoder using data from a subset of neurons from specific areas in the brain of a well-trained behaving animal has accuracy significantly better than the behavioral accuracy of the animal on novel trials, even though the animal is motivated to do well on the task. This implies that those animals have better innate generalization to true labels ability on that task – which can be easily decoded from a subset of their neurons – than is manifested by their behavior. It may therefore be that this is a phenomenon shared between brains and machines, whose underlying mechanisms and potential trade-offs remain to be investigated.

### Broader impact statement

The Minimum Angle Subspace Classifier (MASC) is primarily an analytical probing method, and its direct societal risks are therefore limited. However, caution is advised when applying it to models trained on sensitive or proprietary data. In addition, probe-derived labels should not be repurposed for model retraining without appropriate safeguards, as this may reinforce existing biases or propagate systematic mislabeling; robust validation procedures, uncertainty thresholds, and human oversight are recommended. Finally, the diagnostic performance of MASC should not be interpreted as equivalent to the underlying model's generalization to true labels ability. MASC's results are about understanding representations and not about predicting how the model will perform when deployed.

### Acknowledgments

Simran Ketha was supported by an APPCAIR Fellowship, from the Anuradha & Prashanth Palakurthi Centre for Artificial Intelligence Research. The work was supported in part by an Additional Competitive Research Grant from BITS to Venkatakrishnan Ramaswamy. The authors acknowledge the computing time provided on the High Performance Computing facility, Sharanga, at the Birla Institute of Technology and

Science - Pilani, Hyderabad Campus. We thank Harsha Varun Marisetty for compute assistance for training some of the models used here.

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

# A    Model details

Multilayer Perceptron (MLP) model has 4 hidden layers with 128, 512, 2048 and 2048 units respectively. *ReLU* activation was used after every layer and for classification *softmax* activation was applied. We have trained the models with two different optimizers namely, *SGD* and *Adam*. Learning rate of $1 \times 10^{-3}$ and momentum $= 0.9$ was used with *SGD* optimizer. Learning rate $= 1 \times 10^{-4}$ was used for *Adam* in experiments. Batch size of 32 was used in all the models. The dataset was normalized by dividing each pixel value with 255.

Convolutional Neural Network (CNN) network has 3 blocks, each consisting of two convolutional layers, one max pooling layer. These blocks are followed by three fully connected layers. Convolutional layers have 16, 32, and 64 filters, respectively with stride=1 and filter size $= 3 \times 3$. Max pooling layer has stride of 1 and filter size of $2 \times 2$. The fully connected layers at the end has 250 units each. It was trained with *Adam* optimizer with learning rate of 0.0002. For MNIST and Fashion-MNIST batch size of 32 whereas for CIFAR-10 batch size of 128 were used. The dataset was normalized by subtracting the mean and diving by the standard deviation for each channel. *ReLU* activation was used after every layer except pooling and *softmax* activation for classification.

AlexNet model was slightly modified for the use of each dataset. *Adam* optimizer with learning rate of 0.0001 was used. For CIFAR-100, batch size of 128 and for Tiny ImageNet, batch size of 500 was used. All the results with respect to test on AlexNet trained on Tiny ImageNet are shown with the validation dataset. CIFAR-100 dataset before training was normalized by subtracting the mean and diving by the standard deviation for each channel. No data normalization was performed on Tiny ImageNet dataset.

ResNet-18 model was slightly modified for the use of CIFAR-10 dataset. *SGD* optimizer with learning rate of 0.001 and momentum of 0.9 was used. Batch size of 32 was used for training. The dataset was normalized by subtracting the mean and diving by the standard deviation for each channel.

The experiments were performed on workstations/servers with a variety of GPUs, including Nvidia GeForce RTX3080s, GeForce RTX3090s, Tesla V100s and A100s.

A summary of the models and datasets with training set size and number of parameters is in Table 2. The average training and test accuracies of the models over three different runs used in the paper are shown in Tables 3 and 4.

Table 2: Training set size of the datasets and the number of parameters of the models.

| Model | Dataset | Training set size | Number of parameters |
|---|---|---|---|
| MLP | MNIST | 60,000 | 5,433,994 |
|  | CIFAR-10 | 50,000 | 5,726,858 |
| CNN | MNIST | 60,000 | 344,042 |
|  | Fashion MNIST | 60,000 | 344,042 |
|  | CIFAR-10 | 50,000 | 456,330 |
| AlexNet | CIFAR-100 | 50,000 | 38,738,952 |
|  | Tiny ImageNet | 100,000 | 39,776,464 |
| ResNet-18 | CIFAR-10 | 50,000 | 11,173,962 |

## A.1    General terminology

The general terminology used in this work is as follows:

- **Model Training Accuracy**: The accuracy of the model on the training set with corrupted labels.

- **Model Test Accuracy**: The accuracy of the model on the test dataset with true labels.

Table 3: Average training accuracy in percentages of all the models over three runs over different corruption degrees (indicated in the last six columns). The values are rounded to the second decimal place.

| Model | Dataset | Parameter | 0% | 20% | 40% | 60% | 80% | 100% |
|-------|---------|-----------|-----|-----|-----|-----|-----|------|
| MLP | MNIST | SGD | 99.99 | 99.99 | 99.99 | 99.99 | 100.0 | 100.0 |
| | | Adam | 100.0 | 99.87 | 99.73 | 99.77 | 99.73 | 99.66 |
| | CIFAR-10 | SGD | 99.99 | 99.99 | 99.99 | 99.99 | 99.99 | 99.99 |
| | | Adam | 99.63 | 99.53 | 99.43 | 99.61 | 99.52 | 30.21 |
| CNN | MNIST | Adam | 99.90 | 99.32 | 98.62 | 97.25 | 95.11 | 94.92 |
| | Fashion-MNIST | Adam | 99.15 | 99.14 | 97.90 | 96.25 | 91.65 | 83.14 |
| | CIFAR-10 | Adam | 99.70 | 99.29 | 99.26 | 99.03 | 99.02 | 39.69 |
| AlexNet | CIFAR-100 | Adam | 99.19 | 99.15 | 99.11 | 99.16 | 99.14 | 97.88 |
| | Tiny ImageNet | Adam | 99.92 | 99.90 | 99.91 | 99.93 | 87.71 | 85.95 |
| ResNet-18 | CIFAR-10 | SGD | 100.0 | 100.0 | 100.0 | 100.0 | 100.0 | - |

Table 4: Average test accuracy in percentages of all the models over three runs over different corruption degrees (indicated in the last six columns). The values are rounded to the second decimal place.

| Model | Dataset | Parameter | 0% | 20% | 40% | 60% | 80% | 100% |
|-------|---------|-----------|-----|-----|-----|-----|-----|------|
| MLP | MNIST | SGD | 97.87 | 87.38 | 73.28 | 54.16 | 32.09 | 9.81 |
| | | Adam | 98.31 | 90.49 | 76.78 | 55.59 | 31.85 | 9.77 |
| | CIFAR-10 | SGD | 56.37 | 48.62 | 40.35 | 30.55 | 19.68 | 9.80 |
| | | Adam | 52.24 | 44.11 | 35.55 | 25.24 | 15.18 | 10.07 |
| CNN | MNIST | Adam | 99.15 | 87.51 | 69.44 | 47.10 | 28.30 | 9.85 |
| | Fashion-MNIST | Adam | 90.74 | 77.74 | 61.35 | 43.26 | 25.57 | 10.08 |
| | CIFAR-10 | Adam | 74.95 | 60.48 | 46.15 | 30.96 | 18.32 | 9.89 |
| AlexNet | CIFAR-100 | Adam | 36.75 | 28.44 | 20.53 | 9.64 | 3.43 | 0.96 |
| | Tiny ImageNet | Adam | 15.88 | 9.74 | 5.44 | 2.02 | 0.73 | 0.43 |
| ResNet-18 | CIFAR-10 | SGD | 80.02 | 67.39 | 51.53 | 34.48 | 20.02 | - |

- **Minimum Angle Subspace Classifier (MASC) Accuracy on Corrupted Training**: Using Algorithm 1 (main paper), the accuracy of MASC predicted class labels with respect to corrupted labels of training dataset.

- **Minimum Angle Subspace Classifier (MASC) Accuracy on Original Training**: Using Algorithm 1 (main paper), the accuracy of MASC predicted class labels with respect to true labels of training dataset.

- **Minimum Angle Subspace Classifier (MASC) Accuracy on Test**: Using Algorithm 1 (main paper), the accuracy of MASC predicted class labels with respect to true labels of test dataset.

# B    Minimum Angle Subspace Classifier

A schematic illustrating the retraining process using Minimum Angle Subspace Classifier (MASC) is shown in Figure 9.

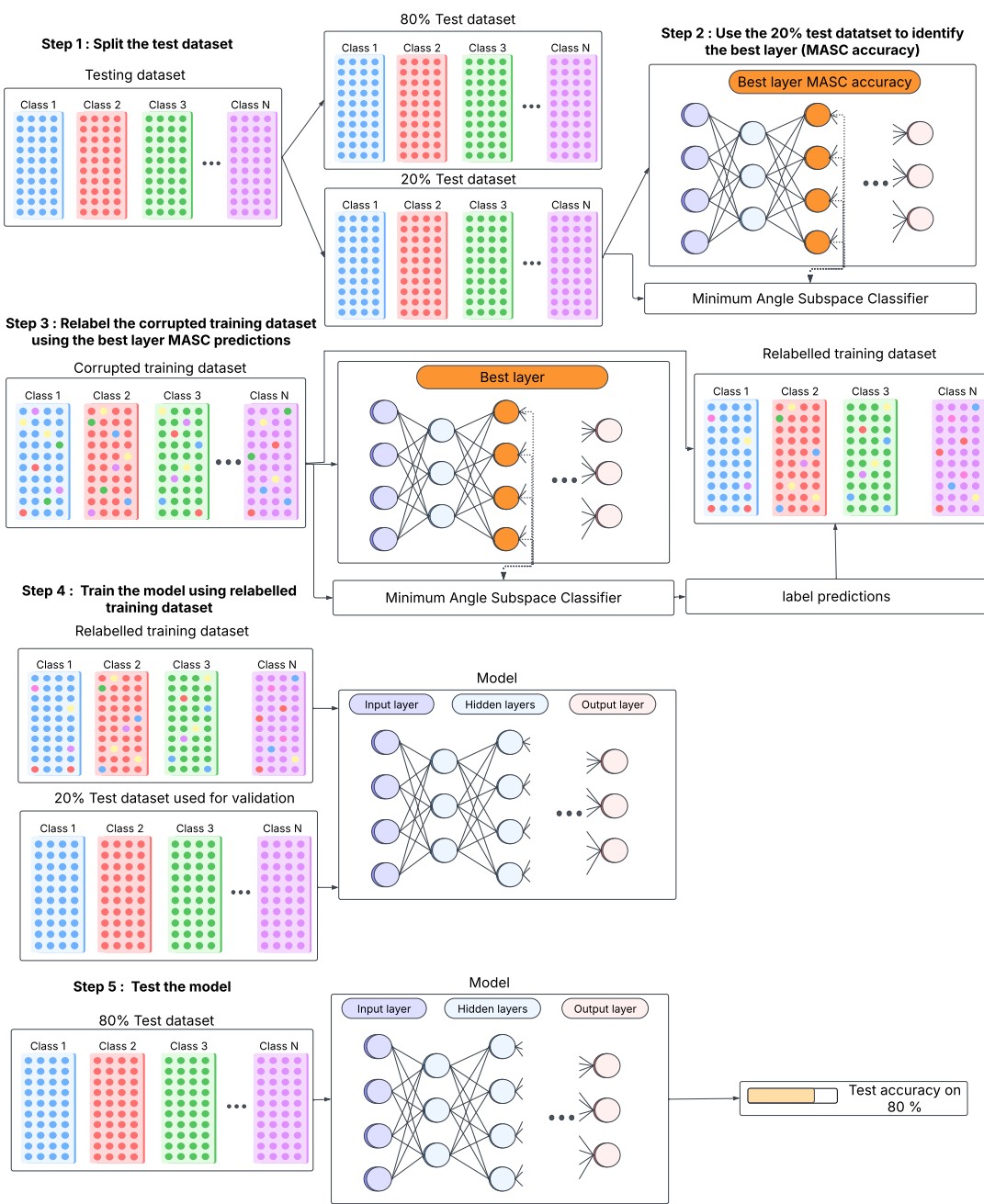

Figure 9: A schematic of the memorized model retraining pipeline that leverages MASC. Step 1: We first split the test dataset into 80%-20%. Step 2: We use the MASC accuracy on the corrupted subspaces on the 20% of the test dataset to identify the model's best-layer. Step 3: Using the best-layer MASC predictions, we relabel the corrupted training dataset. Step 4: We train with the relabeled corrupted training dataset for upto 30 epochs and perform early stopping with patience of 3 by considering the 20% test dataset as a validation dataset. A similar process was followed while working with subspaces corresponding to true labels. Step 5: The test accuracy on the models is calculated with respect to the 80% test dataset.

# C   Generalization to true labels comparison: MASC vs. other probes

## C.1   Experiment setup

For comparison, we evaluated five probes across all network layers: Logistic Regression (LR), K-Nearest Neighbors (KNN), Linear Discriminant Analysis (LDA), Quadratic Discriminant Analysis (QDA), and Nearest Class Mean (NCM).

For LR, we load the full corrupted training and test layer outputs along with labels, construct respectively data loaders with batch size 128, and train a PyTorch logistic regression model for 20 epochs using the *Adam* optimizer (learning rate $1 \times 10^{-3}$) with cross-entropy loss. The test loader is used for inference.

For NCM, we load the corrupted training and test layer outputs along with labels, compute class-wise mean vectors from the training data, and classify each test example by cosine similarity to these class means.

For LDA and QDA, we load the same training and test layer outputs along with labels. We use scikit-learn library to build and train the corresponding models on the corrupted training layer outputs. Evaluating is performed using the test layer outputs.

For KNN, we use scikit-learn library to build the model with n_neighbors=5 and cosine distance, following the same training and evaluation protocol as LDA.

## C.2   Results on additional models

Test accuracy of various probes over the layers of the MLP trained on CIFAR-10, CNN trained on MNIST and CIFAR-10, and AlexNet trained on Tiny ImageNet are shown in Figure 10. MASC test accuracy was overlaid for comparison.

## C.3   Computational cost across layers

The computational costs (in GFLOPs) for the various probes over the layers of the network are presented in Figure 11 for MLP–MNIST, CNN–Fashion-MNIST, and AlexNet–CIFAR-100, and in Figure 12 for the remaining models.

## C.4   Comparing generalization to true labels of MASC with logistic regression and nearest class mean

Test accuracy of Logistic Regression (LR), Nearest Class Mean (NCM) and MASC over the layers of the network for MLP-MNIST, CNN-Fashion-MNIST, and AlexNet-CIFAR-100 are shown in Figure 13 and for MLP-CIFAR-10, CNN-MNIST, CNN-CIFAR-10, and AlexNet-Tiny ImageNet are shown in Figure 14.

The three probes exhibit distinct behaviors across network layers and corruption degrees. For AlexNet–CIFAR-100 and MLP–MNIST at 20% and 40% corruption, MASC outperforms both NCM and LR on most layers. In contrast, for CNN-based models, NCM generally achieves higher accuracy than MASC across the majority of layers, while MASC typically performs better than LR. However, there are instances where MASC matches or surpasses NCM – for example, in CNN–MNIST and CNN–Fashion-MNIST at 20%, 40%, and 60% corruption degrees, at the FLAT (576) layer.

# D   MASC comparison results with various PCA variance thresholds.

In this section, we compare MASC performance using subspaces constructed with different variance thresholds: 99%, 75%, 50%, 25%, and a single principal component (1PC). Results for MLP–MNIST, CNN–Fashion-MNIST, and AlexNet–CIFAR-100 are presented in Figure 15, with additional model results shown in Figure 16.

For these experiments, a single trained model was used, and the MASC computations for the variance comparison were repeated across different seeds. To assess robustness with respect to randomness, we

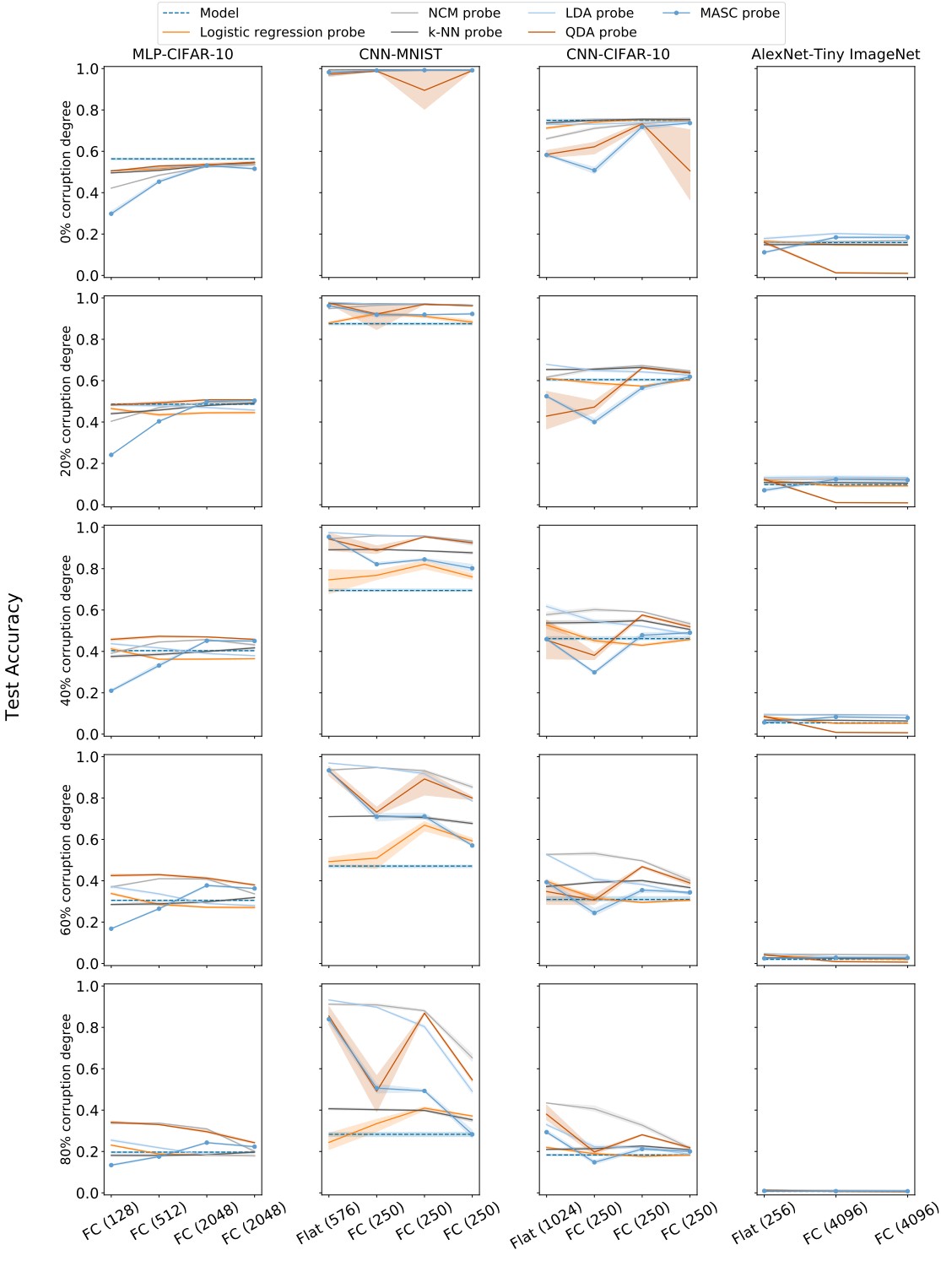

Figure 10: Test accuracies for different probes over the layers of the network. Rows corresponds to plots which have the same corruption degree and the columns correspond to the models as noted. Test accuracy of the model and MASC are shown for comparison. FC corresponds to fully connected layer with ReLU activation whereas Flat corresponds to flatten layer without ReLU activation.

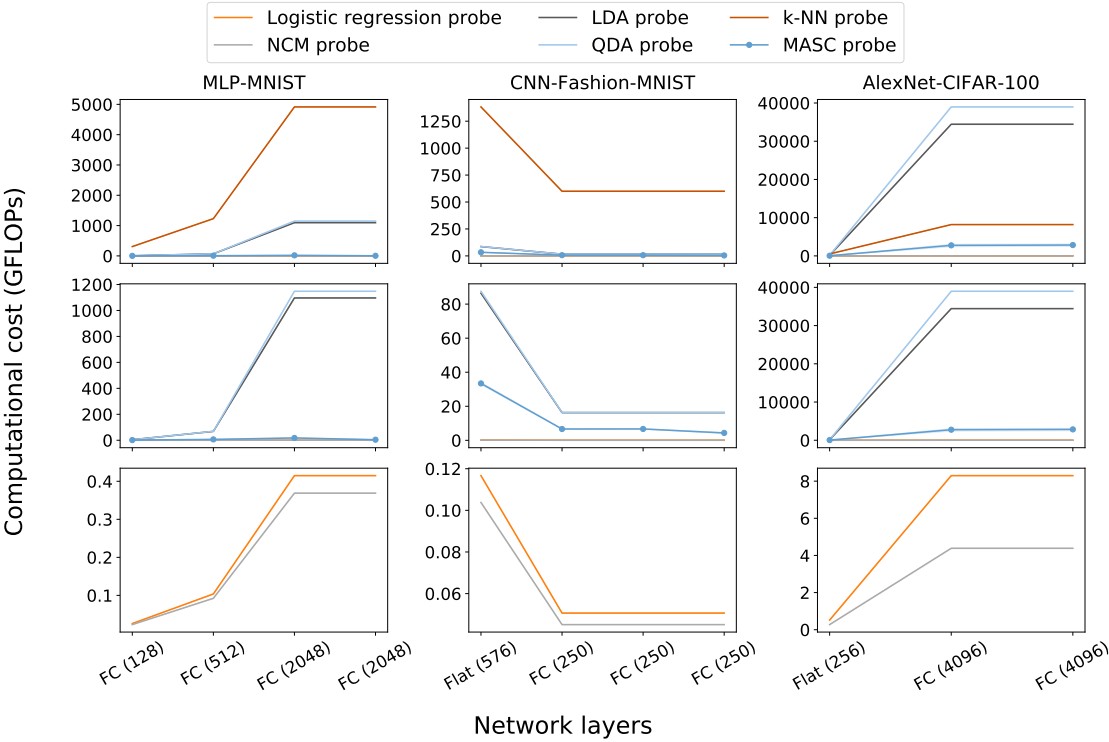

Figure 11: Computational costs (GFLOPS) for different probes over the layers of the network. Computational costs (GFLOPS) of MASC are shown for comparison. FC corresponds to fully connected layer with ReLU activation whereas Flat corresponds to flatten layer without ReLU activation.

performed five runs using seeds 10, 20, 30, 40, 50, and report the results as the mean along with the 95% confidence interval (CI).

For most models, we observed that under non-zero corruption degrees, several variance thresholds across many layers outperform the model's generalization to true labels. However, no consistent pattern emerges across all model–dataset pairs as the variance thresholds vary.

For MLP–MNIST and CNN–Fashion-MNIST, subspaces capturing variance of 75%, 50%, 25%, and even 1-PC often outperformed the subspace capturing 99% variance across multiple layers. In contrast, for AlexNet–CIFAR-100, the 99% variance threshold performed better than the other thresholds at 20% and 40% corruption degrees. Additionally, we found that the 1-PC subspace exceeded the performance of other thresholds in several models.

## E  Computational costs of MASC

In this section, we report total time taken (in seconds) and computation cost (GFLOPS) of MASC for building the subspaces and inference over the layers of the network, for multiple models/datasets are shown in Figure 17 and Figure 18. The computational cost (GFLOPS) for building per-class subspaces are shown in Figure 19.

It was observed that, although inference[11] requires more computational operations (in GFLOPs) than subspace construction, the total time taken for inference is nevertheless lower. This is likely because inference

---

[11]By inference in the context of MASC, we mean act of using the already determined subspaces to classify incoming data point(s).

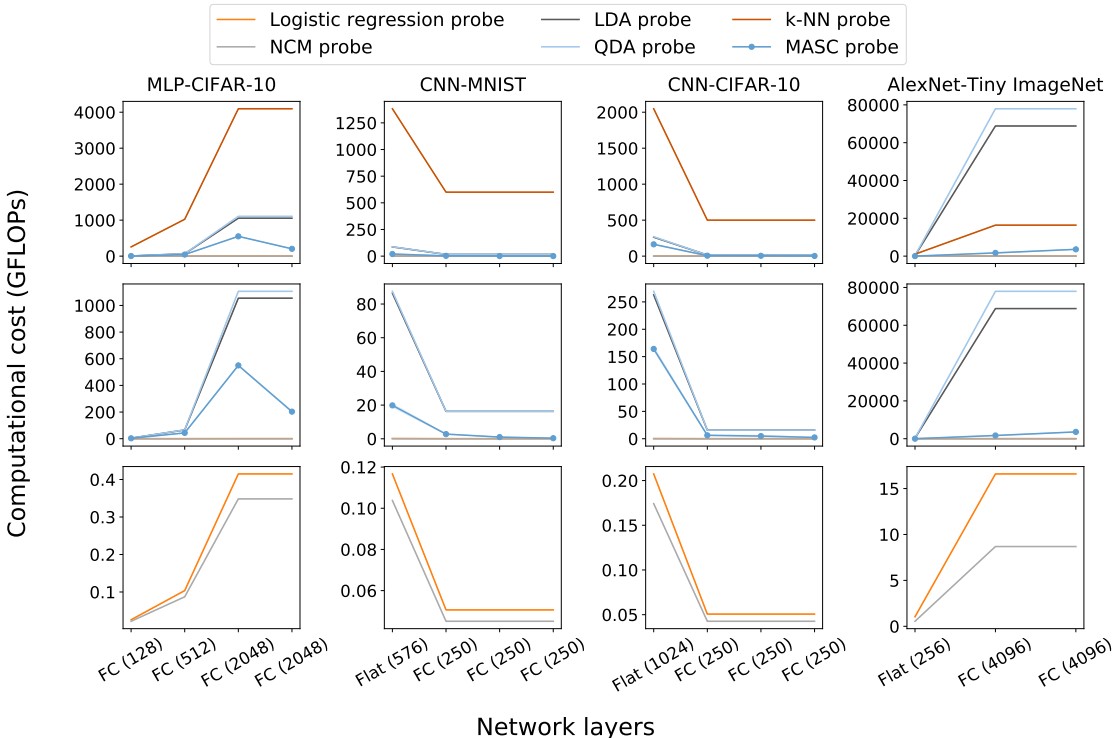

Figure 12: Computational costs (GFLOPS) for different probes over the layers of the network. Computational costs (GFLOPS) of MASC are shown for comparison. FC corresponds to fully connected layer with ReLU activation whereas Flat corresponds to flatten layer without ReLU activation.

computations are highly optimized and benefit more from hardware acceleration than the covariance and PCA steps involved in building subspaces.

We also find a clear scaling trend with respect to layer dimensionality. Layers with higher feature dimensionality – such as FC(2048), FC(4096), or Flat(1024/576) – consistently require higher computational cost during subspace construction. In contrast, lower-dimensional layers (e.g., FC(128) or FC(250)) remain comparatively inexpensive. This behavior aligns with the expected quadratic dependence of PCA based subspace construction on the feature dimension, and highlights that computational cost grows steadily as the layer dimensionality increases.

# F Experiments with AlexNet model trained on Tiny ImageNet

MASC test accuracy over the layers of AlexNet trained on Tiny ImageNet when the data is projected onto corrupted training subspaces with the indicated corruption degree is shown in Figure 20. Test accuracy of the model and best model test accuracy is shown for comparison. Best model test accuracy corresponds accuracy of the test data of the model if early stopping was used.

Even for AlexNet-Tiny ImageNet corrupted model (with non-zero corruption degree), except those with 100% corruption degree, we find that our Minimum Angle Subspace Classifier (MASC) in at least one layer has better test accuracy than the corresponding model itself.

MASC test accuracy over the layers of AlexNet trained on Tiny ImageNet when the data set is projected onto corrupted training and true training subspace is shown in Figure 21.

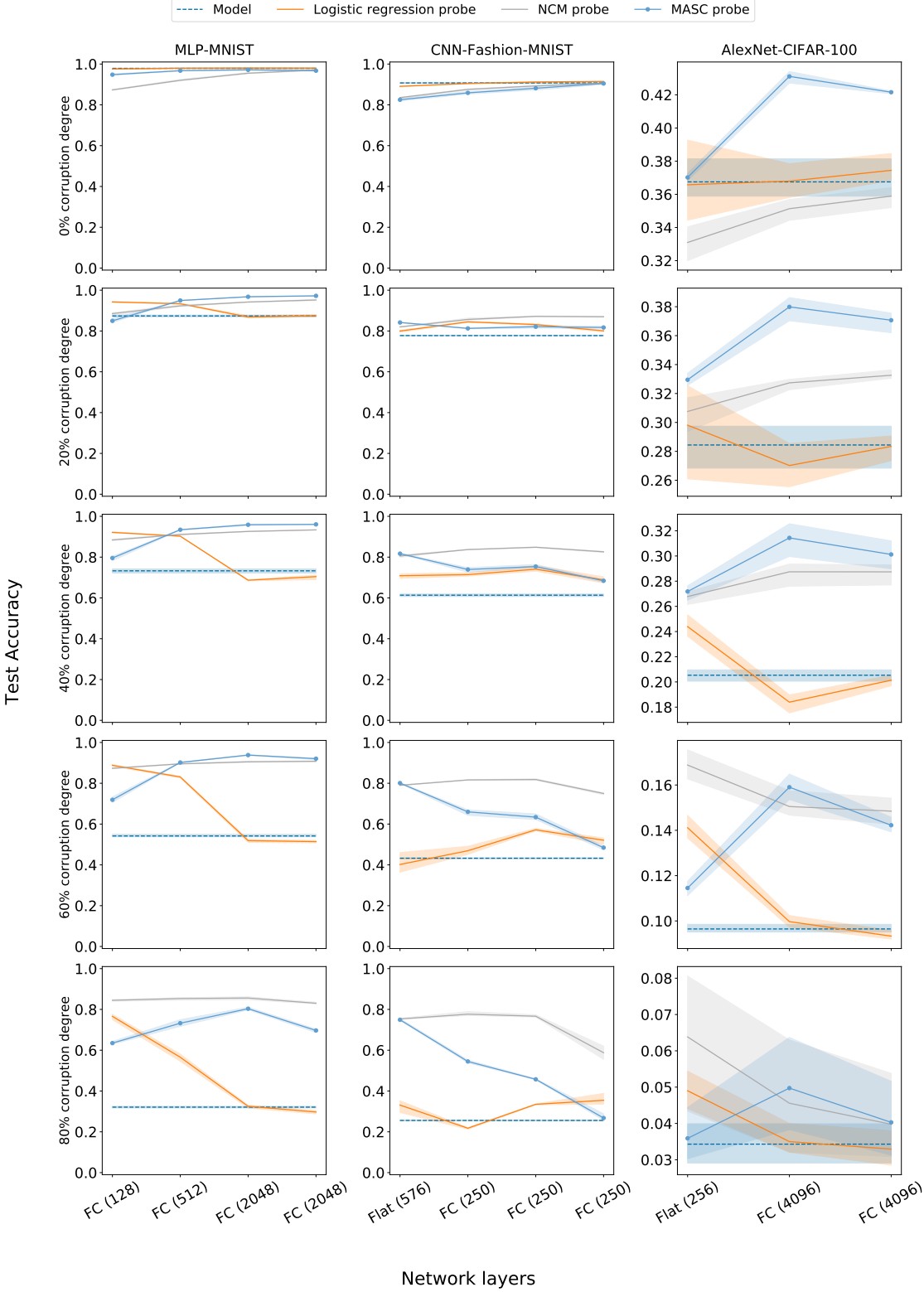

Figure 13: Test accuracies for different probes over the layers of the network. Rows corresponds to plots which have the same corruption degree and the columns correspond to the models as noted. Test accuracy of the model and MASC are shown for comparison. FC corresponds to fully connected layer with ReLU activation whereas Flat corresponds to flatten layer without ReLU activation.

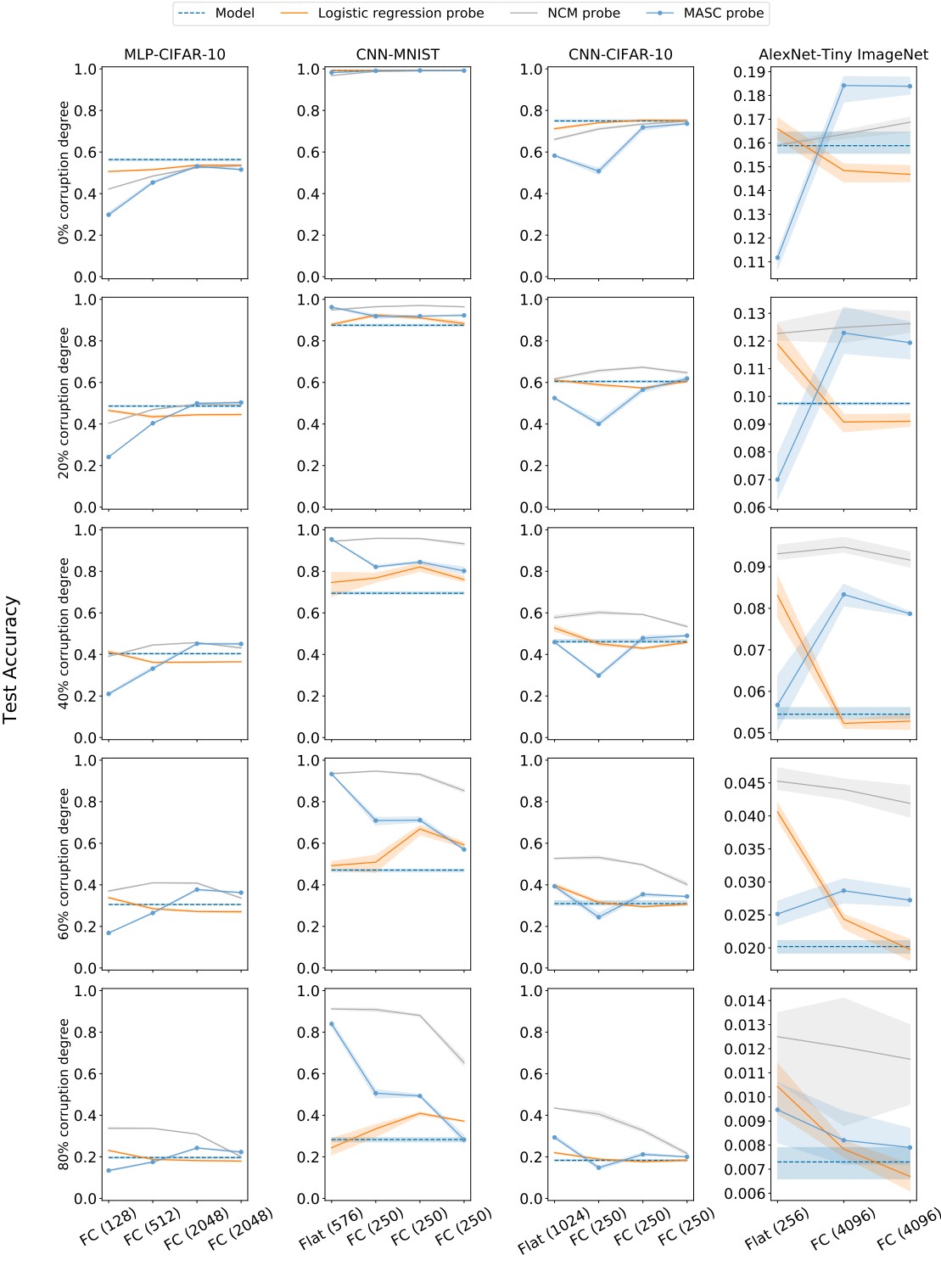

Figure 14: Test accuracies for different probes over the layers of the network. Rows corresponds to plots which have the same corruption degree and the columns correspond to the models as noted. Test accuracy of the model and MASC are shown for comparison. FC corresponds to fully connected layer with ReLU activation whereas Flat corresponds to flatten layer without ReLU activation.

# G   Random initializations of control models versus trained models

This section covers a set of control experiments to show MASC performance on random initialized model and contrast this with the trained models presented in the main text. We have verified that, for every such control model, the model training and test accuracies for the randomly initialized models is at chance level, for the corresponding dataset in question.

MASC accuracy on test for randomly initialized model and trained model when data is projected on corrupted training subspaces is shown in Figure 22 and 24. Trained model training and test accuracies are shown for reference.

We find that indeed accuracies of the MASC classifier on the random initialization outperforms the network, except for low corruption degrees (i.e. $<= 20\%$ corruption degree). However, in the experiments where subspaces are trained on corrupted training data from corrupted models, by-and-large, the MASC classifier usually, and on at least one layer outperforms the MASC classifier trained on the random initialization with exceptions being the 80% corruption degree models on MLP-MNIST, AlexNet-Tiny ImageNet and 100% corruption degree on CNN-Fashion-MNIST.

MASC accuracy on test for random initialized model and trained model when data is projected on subspaces corresponding to true training labels is shown in Figure 23 and 25. Notably, for the experiments where subspaces are constructed with true labels on corrupted models, the MASC classifier on these models outperforms the MASC classifier on random initializations usually and certainly in at least one layer on every model tested. These results are consistent with the main message of the paper, namely that even with memorized models, the layerwise representations of the models are organized in a manner that they develop significant ability to generalize over and above that bestowed by a random initialization, and in particular, they do not lose this ability, as one might have naively expected, due to label noise. If they were losing this ability, then the MASC classifier on the subspaces would end up performing significantly worse than the MASC classifier run on randomly initialized models.

Although it is interesting that random projection have good generalization to true labels, it is not surprising as this has been shown (Alain & Bengio, 2018) and also studied (Jarrett et al., 2009).

# H   Investigating latent memorization capabilities of uncorrupted models

Conversely, we ask how well a network trained on true labels can manifest memorization of an arbitrary relabeling of its training data. More specifically, we built a MASC classifier on a model trained on true training labels, with the goal of memorizing training data whose labels are corrupted to varying degrees *post hoc*.

To do this, we shuffle the labels of the training set to some corruption degree and construct the corresponding class-specific subspaces with respect to the layerwise outputs of the generalized models, i.e. models trained with uncorrupted training data. We then build a MASC classifier corresponding to these subspaces.

MASC accuracy on corrupted training data, original training data and MASC accuracy on test data over the layers of the networks are shown in Figure 26. We have results on additional models, namely MLP trained on CIFAR-10, CNN trained on MNIST and CIFAR-10, and AlexNet trained on Tiny ImageNet; see Figure 29. In Table 5, we show by what percentage the MASC classifier outperformed the model train accuracies on the corrupted labels for the best layer for corruption degrees 20%, 40%, 60% and 80%.

Interestingly, we find a dichotomy in model behavior here, with some models trained on specific datasets having the propensity to memorize to a high degree, whereas others not demonstrating such ability. Specifically, we observe that for uncorrupted models with low/modest model test accuracies (i.e. AlexNet-CIFAR-100 and AlexNet-Tiny ImageNet), the MASC classifiers described above have high accuracies on the corrupted training set (i.e. appear to be able to memorize with high accuracies). Conversely, in most uncorrupted models with high model test accuracies (i.e. MLP-MNIST and CNN-Fashion-MNIST), we find that these MASC classifiers have more modest accuracies on the training set with corrupted labels (i.e. do not memorize to a high degree). This suggests the hypothesis that high generalization to true labels inhibits the innate

Table 5: Percentage by which the MASC classifier (run on the best layer) outperformed the model's train accuracy on corrupted labels. The model is trained on images with true training labels. MASC classifier is operating on subspaces corresponding to corrupted training data. The best layer corresponds to the one that has the highest measured MASC train accuracy among the layers for the said model/dataset. The accuracies in each case are averaged over three runs. The values are rounded to the second decimal place.

| Corruption degree | 20% | 40% | 60% | 80% |
|---|---|---|---|---|
| MLP-MNIST | -2.74% | -2.35% | -1.81% | 3.11% |
| MLP-CIFAR-10 (Appendix) | 18.4% | 51.28% | 112.81% | 255.93% |
| CNN-MNIST (Appendix) | -0.24% | 0.40% | 3.50% | 26.76% |
| CNN-Fashion-MNIST | -5.33% | -1.01% | 7.79% | 50.05% |
| CNN-CIFAR-10 (Appendix) | 13.17% | 42.65% | 101.38% | 243.12% |
| AlexNet-CIFAR-100 | 27.91% | 70.03% | 152.78% | 394.89% |
| AlexNet-Tiny ImageNet (Appendix) | 23.19% | 63.18% | 144.52% | 389.85% |

ability of models to memorize and conversely that low/moderate generalization to true labels is accompanied by representations that offer greater propensity to memorize, at least as manifested by the MASC classifier. However, more work is needed to study this phenomenon to understand the trade-offs and the mechanisms underlying them, which has been beyond the scope of the present work. Also, surprisingly, we find that MASC classifiers often have test accuracies that approach or exceed uncorrupted model test accuracies, even though they correspond to corrupted subspaces (see e.g. AlexNet-CIFAR-100 and AlexNet-Tiny ImageNet).

## I   Experimental results on additional models (MLP-CIFAR-10, CNN-MNIST, CNN-CIFAR-10, AlexNet-Tiny ImageNet and ResNet-18-CIFAR-10)

In Table 6, we report the accuracy difference between the MASC classifier – when the data is projected onto corrupted training subspaces – and the model for the best such layer.

Table 6: Accuracy difference between the MASC classifier (run on the best layer) and the model's test accuracy when the data is projected onto corrupted training subspaces. The model's test accuracy are reported in Table 4. The best layer corresponds to the one that has the highest measured MASC test accuracy among the layers for the said model/dataset. The accuracies in each case are averaged over three runs. The values are rounded to the second decimal place. The values are rounded to the second decimal place.

| Corruption degree | 20% | 40% | 60% | 80% |
|---|---|---|---|---|
| MLP-MNIST | 9.55 | 23.18 | 40.65 | 51.33 |
| MLP-CIFAR-10 (Appendix) | 4.81 | 9.85 | 14.35 | 12.75 |
| CNN-MNIST (Appendix) | 8.59 | 25.72 | 46.49 | 56.91 |
| CNN-Fashion-MNIST | 5.82 | 20.55 | 36.74 | 48.33 |
| CNN-CIFAR-10 (Appendix) | 1.39 | 2.89 | 8.37 | 11.02 |
| AlexNet-CIFAR-100 | 9.55 | 10.90 | 6.26 | 1.54 |
| AlexNet-Tiny ImageNet (Appendix) | 2.68 | 2.91 | 0.92 | 0.10 |
| ResNet-18-CIFAR-10 | -0.28 | 10.48 | 23.20 | 23.86 |

In Table 7, we show by what percentage the MASC classifier (subspace corresponding to true labels) outperformed the model for the best layer for corruption degrees 20%, 40%, 60% and 80%. In fact, the MASC test accuracies for the corrupted models (with non-zero corruption degree) are sometimes fairly close to the test accuracy of the uncorrupted model. In Table 8, we also report the accuracy difference between the MASC

classifier – when the data is projected onto training subspaces corresponding to true labels – and the model for the best such layer.

Table 7: Percentage by which the MASC classifier (run on the best layer) outperformed the model's test accuracy when the data is projected onto training subspaces corresponding to true labels. The best layer corresponds to the one that has the highest measured MASC test accuracy among the layers for the said model/dataset. The accuracies in each case are averaged over three runs. The values are rounded to the second decimal place.

| Corruption degree | 20% | 40% | 60% | 80% |
|---|---|---|---|---|
| MLP-MNIST | 10.96% | 32.01% | 77.77% | 198.43% |
| MLP-CIFAR-10 (Appendix) | 11.00% | 30.27% | 66.92% | 146.09% |
| CNN-MNIST (Appendix) | 10.69% | 39.50% | 104.97% | 239.44% |
| CNN-Fashion-MNIST | 8.37% | 33.09% | 88.97% | 212.42% |
| CNN-CIFAR-10 (Appendix) | 3.29% | 13.43% | 58.23% | 138.65% |
| AlexNet-CIFAR-100 | 38.78% | 68.91% | 133.76% | 337.51% |
| AlexNet-Tiny ImageNet (Appendix) | 33.31% | 83.85% | 157.10% | 212.33% |
| ResNet-18-CIFAR-10 | 2.92% | 32.23% | 93.55% | 228.64% |

Table 8: Accuracy difference between the MASC classifier (run on the best layer) and the model's test accuracy when the data is projected onto training subspaces corresponding to true labels. The model's test accuracy are reported in Table 4. The best layer corresponds to the one that has the highest measured MASC test accuracy among the layers for the said model/dataset. The accuracies in each case are averaged over three runs. The values are rounded to the second decimal place.

| Corruption degree | 20% | 40% | 60% | 80% |
|---|---|---|---|---|
| MLP-MNIST | 9.57 | 23.46 | 42.13 | 63.69 |
| MLP-CIFAR-10 (Appendix) | 5.35 | 12.21 | 20.44 | 28.76 |
| CNN-MNIST (Appendix) | 9.36 | 27.43 | 49.45 | 67.77 |
| CNN-Fashion-MNIST | 6.51 | 20.30 | 38.49 | 54.32 |
| CNN-CIFAR-10 (Appendix) | 1.99 | 6.20 | 18.03 | 25.40 |
| AlexNet-CIFAR-100 | 11.03 | 14.15 | 12.90 | 11.58 |
| AlexNet-Tiny ImageNet (Appendix) | 3.25 | 4.57 | 3.17 | 1.55 |
| ResNet-18-CIFAR-10 | 1.97 | 16.61 | 32.26 | 45.77 |

All the experimental results on additional models i.e, MLP-CIFAR-10, CNN-MNIST, CNN-CIFAR-10 and AlexNet-Tiny ImageNet are shown in this section.

MASC accuracy over the layers of the MLP trained on CIFAR-10, CNN trained on MNIST, CNN trained on CIFAR-10, and AlexNet trained on Tiny ImageNet when the data is projected onto corrupted training subspaces is shown in Figure 27. MASC accuracy over the layers of the MLP trained on CIFAR-10, CNN trained on MNIST, CNN trained on CIFAR-10, and AlexNet trained on Tiny ImageNet when the data set is projected subspace corresponding to true training labels is shown in Figure 28. MASC accuracy over the layers of the generalized MLP network trained on CIFAR-10, CNN trained on MNIST, CNN trained on CIFAR-10 and AlexNet network trained on Tiny ImageNet when the data is projected onto corrupted training subspaces is shown in Figure 29.

## J  Number of PCA components

This section covers the number of class-wise PCA components used in all the experiments.

Number of class-wise PCA components of corrupted training subspace over the layer of MLP-MNIST,CNN-Fashion-MNIST, and AlexNet-CIFAR-100 is shown in Figure 30 and MLP-CIFAR-10,CNN-MNIST, CNN-CIFAR-10 and AlexNet-Tiny ImageNet is shown in Figure 31. For generalized models, it is observed that for 99% variance captured, the number of PCA components is significantly smaller in comparison to the ambient dimensionality of the layer (number of units in that layer). Over corruption, it is observed that for MLP-MNIST, MLP-CIFAR-10, and CNN-Fashion-MNIST, the number of class-wise PCA components increase. And the variance between the number of dimensions decrease. For AlexNet-CIFAR-100 and AlexNet-Tiny ImageNet, it is the opposite case, wherein the number of PCA components over corruption decreases.

Number of class-wise PCA components of original training subspaces over the layer of networks is shown in Figure 32 and Figure 33. We find that for original training subspaces, although the dimensionality has increased with corruption degree, the variance has remained the approximately similar.

Number of class-wise PCA components of original training subspaces over the layer of networks of the generalized model is shown in Figure 34 and Figure 35 .

## K   SGD vs Adam

We have also trained MLP networks with *Adam* optimizer on MNIST and CIFAR-10 with various degrees of corruption. The results for MASC accuracy using corrupted subspaces is shown in Figure 36 and its respective average number of PCA components is shown in Figure 37 . With both optimizer choices, even with high corruption degrees, we find that the MASC have better accuracy than the model on the test data. MASC on corrupted training accuracy in most cases reaches the models training accuracy at the latter layers of the network with an exception of MLP trained on MNIST. In most cases, at the initial FC (128) layer of the network, there is a drop in accuracy observed in comparison to the corresponding value for the input and then an increase in latter layers of the network; the MLP trained on CIFAR-10 with SGD is an exception, however.

For 40% corruption degree, although approximately 36% of labels are flipped, with MLP trained on MNIST, model trained on *Adam* has MASC test accuracy of around 95%, and around 50% for CIFAR-10. This is better than the MASC accuracy at the input layer and models test accuracy. For 60% corruption degree with MLP trained on MNIST, model trained on *Adam* has MASC test accuracy of around 90% whereas for model trained on *SGD* is around 85%. Although the networks are trained with 56% of label corruption, yet in FC (512) layer, the MASC training original accuracy is about 70% for model trained on *Adam* whereas it is 85% for model trained on *SGD*. MASC on original training data does unfavorably on model trained with *SGD* rather than with *Adam*, although further investigation is required.

The results for MASC accuracy using original subspaces are shown in Figure 38 and its respective average number of PCA components are shown in Figure 39. MLP models trained with *Adam* have qualitatively similar results. The results for MASC accuracy using corrupted subspaces of generalized models are shown in Figure 40 and its respective average number of PCA components are shown in Figure 41. MLP models trained with Adam have qualitatively similar results.

## L   Retraining experiment results on AlexNet models and additional analysis results.

Retraining the model using MASC experiment results on AlexNet models i.e, AlexNet-CIFAR-100 and AlexNet-Tiny ImageNet for different corruption degrees are shown in this section. Test accuracies averaged over three runs on the 80% test dataset is plotted for different AlexNet-dataset pairs for various corruption degrees and for various models/MASC classifiers is shown in Figure 42.

Decrease in fraction of incorrectly labeled data points resulting from relabeling the training set expressed as a fraction of size incorrectly labeled points in the existing corrupted data, using the corresponding best-layer MASC classifier (corrupted subspace and true subspace) is shown in Figure 43 for all models-dataset pairs and corruption degree 20%, 40%, 60%, & 80%. The average values for reference are available in Table 9. The percentage of incorrect labels that are correctly relabeled using MASC (corrupted subspace and true subspace) for all models-dataset and corruption degrees 20%, 40%, 60%, 80% is shown in Figure 44. The

average values for reference are available in Table 10. The percentage of incorrect labels that are correctly relabeled using MASC (corrupted subspace and true subspace) for all models-dataset and corruption degrees 20%, 40%, 60%, 80% is shown in Figure 45. The average values for reference are available in Table 11.

Table 9: The average decrease in fraction of incorrectly labeled data points resulting from relabeling the training set expressed as a fraction of size incorrectly labeled points in the existing corrupted data, using the corresponding best-layer MASC classifier. The values are rounded to the second decimal place.

| Subspace | Corrupted labels | | | | True labels | | | |
|---|---|---|---|---|---|---|---|---|
| Corruption degree | 20% | 40% | 60% | 80% | 20% | 40% | 60% | 80% |
| MLP-MNIST | 76.93% | 38.63% | 19.90% | 7.75% | 97.83% | 99.66% | 99.93% | 99.95% |
| MLP-CIFAR-10 | 3.45% | 2.38% | 1.10% | 0.22% | 95.49% | 97.91% | 98.75% | 99.14% |
| CNN-MNIST | 88.15% | 89.19% | 86.44% | 71.12% | 92.42% | 96.46% | 96.93% | 97.12% |
| CNN-Fashion-MNIST | 40.69% | 64.96% | 69.38% | 62.61% | 35.96% | 66.08% | 79.26% | 80.59% |
| CNN-CIFAR-10 | 6.25% | 3.30% | 9.18% | 3.79% | 16.44% | 40.22% | 81.75% | 80.09% |
| AlexNet-CIFAR-100 | -0.09% | 0.26% | -0.20% | -0.32% | 99.61% | 98.96% | 90.20% | 95.13% |
| AlexNet-Tiny ImageNet | -92.46% | -26.50% | -3.95% | -11.55% | -91.32% | -20.24% | 2.89% | -12.38% |

Table 10: The average percentage of incorrect labels that are correctly relabeled using MASC (corrupted subspace and true subspace) for all models and for corruption degrees 20%, 40%, 60%, 80%. The values are rounded to the second decimal place.

| Subspace | Corrupted labels | | | | True labels | | | |
|---|---|---|---|---|---|---|---|---|
| Corruption degree | 20% | 40% | 60% | 80% | 20% | 40% | 60% | 80% |
| MLP-MNIST | 77.62% | 38.65% | 19.90% | 7.75% | 99.23% | 99.80% | 99.95% | 99.96% |
| MLP-CIFAR-10 | 5.59% | 2.82% | 1.20% | 0.24% | 99.08% | 99.12% | 99.23% | 99.37% |
| CNN-MNIST | 93.71% | 91.34% | 87.64% | 72.08% | 98.27% | 98.54% | 98.26% | 97.88% |
| CNN-Fashion-MNIST | 81.42% | 79.18% | 76.16% | 66.23% | 83.82% | 86.80% | 88.76% | 86.08% |
| CNN-CIFAR-10 | 17.70% | 11.54% | 12.30% | 5.51% | 41.98% | 55.63% | 89.85% | 85.41% |
| AlexNet-CIFAR-100 | 0.15% | 0.43% | 0.19% | 0.06% | 99.84% | 99.32% | 91.06% | 95.34% |
| AlexNet-Tiny ImageNet | 5.94% | 3.96% | 0.51% | 0.41% | 25.01% | 24.43% | 18.77% | 3.26% |

Table 11: The average percentage of correct labels that are incorrectly relabeled using MASC (corrupted subspace and true subspace) for all models and for corruption degrees 20%, 40%, 60%, 80%. The values are rounded to the second decimal place.

| Subspace | Corrupted labels | | | | True labels | | | |
|---|---|---|---|---|---|---|---|---|
| Corruption degree | 20% | 40% | 60% | 80% | 20% | 40% | 60% | 80% |
| MLP-MNIST | 0.15% | 0.01% | 0% | 0% | 0.31% | 0.08% | 0.03% | 0.03% |
| MLP-CIFAR-10 | 0.47% | 0.25% | 0.13% | 0.06% | 0.79% | 0.69% | 0.58% | 0.59% |
| CNN-MNIST | 1.22% | 1.20% | 1.40% | 2.49% | 1.28% | 1.17% | 1.55% | 1.97% |
| CNN-Fashion-MNIST | 8.86% | 8.06% | 8.04% | 9.30% | 10.42% | 11.76% | 11.26% | 14.12% |
| CNN-CIFAR-10 | 2.52% | 4.67% | 3.69% | 4.48% | 5.62% | 8.73% | 9.60% | 13.82% |
| AlexNet-CIFAR-100 | 0.06% | 0.11% | 0.57% | 1.44% | 0.06% | 0.23% | 1.26% | 0.81% |
| AlexNet-Tiny ImageNet | 24.46% | 20.10% | 6.60% | 46.97% | 28.92% | 29.48% | 23.51% | 61.39% |

## M    Retraining experiment: epoch-wise results

To study the dynamics of accuracy during retraining, unencumbered by the early stopping criterion, we also performed a similar experiment without using early stopping, for 10 epochs. The results for model before training, model after retraining on MASC corrupted subspace, and model after retraining on MASC original subspace for all model-dataset pair and for various corruption degrees are shown in 46. The following steps are performed to retrain the model for corrupted and original subspaces independently.

1. Test dataset is split into 80%-20%.

2. The model's best layer is identified using MASC accuracy on the validation set (carved out from 20% of the test dataset).

3. The corrupted training dataset was relabeled using the best layer MASC predictions.

4. The existing corrupted model was loaded for retraining purpose. Using the relabeled corrupted training dataset, the model was further trained for 10 epochs.

5. The remaining 80% test dataset was used to calculate generalization to true labels.

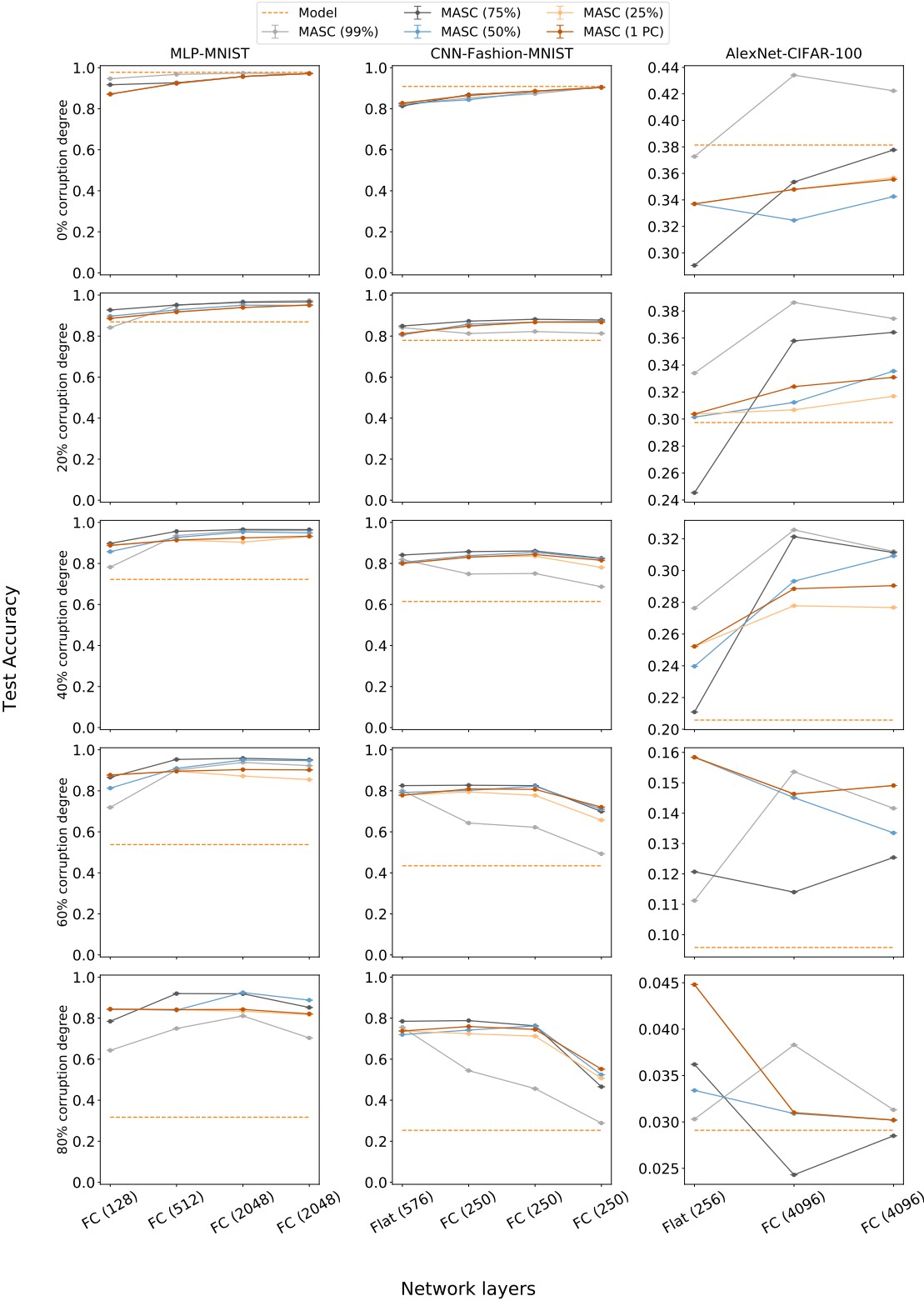

Figure 15: Minimum Angle Subspace Classifier (MASC) test accuracy over the layers of the network when the data is projected onto corrupted training subspaces with the indicated corruption degree, for multiple models/datasets. Rows corresponds to plots with the same corruption degree & the columns correspond to the models, as noted. Test accuracy (dotted line) of the model is shown. FC corresponds to fully connected layer with *ReLU* activation whereas Flat corresponds to flatten layer without *ReLU* activation.

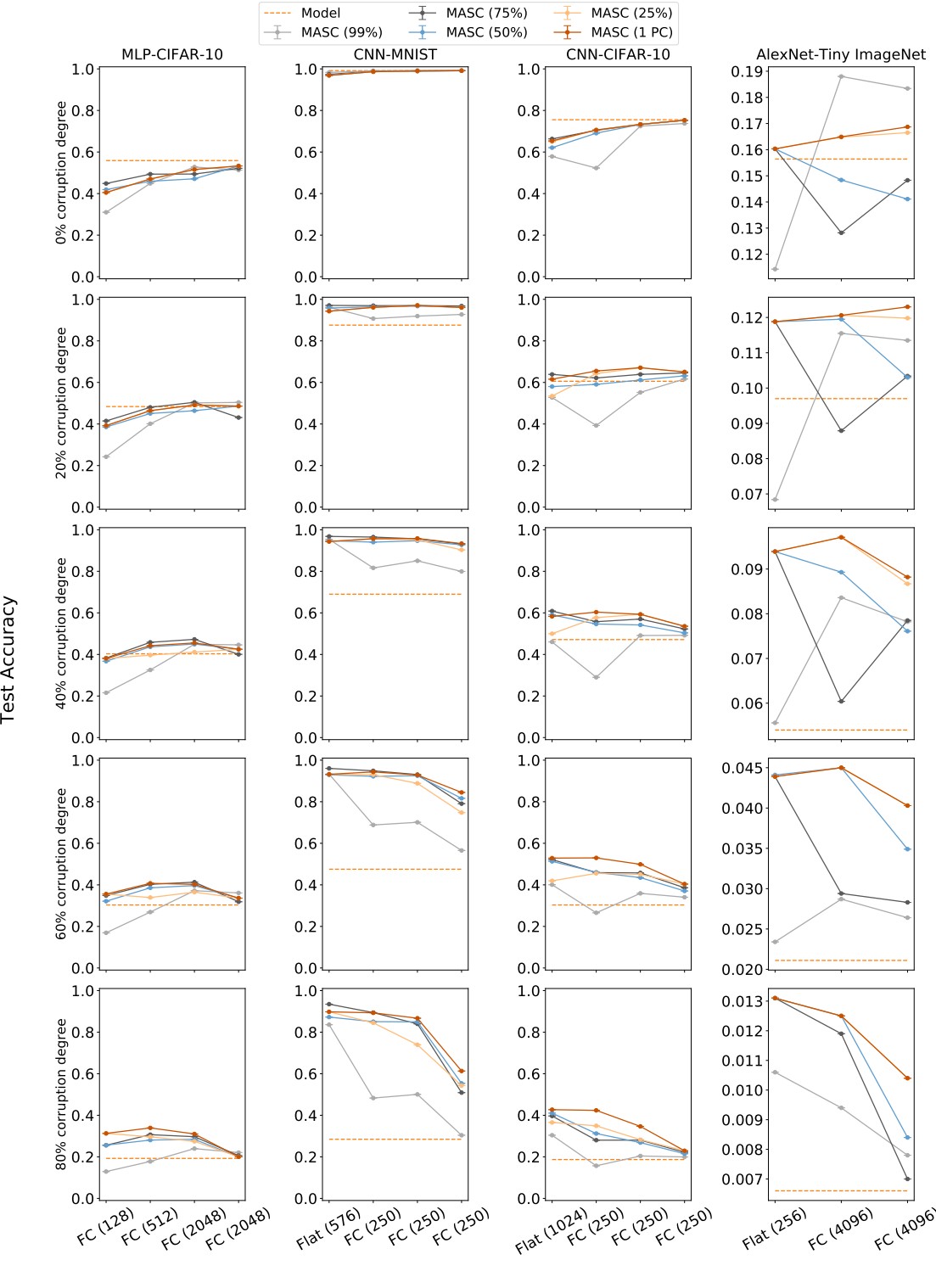

Figure 16: Minimum Angle Subspace Classifier (MASC) test accuracy over the layers of the network when the data is projected onto corrupted training subspaces with the indicated corruption degree, for multiple models/datasets. Rows corresponds to plots with the same corruption degree & the columns correspond to the models, as noted. Test accuracy (dotted line) of the model is shown. FC corresponds to fully connected layer with *ReLU* activation whereas Flat corresponds to flatten layer without *ReLU* activation.

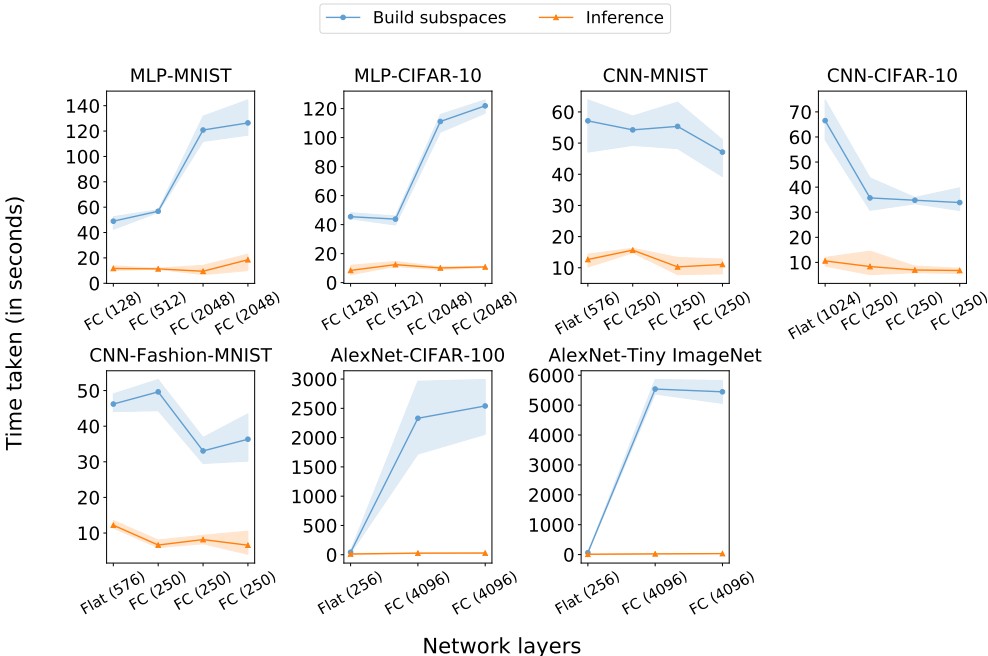

Figure 17: Total time taken in seconds by MASC for building the subspaces and inference over the layers of the network, for multiple models/datasets. FC corresponds to fully connected layer with *ReLU* activation whereas Flat corresponds to flatten layer without *ReLU* activation.

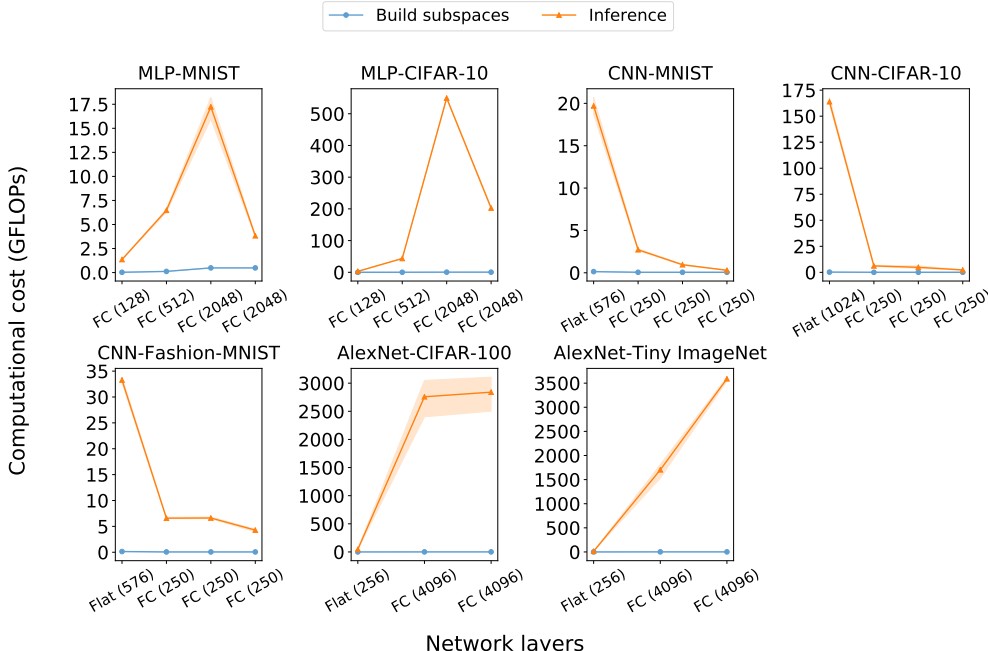

Figure 18: Total computational cost (GFLOPS) by MASC for building the subspaces and inference over the layers of the network, for multiple models/datasets. FC corresponds to fully connected layer with *ReLU* activation whereas Flat corresponds to flatten layer without *ReLU* activation.

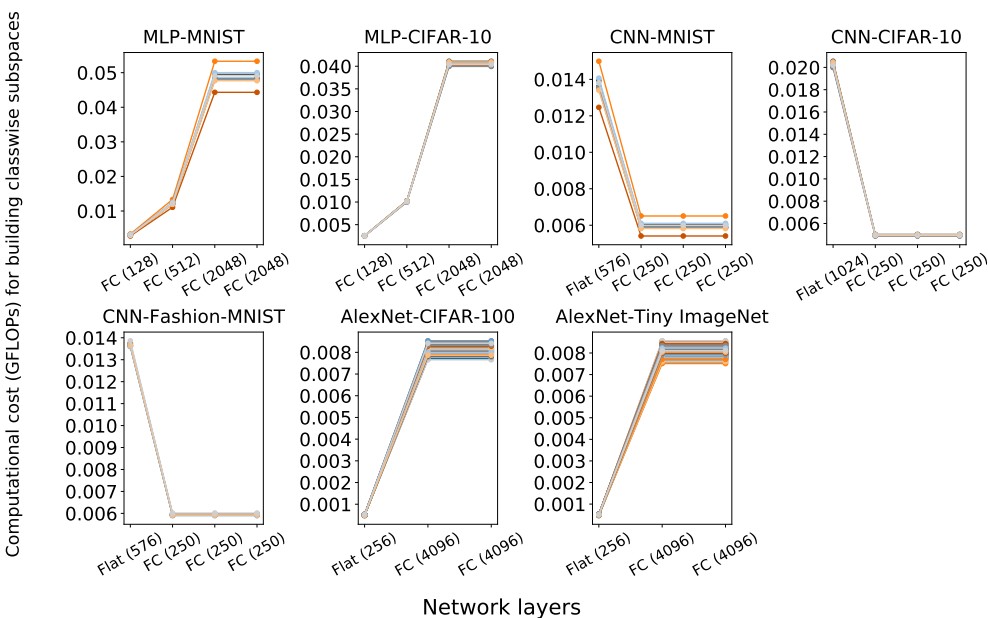

Figure 19: Computational cost (GFLOPS) by MASC for building per-class subspace over the layers of the network, for multiple models/datasets. FC corresponds to fully connected layer with *ReLU* activation whereas Flat corresponds to flatten layer without *ReLU* activation.

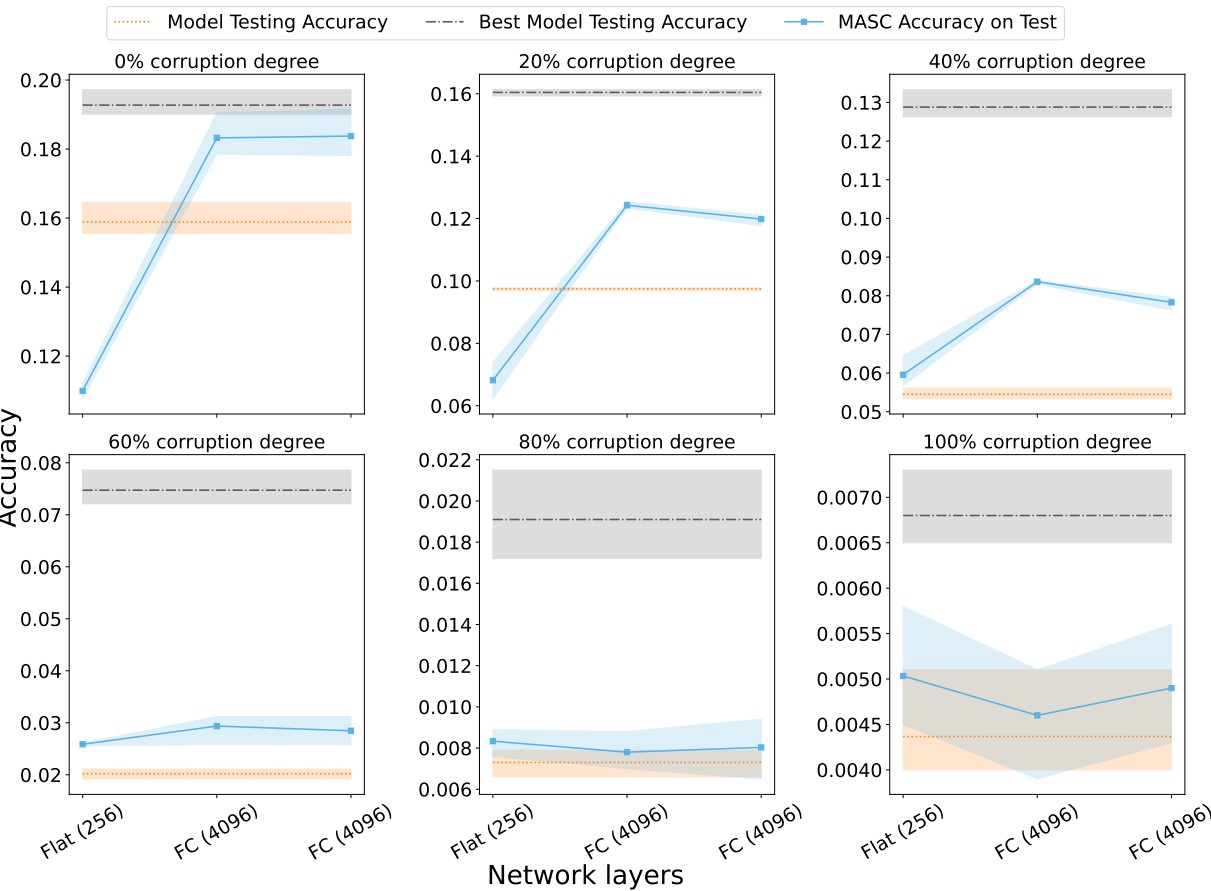

Figure 20: MASC test accuracy over the layers of AlexNet trained on Tiny ImageNet when the data is projected onto corrupted training subspaces with the indicated corruption degree. Test accuracy of the model and best model test accuracy is shown for comparison. Best model test accuracy corresponds accuracy of the test data of the model if early stopping was used.

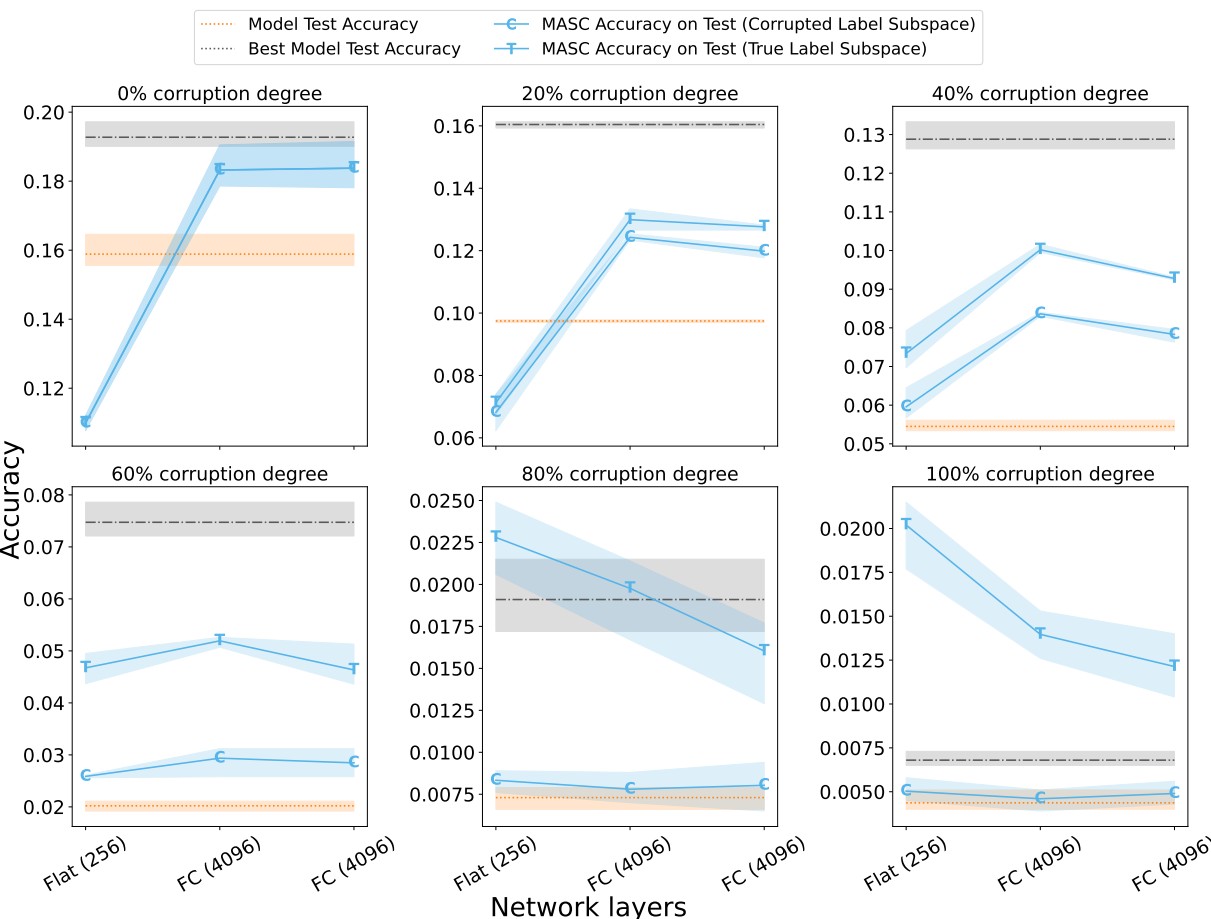

Figure 21: MASC test accuracy over the layers of AlexNet trained on Tiny ImageNet when the data set is projected onto corrupted training and true training subspace. Test accuracy of the model and best model test accuracy is shown for comparison. Best model test accuracy corresponds accuracy of the test data of the model if early stopping was used.

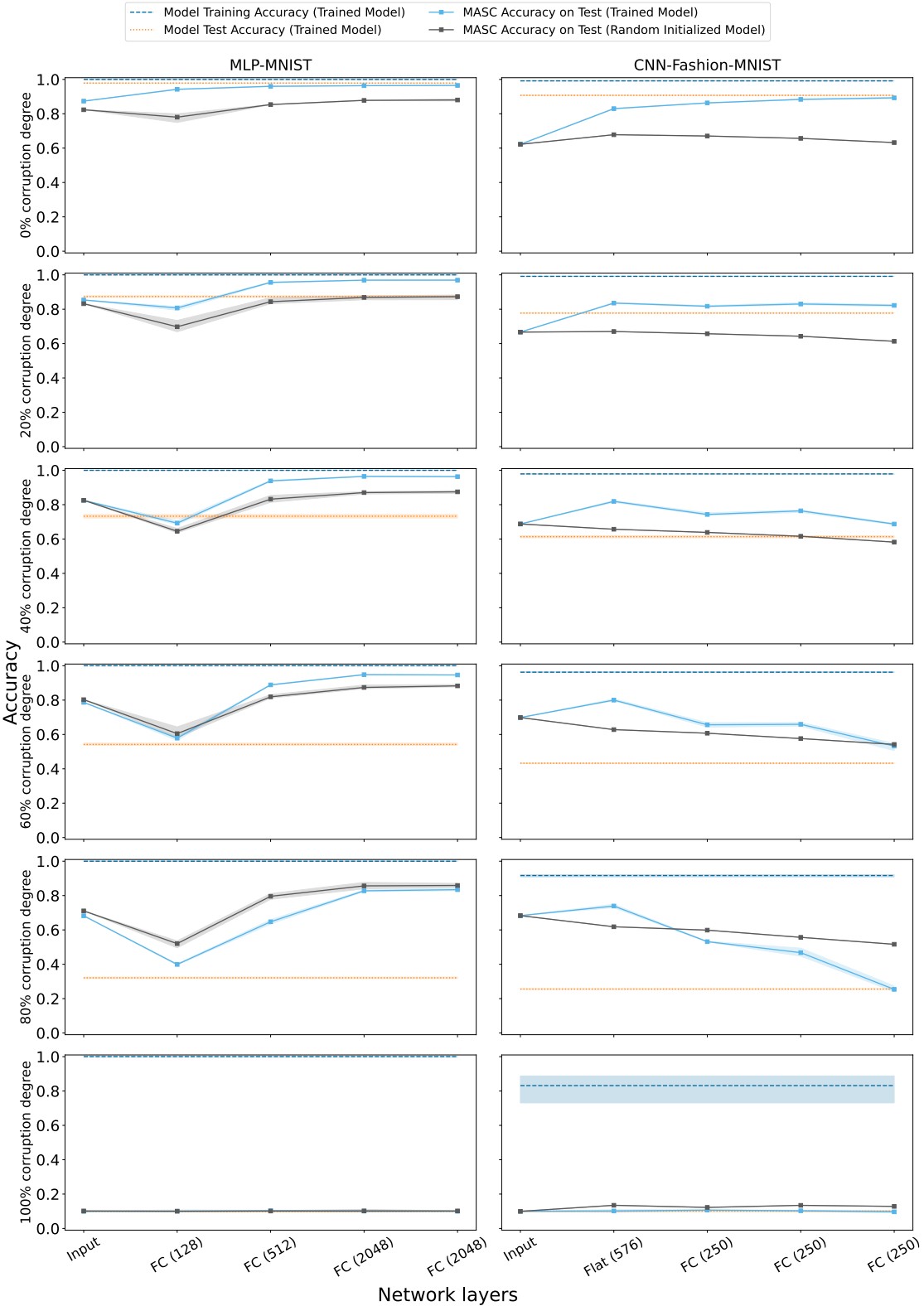

Figure 22: MASC accuracy over the layers of trained and random initialized network when the data is projected onto corrupted training subspaces with the indicated corruption degree.

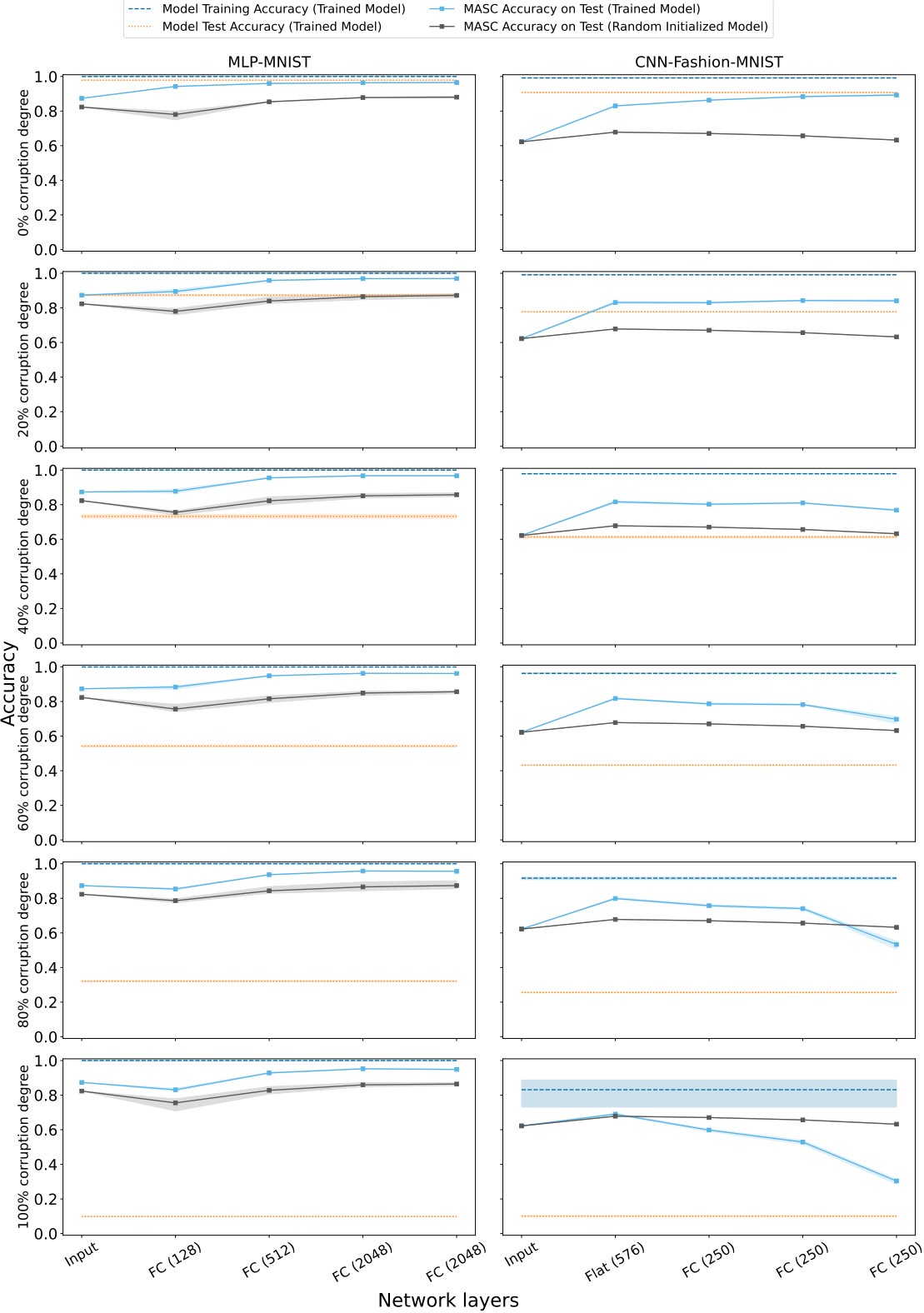

Figure 23: MASC accuracy over the layers of trained and random initialized network when the data set is projected onto subspace corresponding to true training labels. Test accuracy of the trained model is shown for comparison.

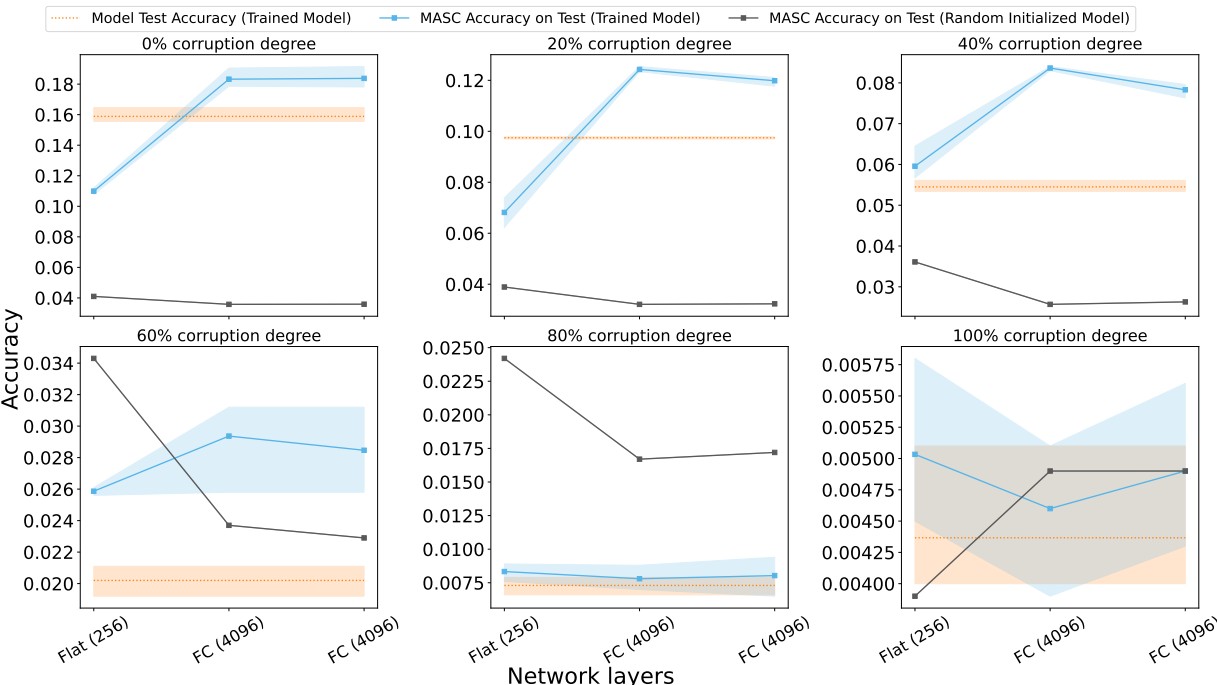

Figure 24: MASC accuracy over the layers of trained and random initialized AlexNet-Tiny ImageNet when the data is projected onto corrupted training subspaces with the indicated corruption degree.

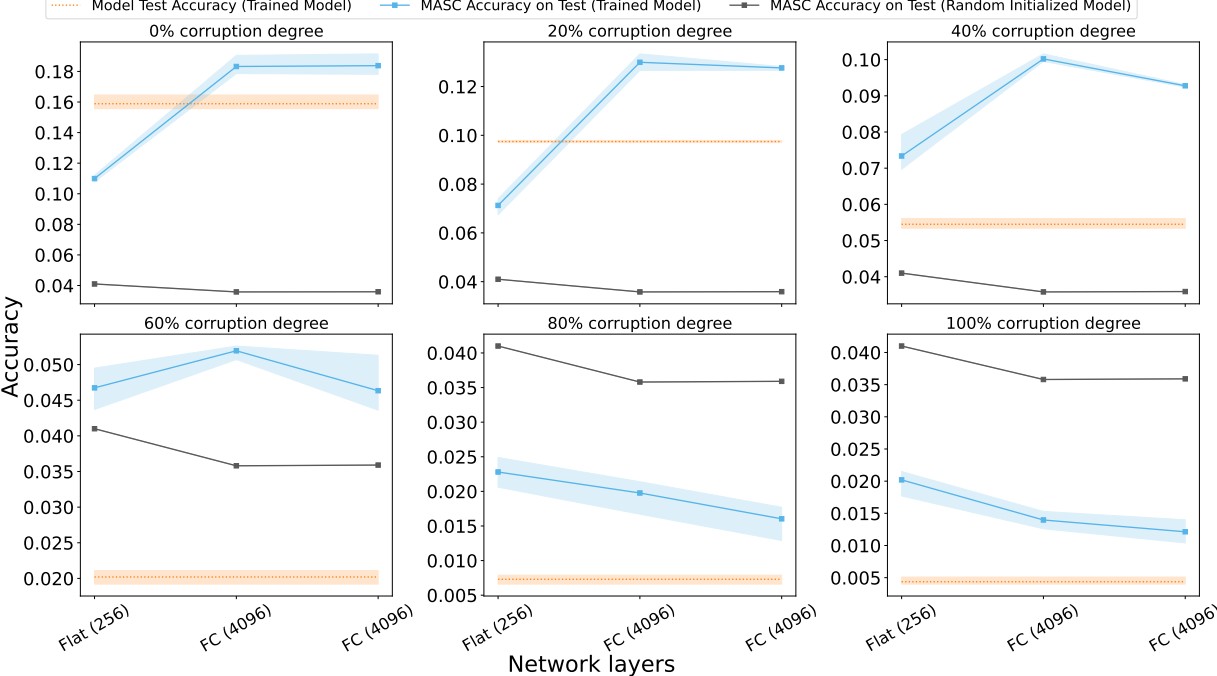

Figure 25: MASC accuracy over the layers of trained and random initialized AlexNet-Tiny ImageNet when the data set is projected onto subspace corresponding to true training labels.

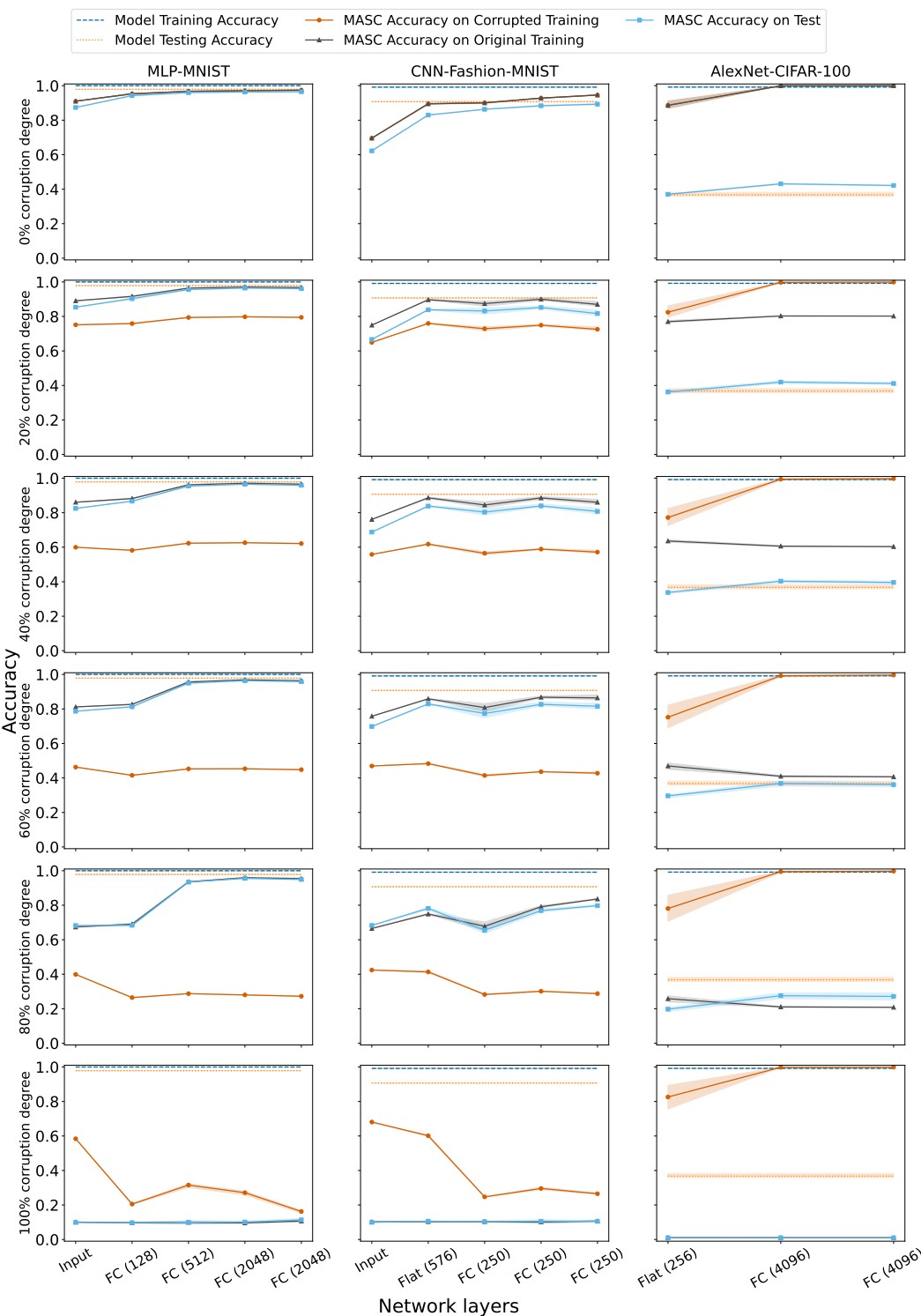

Figure 26: Minimum Angle Subspace Classifier (MASC) accuracy over the layers of the generalized network when the data set is projected onto corrupted training subspaces with the indicated corruption degree. Rows corresponds to plots which have the same corruption degree & the columns correspond to the generalized models as noted. Training & test accuracy of the generalized model is shown. FC corresponds to fully connected layer with *ReLU* activation whereas Flat corresponds to flatten layer without *ReLU* activation. The respective number of class-wise PCA components of the models is shown in Figure 34. *SGD* optimizer was used for training MLP models, whereas *Adam* optimizer was used for other models.

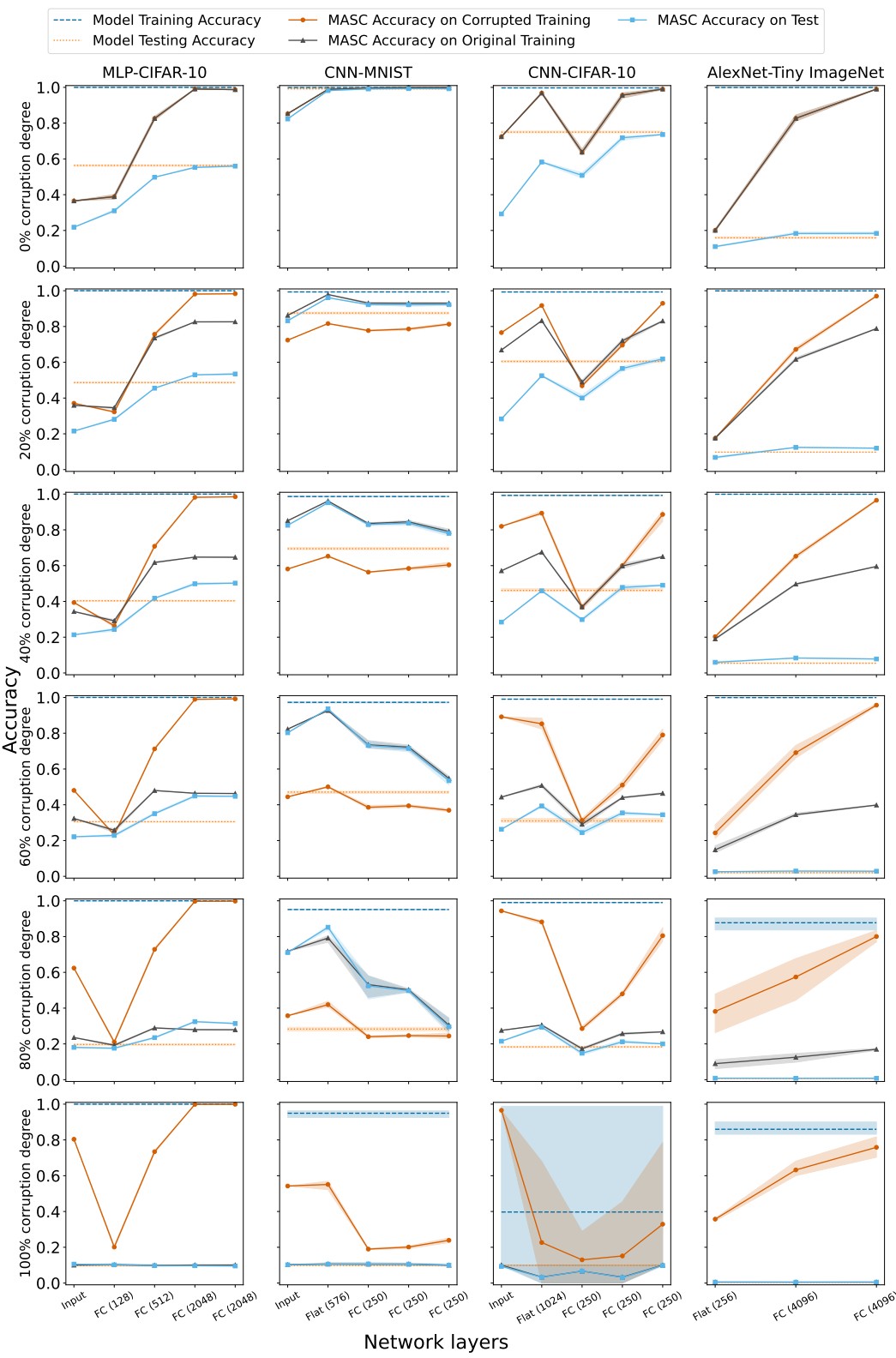

Figure 27: MASC accuracy over the layers of the network when the data is projected onto corrupted training subspaces with the indicated corruption degree. The number of class-wise PCA components of these models are shown in Figure 31.

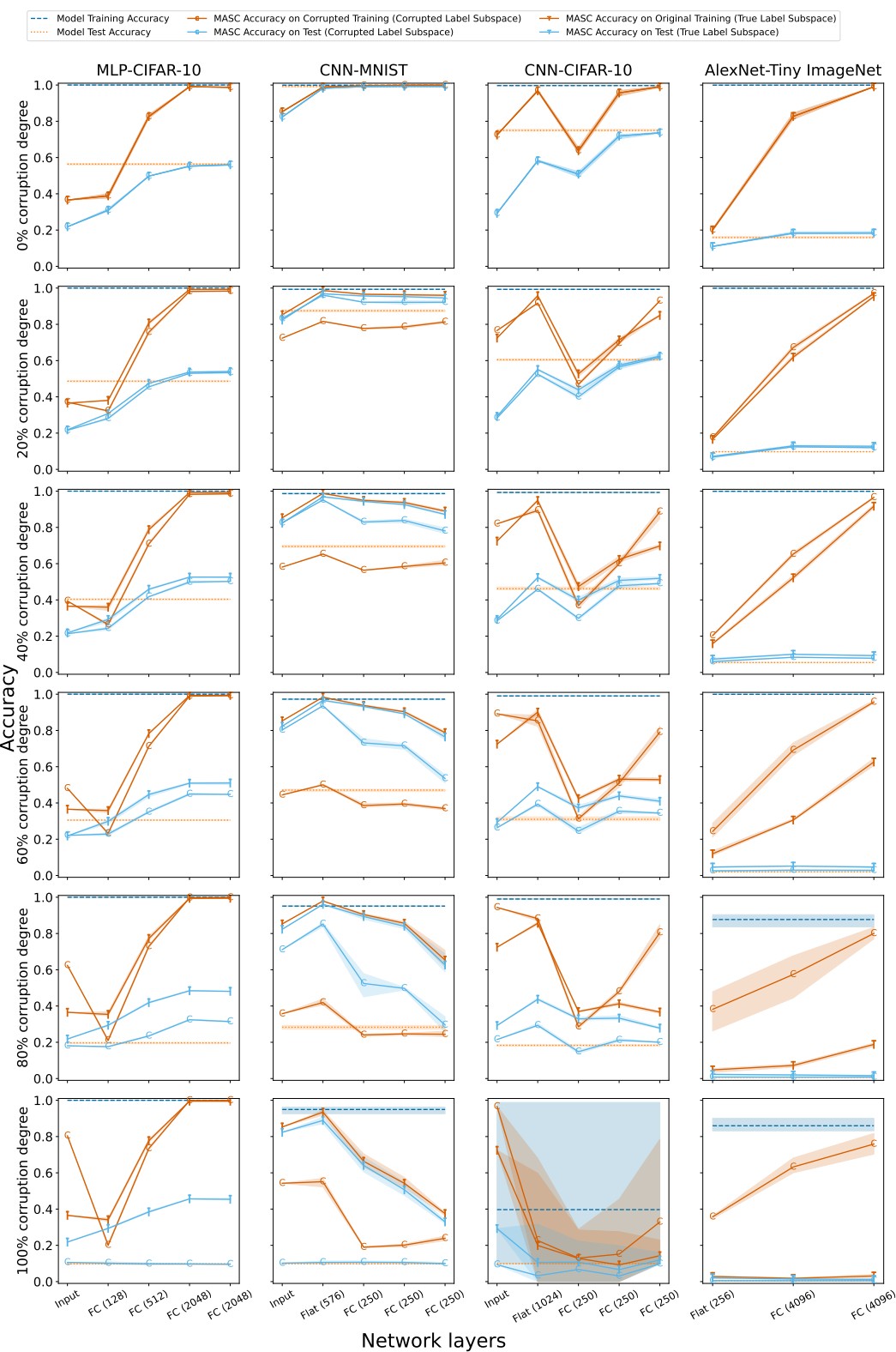

Figure 28: MASC accuracy over the layers of the network when the data set is projected onto corrupted subspace and subspace corresponding to true training labels. The respective number of class-wise PCA components for true training label subspaces of the models is shown in Figure 33.

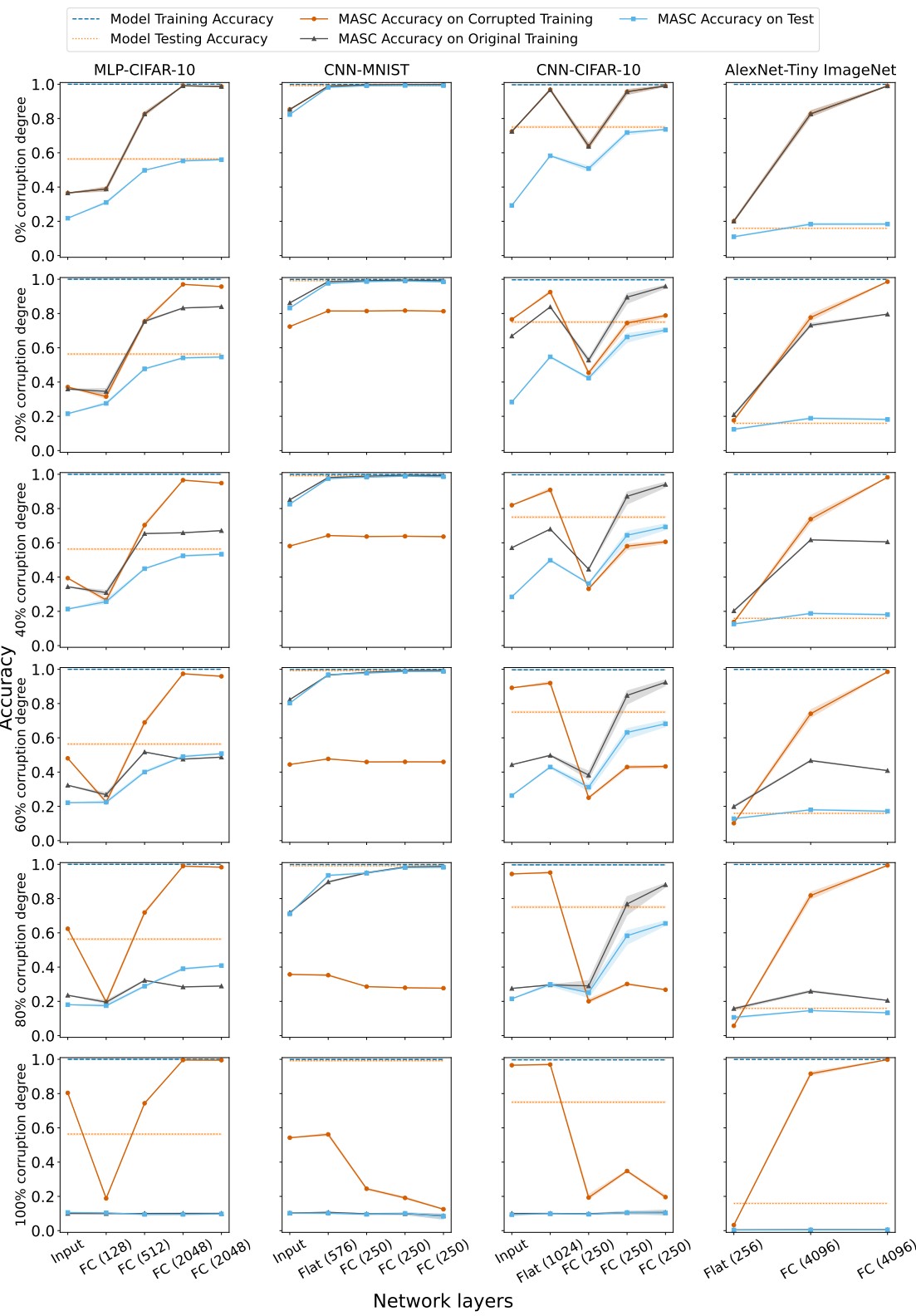

Figure 29: MASC accuracy over the layers of the generalized network when the data set is projected onto corrupted training subspaces with the indicated corruption degree. The respective number of class-wise PCA components of the models is shown in Figure 35.

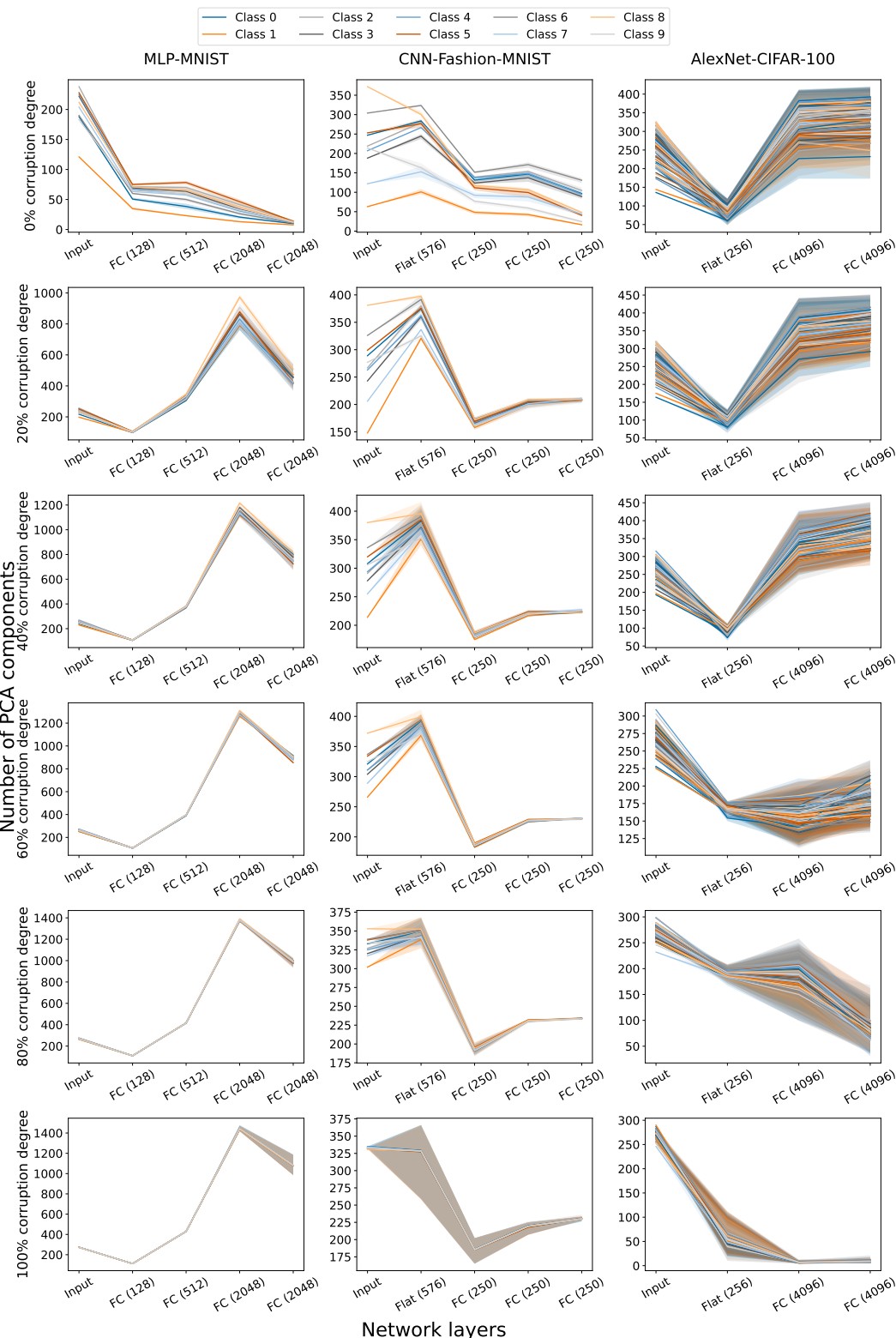

Figure 30: Class-wise number of PCA components of the corrupted training subspace over the layers of multiple networks with various corruptions degrees. Although it is not mentioned in the legend, all the 100 classes of CIFAR-100 are plotted.

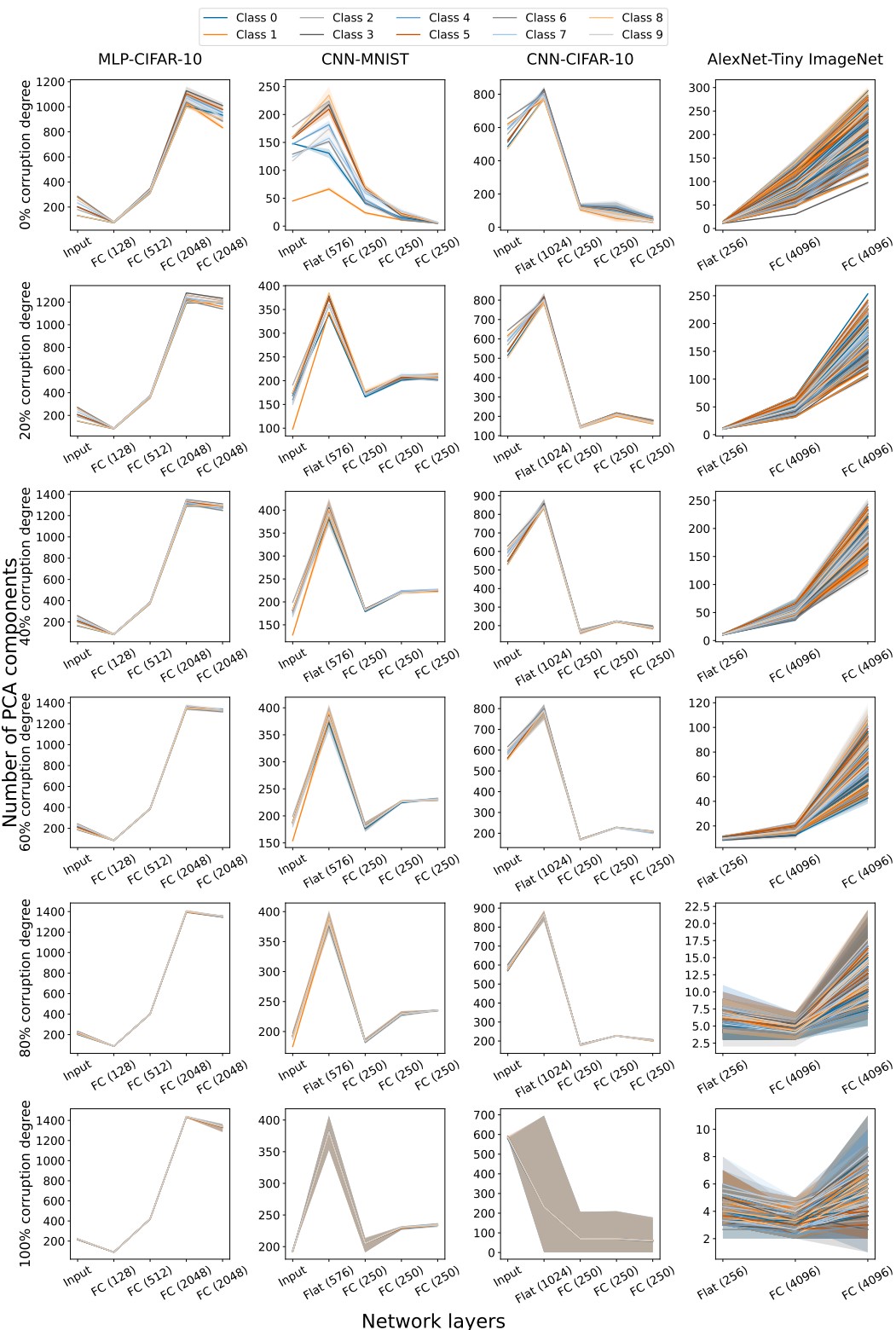

Figure 31: Class-wise number of PCA components of the corrupted training subspace over the layers of multiple networks with various corruptions degrees. Although it is not mentioned in the legend, all the 200 classes of Tiny ImageNet are plotted.

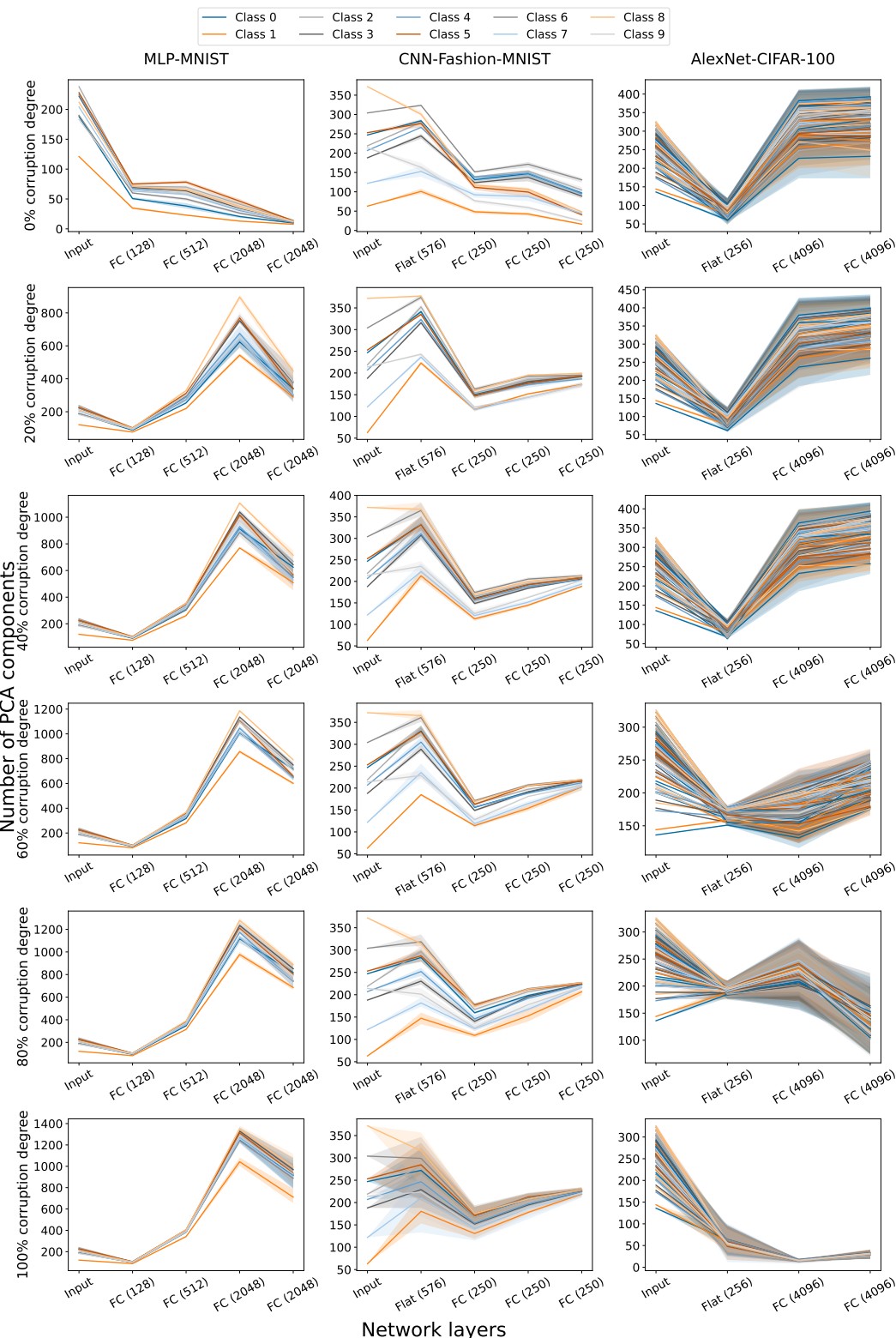

Figure 32: Class-wise number of PCA components of the subspace corresponding to true training labels over the layers of multiple networks with various corruptions. Although it is not mentioned in the legend, all the 100 classes of CIFAR-100 are plotted.

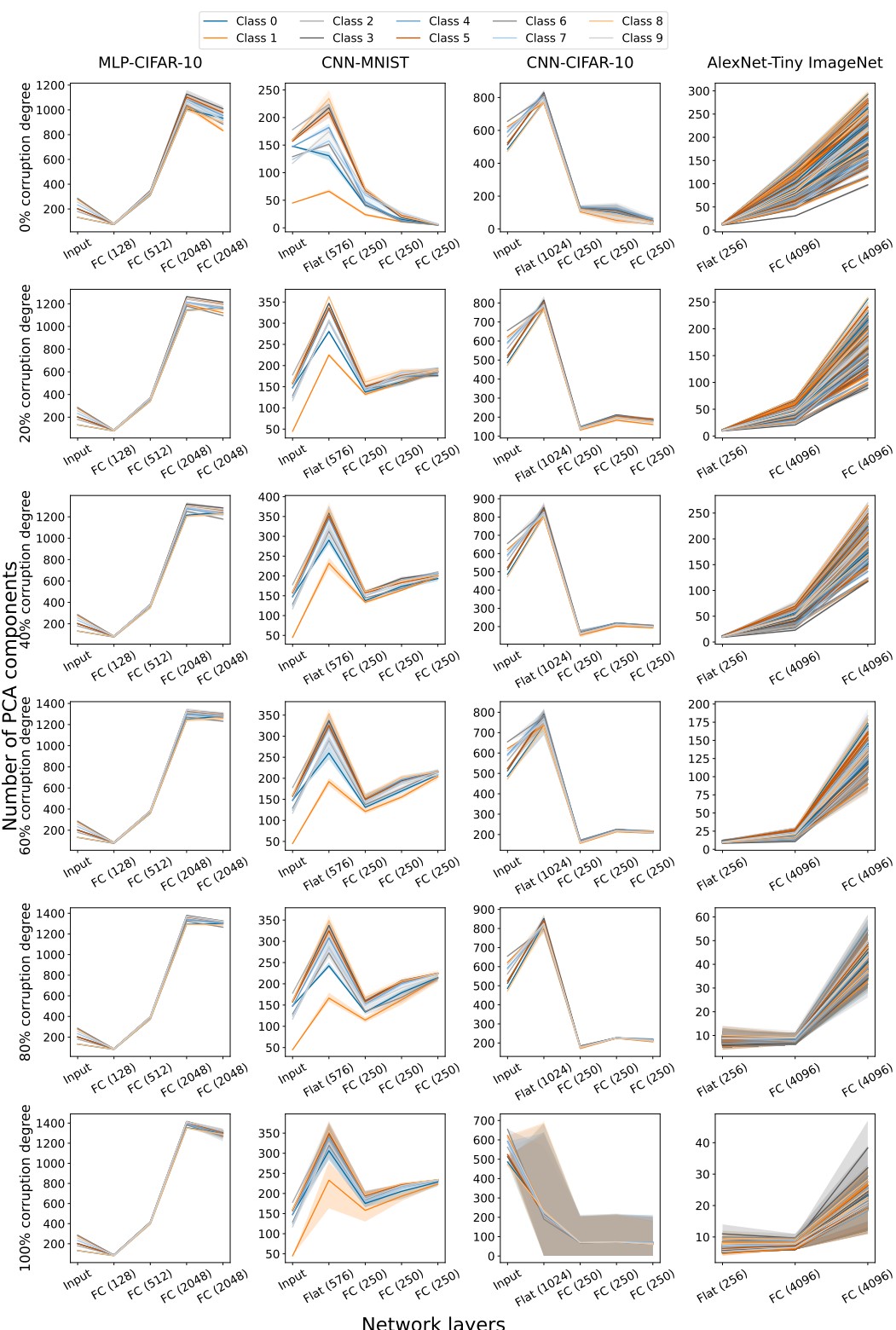

Figure 33: Class-wise number of PCA components of the subspace corresponding to true training labels over the layers of multiple networks with various corruptions. Although it is not mentioned in the legend, all the 200 classes of Tiny ImageNet are plotted.

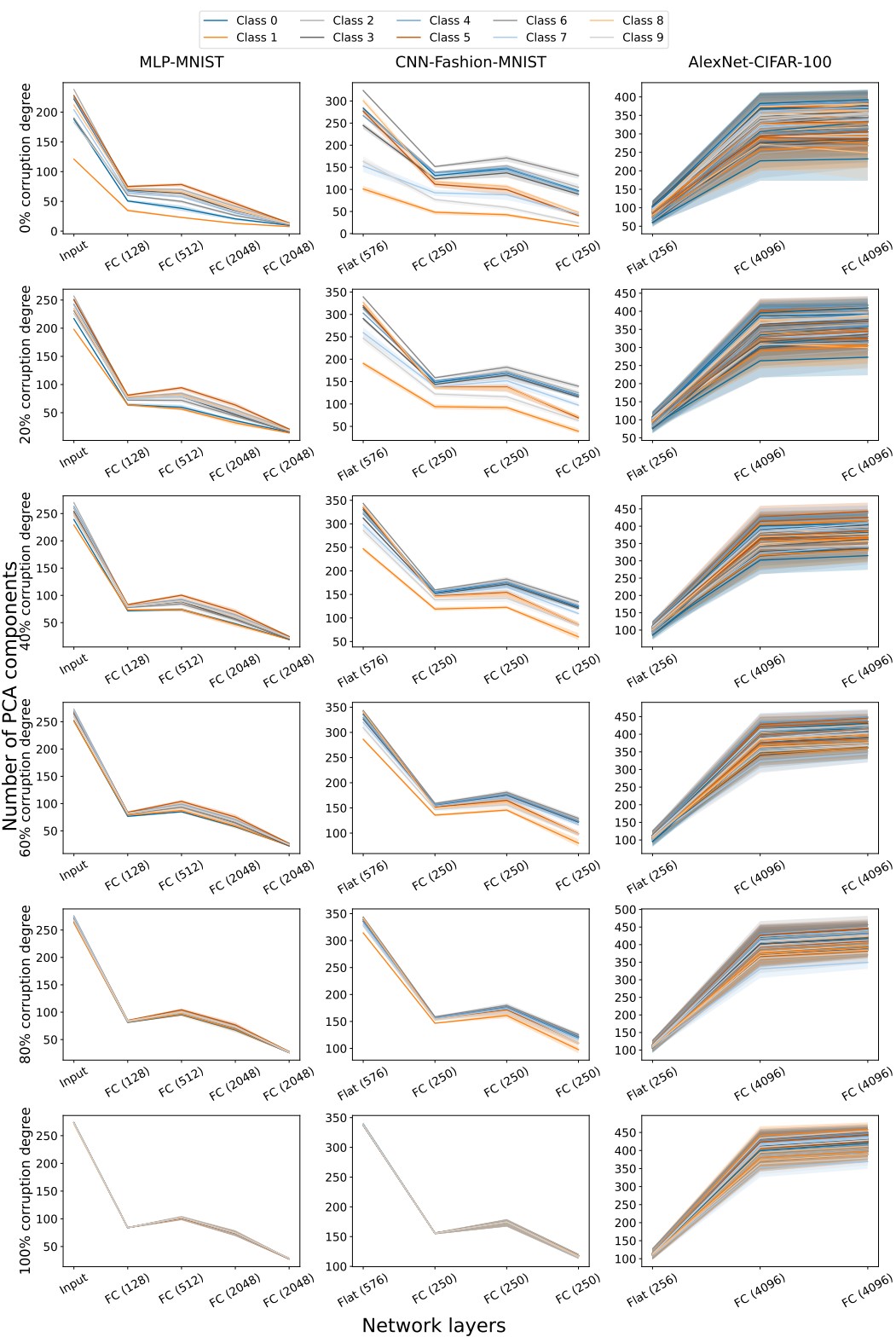

Figure 34: Class-wise number of PCA components of the corrupted training subspace over the layers of multiple generalized networks with various corruption degrees. Although it is not mentioned in the legend, all the 100 classes of CIFAR-100 are plotted.

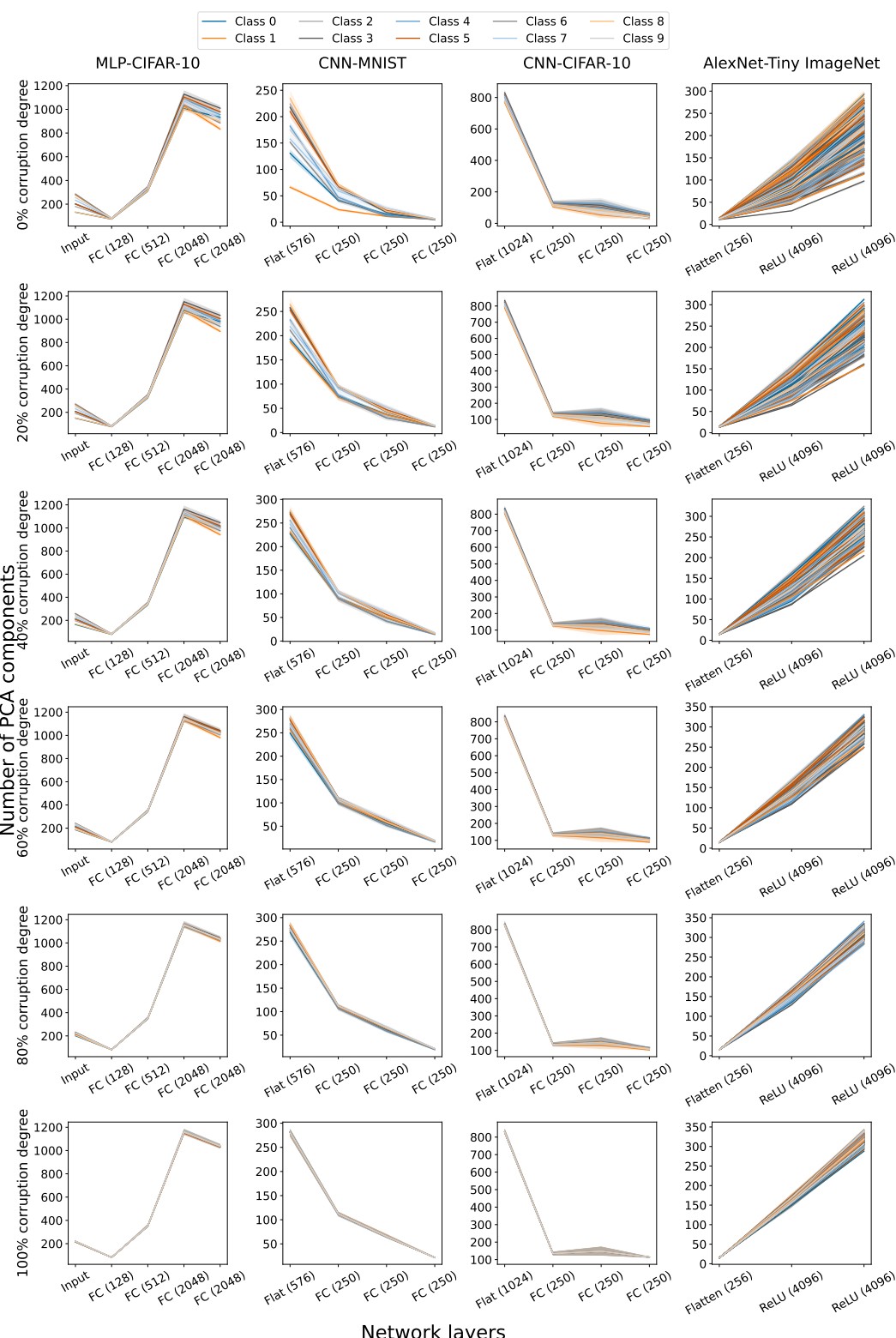

Figure 35: Class-wise number of PCA components of the corrupted training subspace over the layers of multiple generalized networks with various corruption degrees. Although it is not mentioned in the legend, all the 200 classes of Tiny ImageNet are plotted.

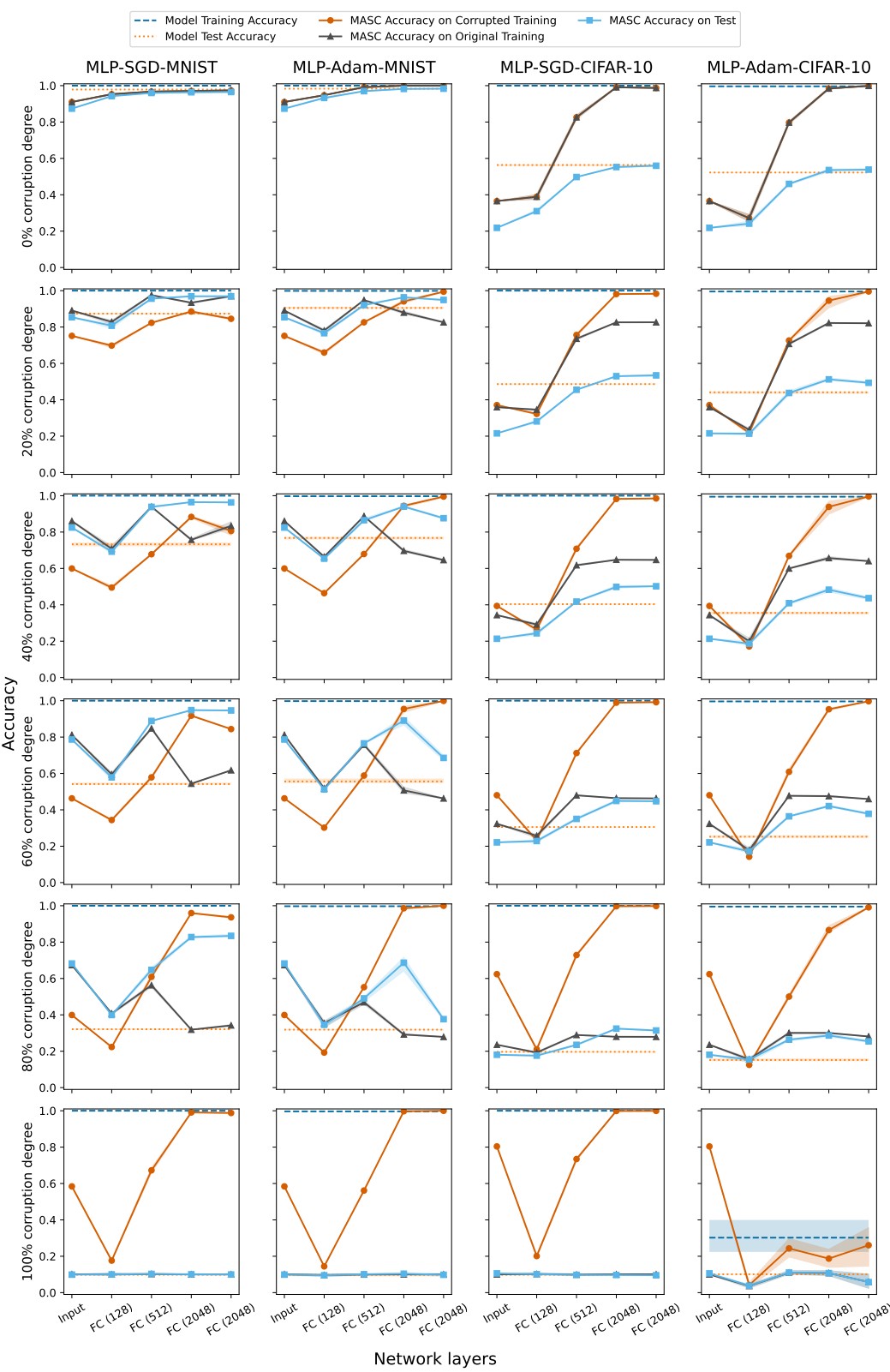

Figure 36: MASC accuracy over the layers of the MLP network when the data is projected onto corrupted training subspaces with the indicated corruption degree, for MLP models with MNIST and CIFAR-10 datasets. Rows corresponds to plots which have the same corruption degree and the columns correspond to the models with *SGD* and *Adam* optimizer as noted. Training and test accuracy of the model is shown.

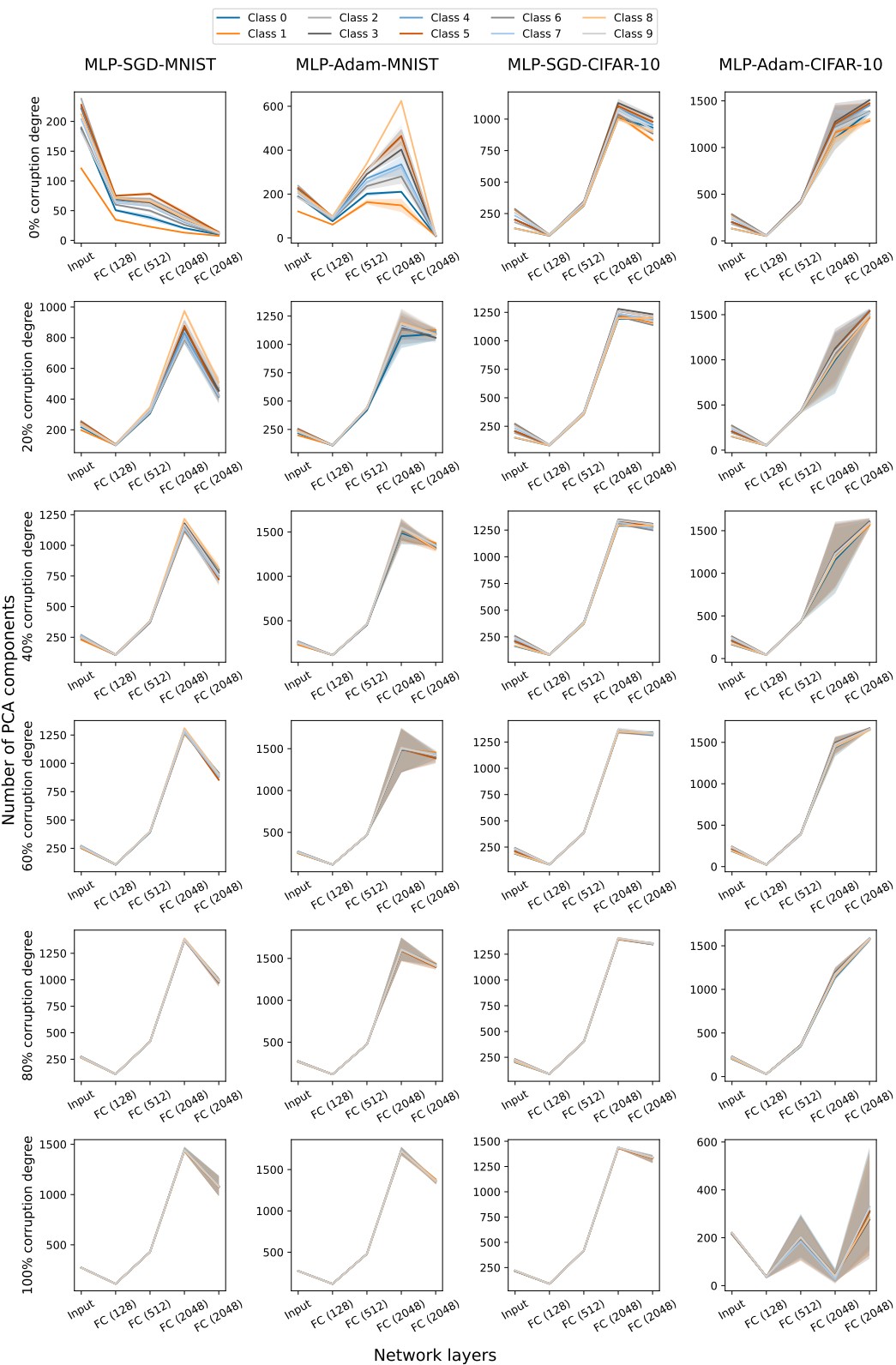

Figure 37: Class-wise number of PCA components of the corrupted training subspace over the layers of MLP networks trained with MNIST and CIFAR-10 datasets with various corruption degree. Rows corresponds to plots which have the same corruption degree and the columns correspond to the models with *SGD* and *Adam* optimizer as noted.

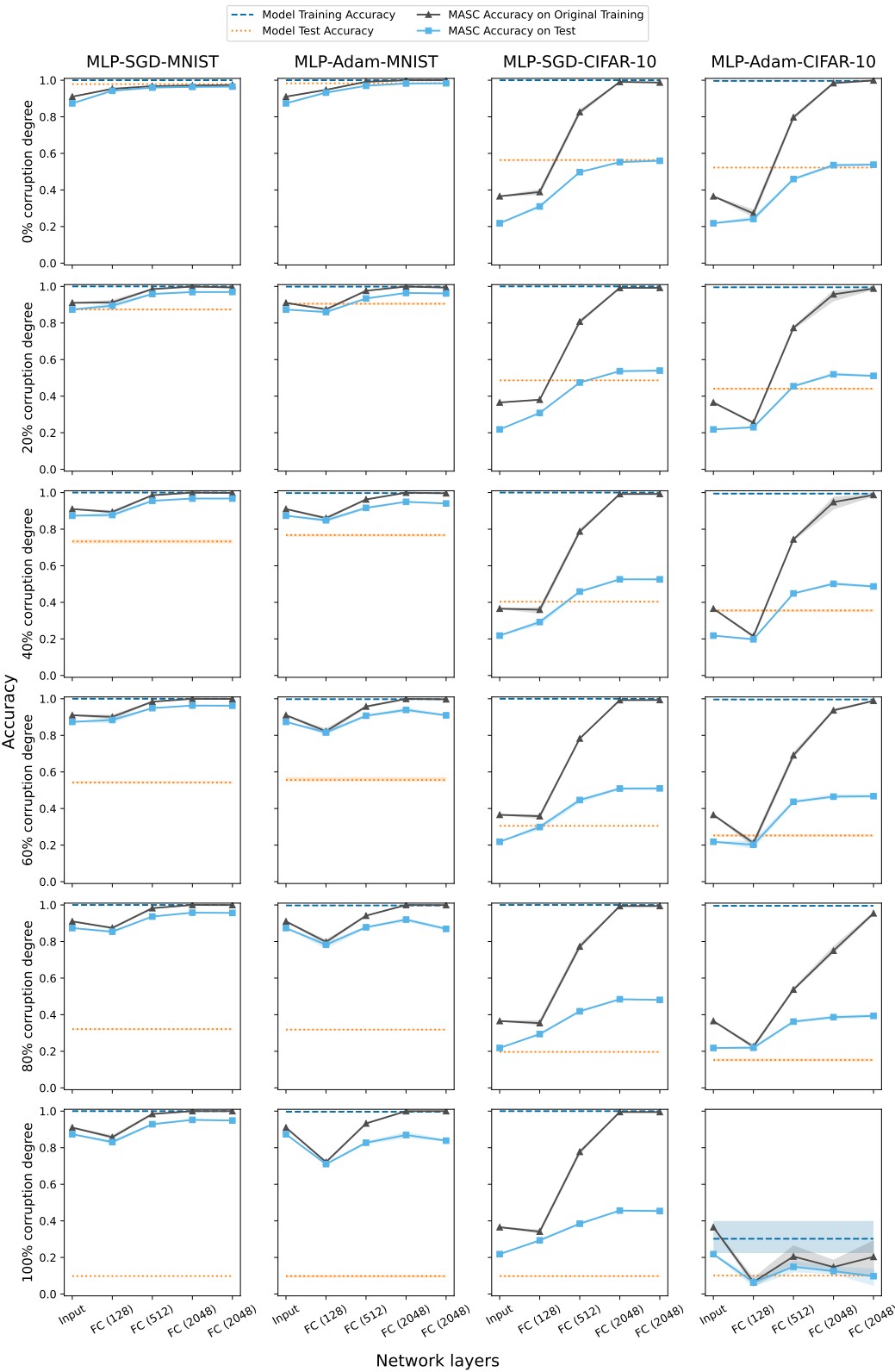

Figure 38: MASC accuracy over the layers of the MLP network when the data set is projected onto subspace corresponding to true training labels. Rows corresponds to plots which have the same corruption degree and the columns correspond to the models with *SGD* and *Adam* optimizer as noted. Training and test accuracy of the model is shown. FC corresponds to fully connected layer with *ReLU* activation.

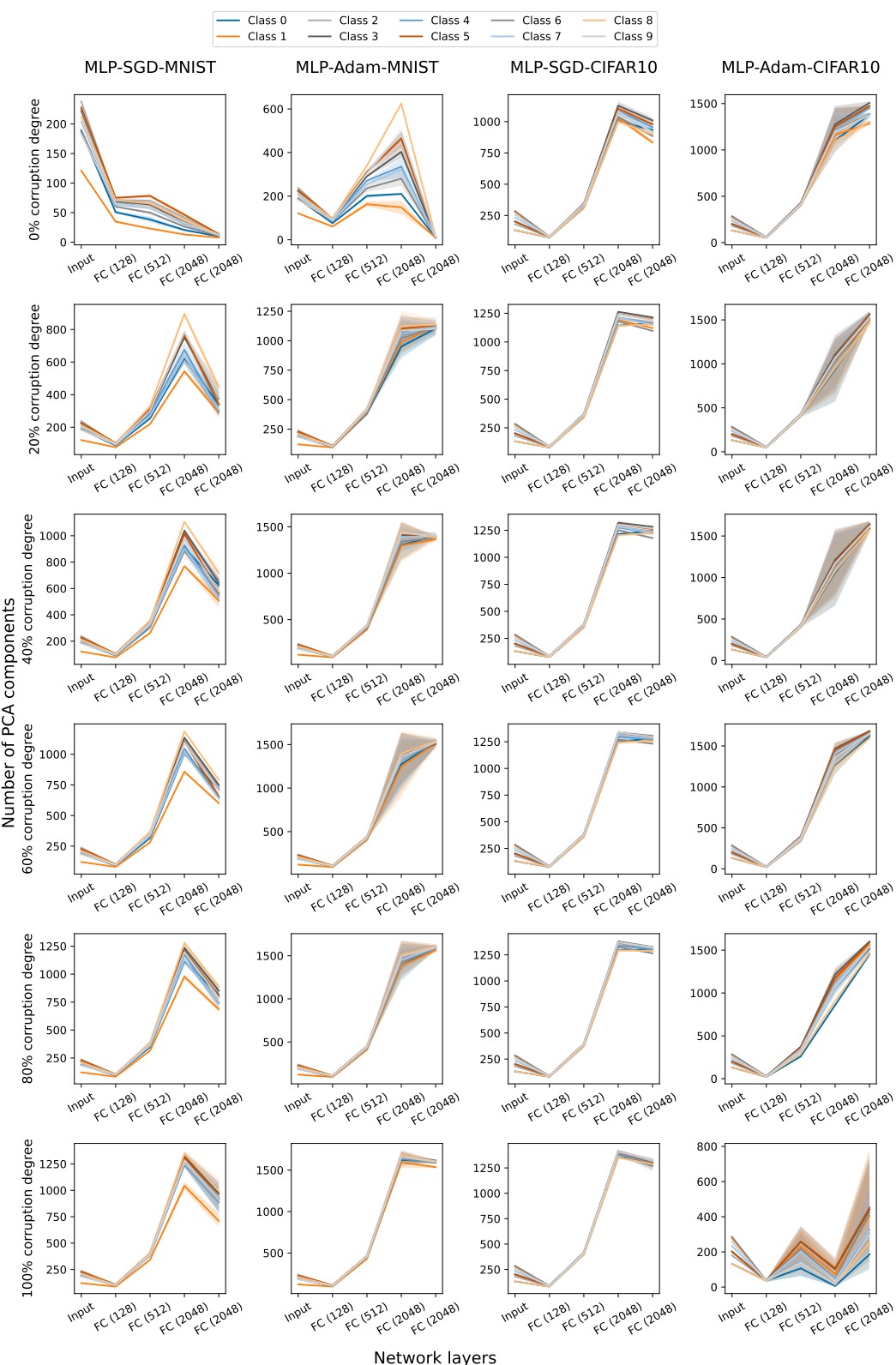

Figure 39: Class-wise number of PCA components of the subspace corresponding to true training labels over the layers of MLP networks with various corruption degrees. Rows corresponds to plots which have the same corruption degree and the columns correspond to the models with *SGD* and *Adam* optimizer as noted.

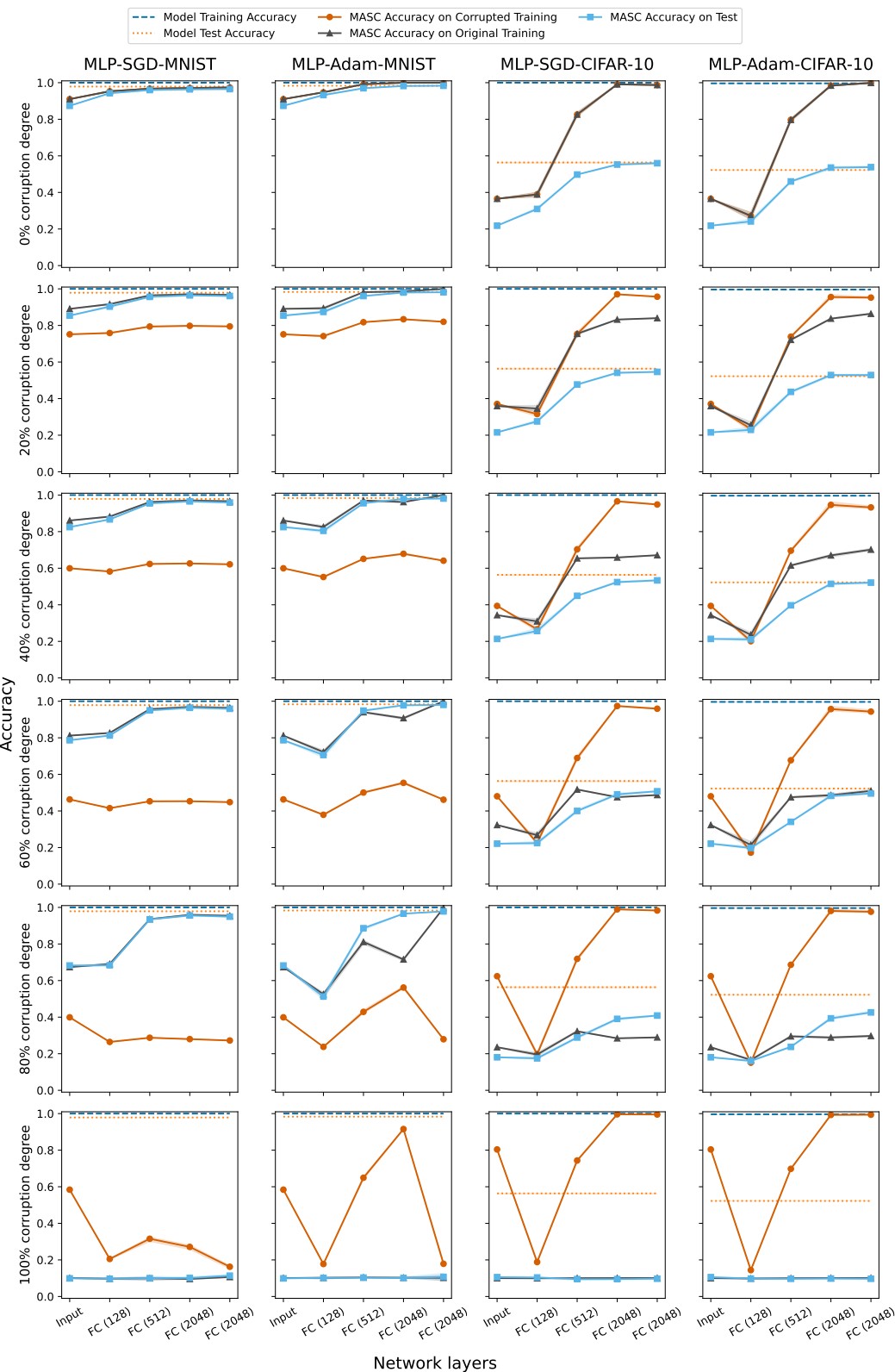

Figure 40: MASC accuracy over the layers of the generalized MLP network when the data set is projected onto corrupted training subspaces with the indicated corruption degree. Rows corresponds to plots which have the same corruption degree & the columns correspond to the generalized models with *SGD* and *Adam* as noted. Training & test accuracy of the generalized model with *SGD* and *Adam* is shown.

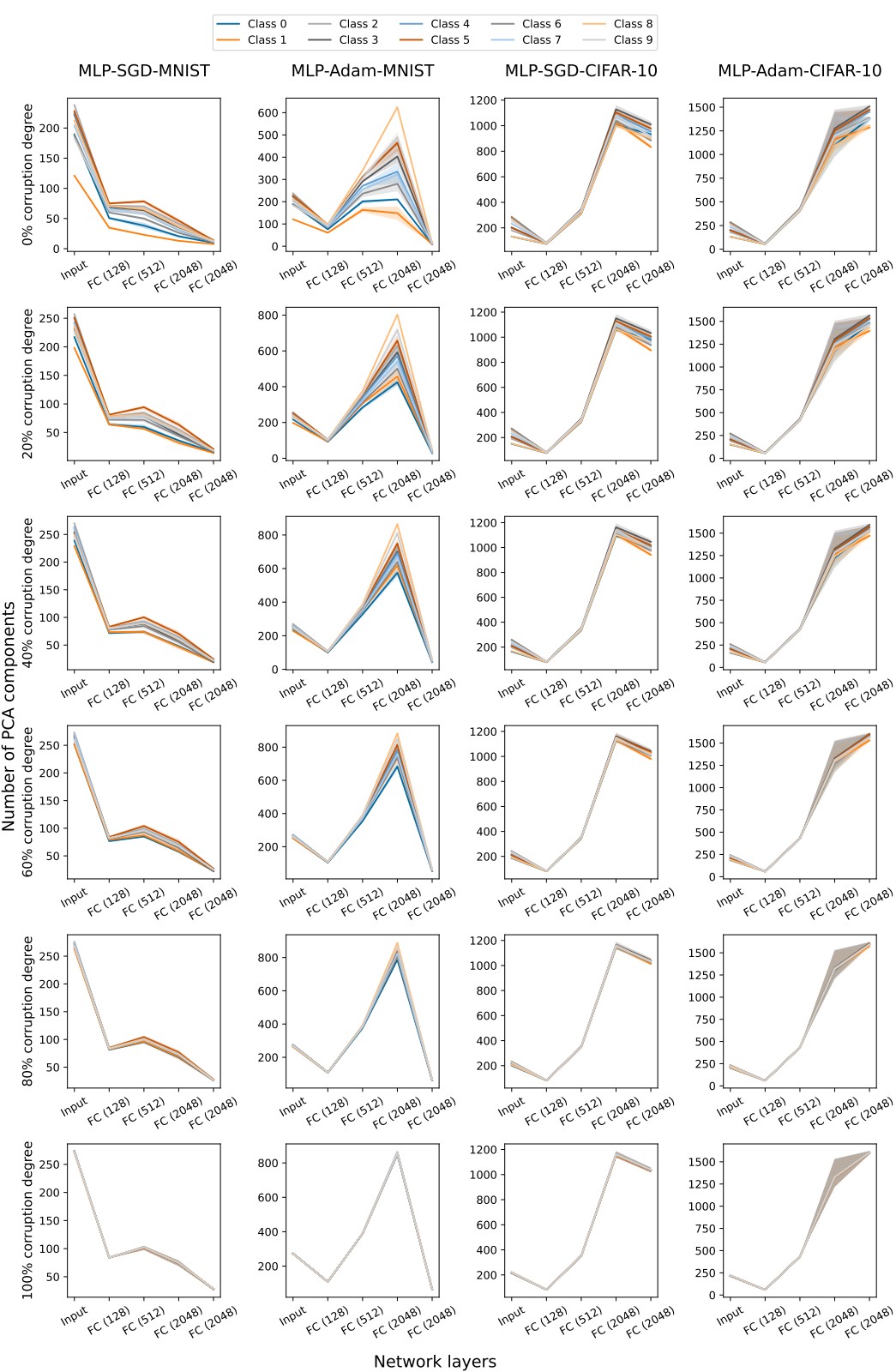

Figure 41: Class-wise number of PCA components of the corrupted training subspace over the layers of generalized MLP network with various corruption degrees. Rows corresponds to plots which have the same corruption degree and the columns correspond to the models with *SGD* and *Adam* optimizer as noted.

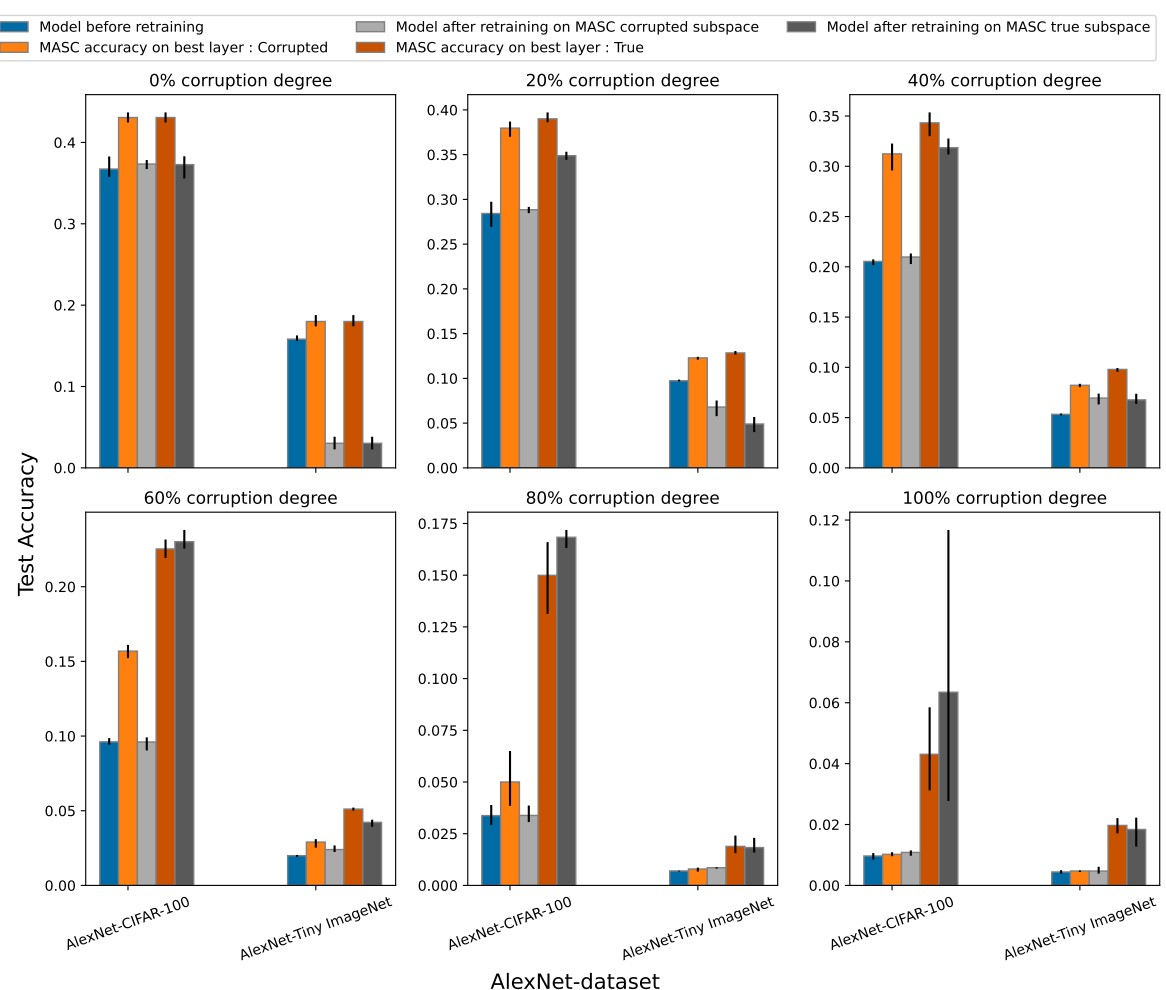

Figure 42: Test accuracies averaged over three runs on the 80% test dataset is plotted for different AlexNet-dataset pairs for various corruption degrees and for various models/MASC classifiers. Model before retraining corresponds to the existing memorized model. Model after retraining on MASC corrupted subspace corresponds to model trained with training dataset relabeled using MASC corrupted subspace predictions on the best layer. Model after retraining on MASC true subspace corresponds to model trained with training dataset relabeled using MASC subspaces corresponding to true label predictions on the best layer. MASC test accuracy on the best layers for corrupted and true label subspaces on existing corrupted models (before retraining) are shown for comparison. The best layer was identified using the validation set which was carved out of the test set for this experiment. Error bar represents the range on three different runs.

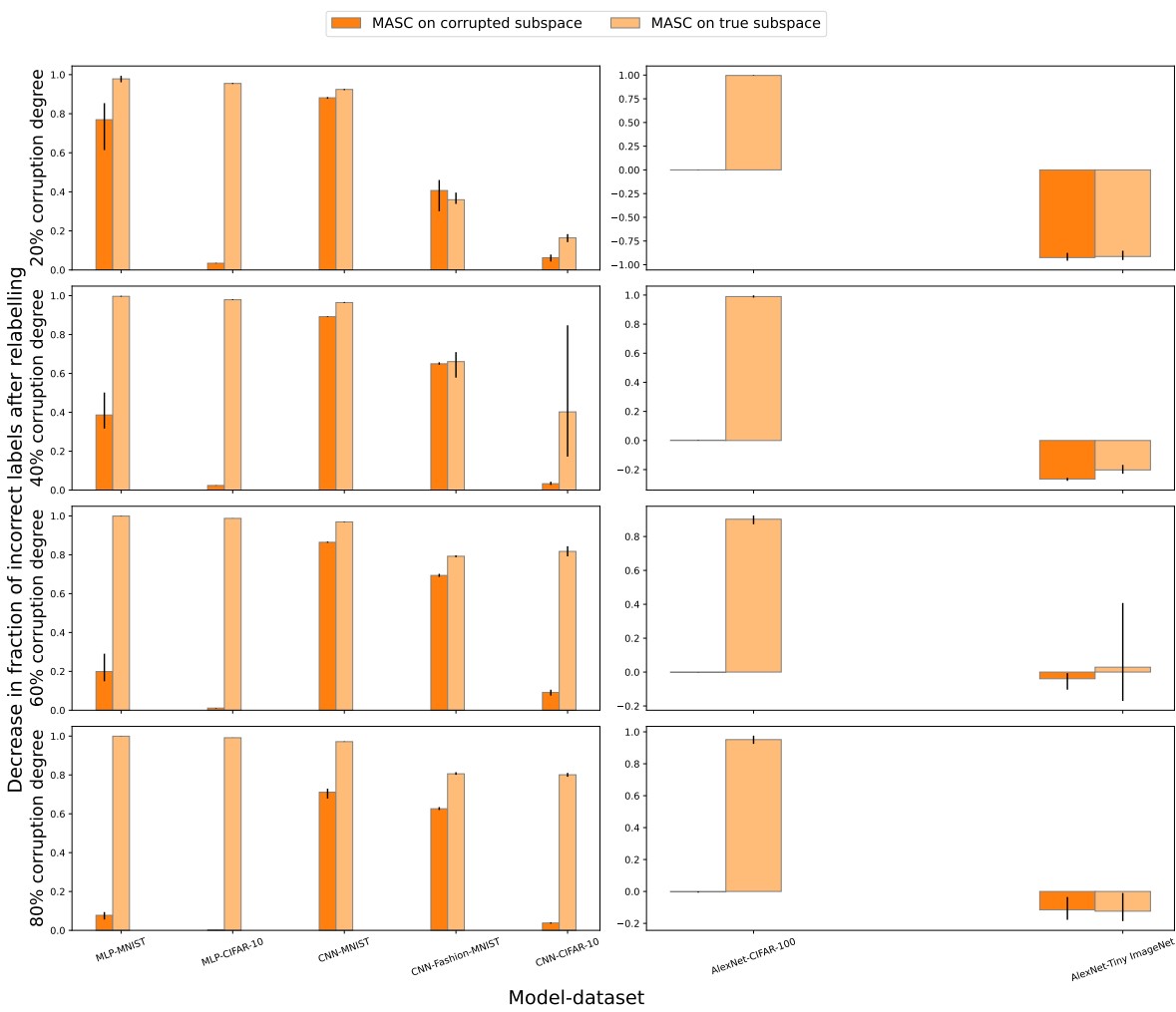

Figure 43: Decrease in fraction of incorrectly labeled data points resulting from relabeling the training set expressed as a fraction of size incorrectly labeled points in the existing corrupted data, using the corresponding best-layer MASC classifier. That is if X is the number of data points incorrectly labeled in corrupted dataset and Y is the number of data points incorrectly labeled in relabeled dataset then each of these plots refer to (X-Y)/X. Rows corresponds to results with the same corruption degree. The first column correspond to results with MLP and CNN models and second column with AlexNet model, as noted. The fractions are averaged over three runs. Error bar represents the range on three different runs.

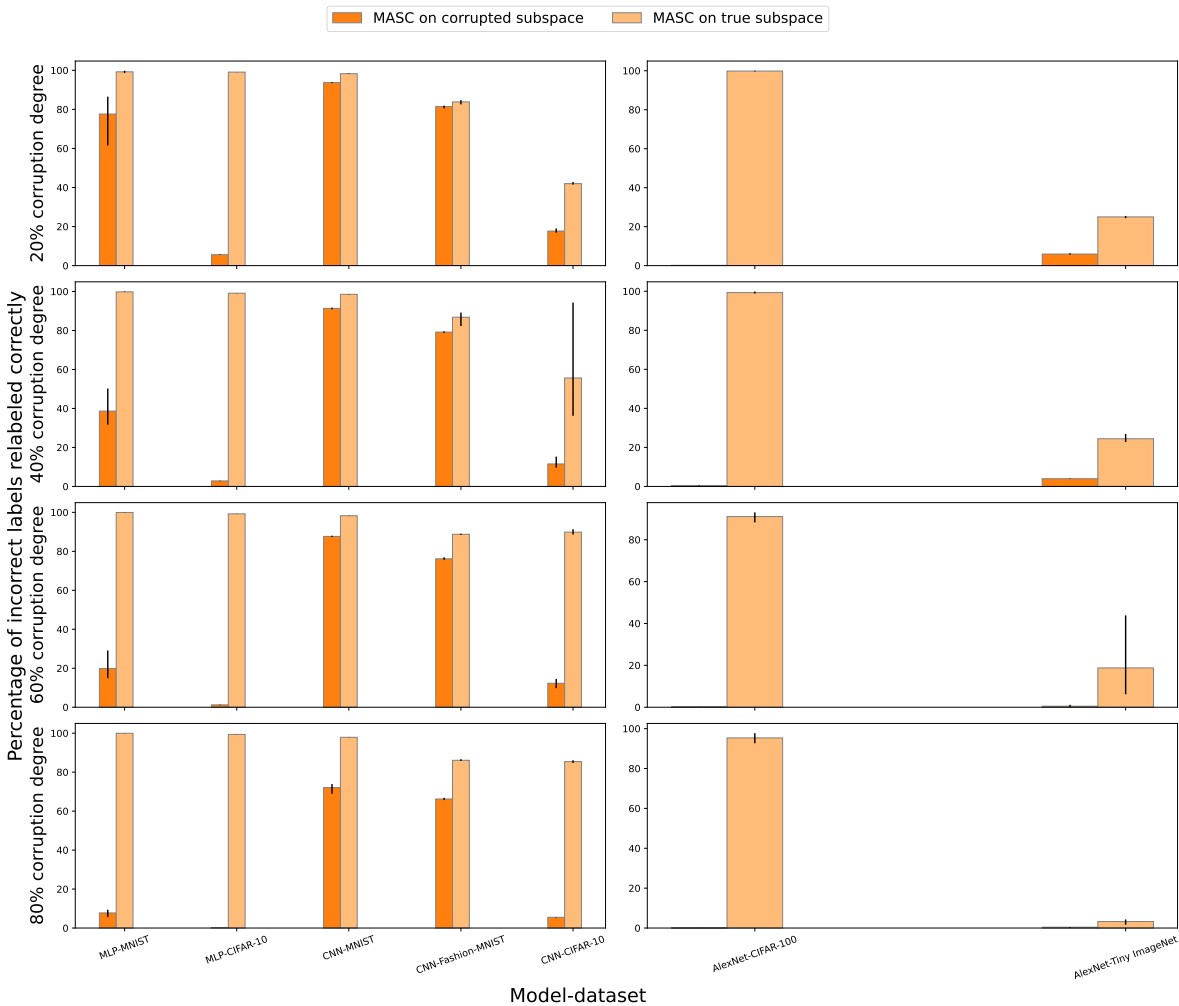

Figure 44: The percentage of incorrect labels that are correctly relabeled using MASC (corrupted subspace and true subspace) for all the models and for corruption degrees 20%, 40%, 60%, 80%. Rows corresponds to results with the same corruption degree. The first column correspond to results with MLP and CNN models and second column with AlexNet model, as noted. The percentage are averaged over three runs. Error bar represents the range on three different runs.

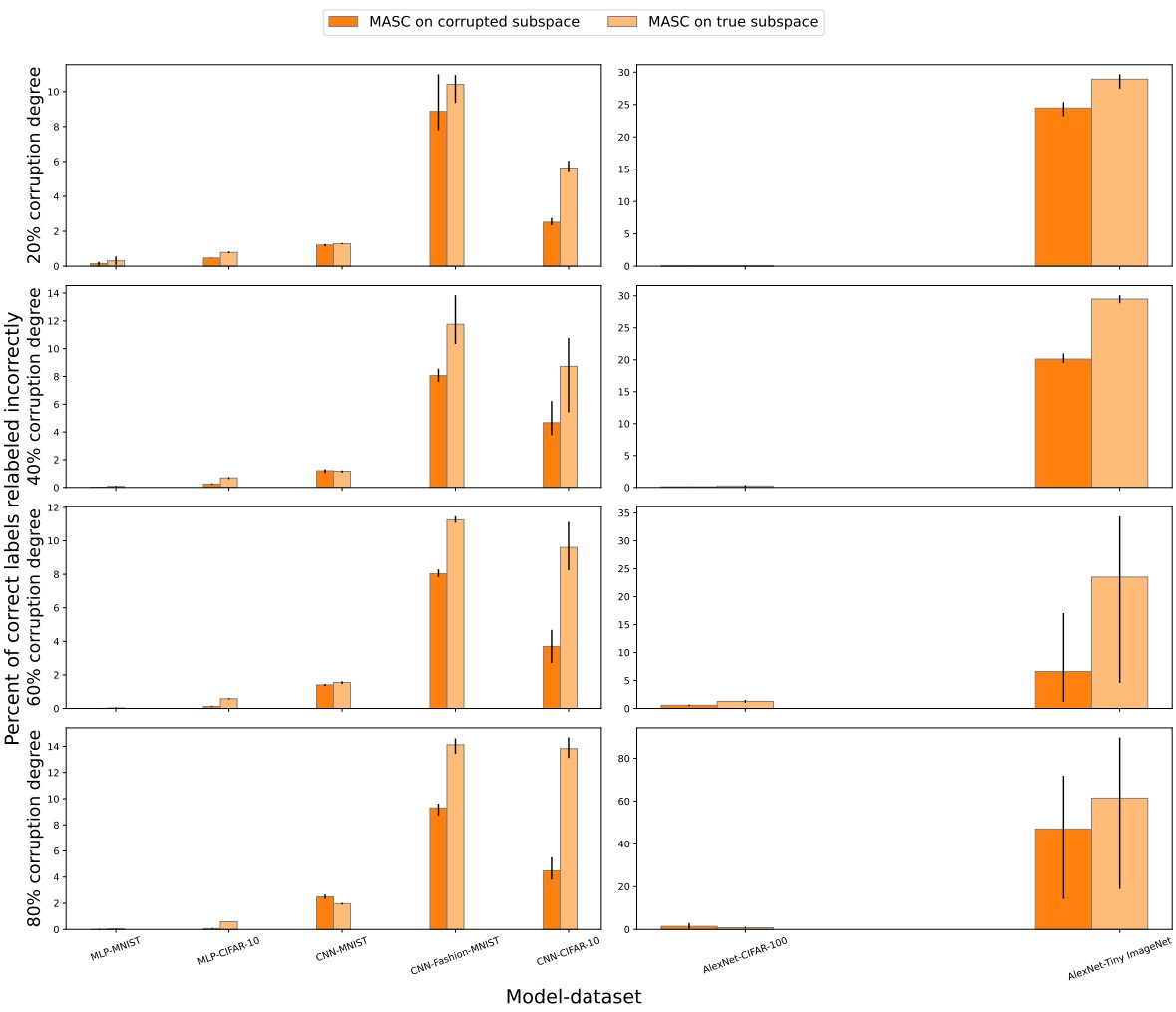

Figure 45: The percentage of correct labels that are incorrectly relabeled using MASC (corrupted subspace and true subspace) for all the models and for corruption degrees 20%, 40%, 60%, 80%. Rows corresponds to results with the same corruption degree. The first column correspond to results with MLP and CNN models and second column with AlexNet model, as noted.The percentage are averaged over three runs. Error bar represents the range on three different runs.

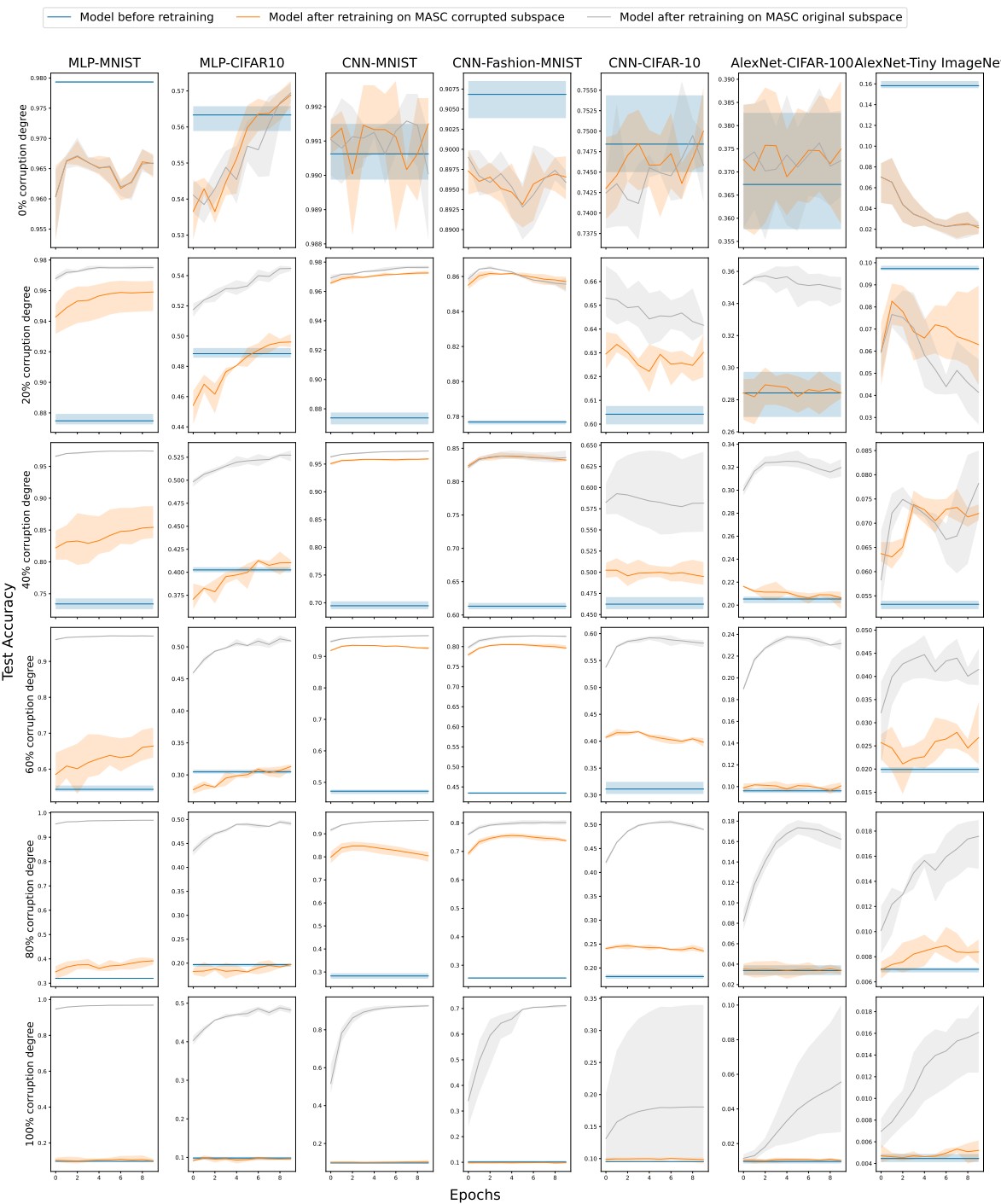

Figure 46: Test accuracy on 80% dataset is plotted for different model-dataset pairs for various corruption degrees over 10 epochs. The experiment was performed without using early stopping. Model before re-training corresponds to the existing memorized model. Model after retraining on MASC corrupted subspace corresponds to model trained with training dataset relabeled using MASC corrupted subspace predictions on the best layer. Model after retraining on MASC true subspace corresponds to model trained with training dataset relabeled using MASC subspaces corresponding to true label predictions on the best layer. Error bar represents the range on three different runs.

