# OpenReview forum: "Decoding Generalization from Memorization in Deep Neural Networks"
_TMLR — Accepted by TMLR_

### Review · Reviewer_fBaQ · 2025-10-04

**Summary Of Contributions:**

## Summary

This paper explores the intriguing question of why overparameterized deep networks trained on corrupted labels tend to generalize poorly, despite achieving perfect training accuracy. The authors provide compelling evidence that such networks still retain **latent generalization capability** within their internal representations. To reveal this hidden potential, they propose the **Model Activation Subspace Classifier (MASC)**—a simple yet effective decoding framework that extracts generalizable information from the learned representations of memorized networks. Through extensive experiments, the authors demonstrate that MASC can dramatically boost test accuracy relative to the original models, even under substantial label noise. Overall, this work offers fresh empirical insights into the nuanced interplay between memorization and generalization in deep learning.

## Strengths
- The paper is clearly written, and largely self-contained, making it accessible to a broad audience.
- The empirical evidence is thorough and convincingly supports the central claim that memorization does not necessarily destroy generalization.
- The results challenge prevailing assumptions in the deep learning literature and introduce a novel interpretability tool (MASC) with strong practical implications.

**Audience:**

Yes

**Audience Explanation:**

The issues of poor generalization and memorization in deep networks are critical in the literature, and understanding how to resolve them and what are the underlying mechanisms is highly essential. Those insights can be further adapted to foundation models, thus boosting the development of modern AI.

**Claims And Evidence:**

Yes

**Claims Explanation:**

The claims in the submission are generally supported by accurate and convincing empirical evidence. The authors conduct comprehensive experiments across multiple architectures (MLP, CNN, and AlexNet) and datasets (MNIST, Fashion-MNIST, CIFAR-100), demonstrating substantial and consistent performance gains achieved by MASC. The quantitative improvements—often exceeding several hundred percent over the baseline—clearly validate the central claim that memorizing networks retain latent generalization ability. The presentation of results is clear, and the comparisons are fair and reproducible. However, while the empirical evidence is strong, the lack of theoretical analysis or experiments on large-scale modern architectures somewhat limits the depth of the conclusions. Overall, the evidence convincingly supports the primary claims but would benefit from additional theoretical or large-scale validation.

**Requested Changes:**

## Suggestions
- The central role of **PCA** in constructing MASC is mentioned but not conceptually well explained. Providing more intuition would help readers better understand its necessity and contribution to the method’s success.
- While the experiments are well executed, they are limited to relatively **small-scale models and datasets**. Evaluating MASC on larger architectures (e.g., ResNet, ViT) and providing **theoretical justification** for its effectiveness would greatly strengthen the paper’s impact and generality.

---

> ### Author Response · Authors · 2025-12-10
> **Author response to Reviewer fBaQ**
>
> We thank the reviewer for their thoughtful & constructive comments, and their recognition of the strengths of our work and its broader implications for furthering a more detailed understanding of generalization and memorization.
>
> Below, we respond individually to the reviewer’s suggestions:
>
> * *The central role of PCA in constructing MASC is mentioned but not conceptually well explained. Providing more intuition would help readers better understand its necessity and contribution to the method’s success.*
>
> We thank the reviewer for bringing this up. Indeed, this is a direction that could have used better exposition. Accordingly, we have added the paragraph below in Section 8 to elaborate on this point.
>
>
> In building MASC, we were motivated by the manifold hypothesis in machine learning Goodfellow et al. (2016) that posits that high-dimensional data typically reside on a low-dimensional manifold. It has also been suggested (Brahma et al., 2015) that such manifolds in layerwise representations flatten across layers of deep networks. Fitting manifolds can be computationally expensive, so we were interested in examining the organization of classwise data in subspaces, even if such subspaces might be somewhat higher dimensional than the corresponding manifolds. Indeed, this view leads to the natural idea of classifying unseen data points by determining which class manifold it is closest to. MASC is simply a formalization of this idea. In particular, this classifier lends strong geometric motivation and intuition, in contrast to e.g. training a standard linear probe that iteratively minimizes a crossentropy loss. However, the reasons for the success of MASC in this setting are still largely unclear to us. The difficulty is that the principles that underlie the nature of layerwise representations in deep networks trained with standard techniques are not well understood at this time and it appears that such representations play a significant role in the success of MASC in the memorization setting. Indeed, it is even a bit surprising that the deep network does not directly leverage this structure to obtain better generalization to true labels, although that may also be because its loss function aims to maximize training accuracy which might run counter to the act of bettering generalization to true labels.
>
> * *While the experiments are well executed, they are limited to relatively small-scale models and datasets. Evaluating MASC on larger architectures (e.g., ResNet, ViT) and providing theoretical justification for its effectiveness would greatly strengthen the paper’s impact and generality.*
>
> Thanks. This is a good point. We have now included results for ResNet-18 trained on CIFAR-10. These updates appear in Table 1, Figure 3 and Figure 5 of the main paper, and the corresponding table (Table 6)  in the Supplementary Material has been revised accordingly.
>
>
> On providing theoretical justification for the effectiveness of MASC: On the face of it, this is something we could have incorporated. For example, one such justification could be the following: “Even with high (e.g. 80%) degrees of corruption, data points from the correct class outnumber datapoints from each of the incorrect classes. PCA run on this data, is likely to capture variance corresponding to this correct class. As a result, MASC, which simply measures closeness, does better on generalization”. However, this explanation ought to apply to every layer and yet, we observe empirically that for some layers, the model generalizes better. Secondly, some of the new probes we have run, in response to a suggestion from another reviewer also often have performance comparable to MASC, and in some cases, in layers distinct from those where MASC has good performance. As such, we believe that this phenomenon holds much more nuance, and depends more specifically on the nature of representations that the model creates at each layer and the decodability of it, as manifested by different probes. The principles governing the same are not well understood yet. Therefore, we think that a deeper empirical examination is in order before we can get down to the goal of proffering a credible theoretical explanation. In short, we have deliberately chosen to stay away from offering a somewhat superficial justification, for the above reasons.

---

### Review · Reviewer_mKtr · 2025-10-12

**Summary Of Contributions:**

The paper introduces a simple **Minimum Angle Subspace Classifier (MASC)** to “decode” latent generalization from trained deep networks—even when those networks have memorized noisy labels. For each layer, the authors estimate **class-conditioned PCA subspaces** from the (possibly corrupted) training features, then classify a test point by the class whose subspace yields the **smallest angle** to the point’s feature vector. Across MLP/CNN/AlexNet models and MNIST/Fashion-MNIST/CIFAR-10/100/Tiny-ImageNet, MASC at **at least one layer** often **substantially exceeds the model’s own test accuracy** under moderate–high label corruption. If true training labels are known post hoc, building subspaces with true labels yields even larger gains. The authors also explore **retraining** with labels relabeled by the best-layer MASC and report mixed improvements. The results support the view that **memorization does not destroy internal structure**; trained networks can retain **linearly decodable, class-structured geometry** in hidden layers despite noisy supervision.

**Key strengths**

* Conceptually clean, training-free probe with **no weight updates**; easy to implement and reason about.
* Consistent empirical trend: **large test-accuracy improvements** over the base model at one or more layers under label noise.
* Clear experimental setup spanning **multiple datasets, architectures, and corruption levels**.
* Interesting **dual result**: true-label subspaces post hoc can unlock even higher generalization; plus a retraining pipeline that leverages the probe.

**Key weaknesses**

* **Theory is minimal**: no analysis of when/why class subspaces preserve angles under label noise; no guarantees for layer selection.
* **Baselines are thin**: lacks comparisons to standard probes (linear probe/logistic regression, LDA/NCM, k-NN in feature space, cosine-to-class-means, ridge/SVM), or other subspace/classifier geometry methods.
* **Hyperparameter transparency**: PCA variance thresholding (99%), sign-flip augmentation, and layer selection appear **heuristic**; sensitivity/ablation is limited.
* **Methodological concern**: best-layer selection and early stopping use a **20% split of the test set** as validation in retraining, which risks evaluation bias.
* Mixed **scalability and practicality** questions: per-class PCA for high-dim features/layers may be costly; unclear guidance for which layer(s) to use in practice.
* Evaluation on **modern backbones** (e.g., ResNet-18/50, ViT) is limited; AlexNet is dated.

**Additional Comments:**

* Writing is clear and figures are helpful. Consider **re-ordering** experiments: first establish the core MASC effect vs. strong baselines on modern backbones, then show **true-label subspaces** and finally the **retraining** study.
* Report **per-layer dimensionalities** of PCA subspaces (already touched in the supplement) alongside accuracy to illuminate the complexity–performance trade-off.
* Consider evaluating **principal angles** (full spectrum) or **subspace affinity** metrics instead of/min addition to the single-projection angle; this might further stabilize performance.
* Explore a **layer aggregation** variant (e.g., voting or stacking simple probes across 2–3 adjacent layers) to avoid brittle best-layer choice.
* Minor: define all notations once (Sk, angles, projection operator); ensure consistent treatment of **activation vs. pre-activation** features.

**Audience:**

Yes

**Audience Explanation:**

The work addresses a **core question** in modern deep learning—how generalization and memorization coexist—and proposes a **lightweight, interpretable probe** that reveals **structure in hidden layers under heavy label noise**. This is directly relevant to readers interested in **representation analysis, robust learning under noisy labels, diagnostics/probing**, and practical techniques to **salvage generalization** from existing models. The results may also stimulate **theoretical work** on subspace geometry and **practical tools** for auditing noisy datasets.

**Broader Impact Concerns:**

The method is primarily an **analysis/probing tool**; direct societal risks are limited. Potential concerns:

* **Data leakage or privacy:** If subspaces can **reconstruct or expose** structure from models trained on sensitive data, this may aid **information extraction** beyond intended usage. Clarify what MASC reveals about **individual examples** vs. class structure.
* **Relabeling risks:** Using probe-generated labels to retrain could **amplify biases** or entrench mislabeled patterns if not validated on clean data. Recommend guardrails (hold-out validation, uncertainty thresholds, human oversight).
* **Misinterpretation:** Emphasize that improved probe accuracy **does not imply** the base model generalizes well; MASC is a **diagnostic** and should not be conflated with deployed performance.

Adding a concise **Broader Impact** paragraph noting these points would be sufficient.

**Claims And Evidence:**

Yes

**Claims Explanation:**

The empirical evidence is **consistent and compelling within the reported scope**: layer-wise MASC repeatedly outperforms the trained model’s test accuracy under label corruption, often by large margins (e.g., >100% relative improvements at higher noise on several model/dataset pairs). The paper reports multiple corruption levels, multiple architectures, and both **corrupted-label** and **true-label** subspace variants, and shows that the phenomenon persists broadly. The retraining study—though mixed—adds credibility by probing **downstream utility** of the decoded signal. However, support would be strengthened with (i) **stronger baselines**, (ii) **statistical tests** across seeds, and (iii) avoiding **test-set splits for model selection**.

**Requested Changes:**

**Critical (affect acceptance):**

1. **Add stronger baselines:** Compare MASC to (a) **linear probes** (logistic regression / ridge / linear SVM) on frozen layer features; (b) **nearest class mean** with cosine; (c) **LDA/QDA**; (d) **k-NN** in feature space; (e) cosine-angle to **class mean subspaces** vs. PCA subspaces. Report accuracy and compute cost across layers.
2. **Eliminate test-set leakage in selection:** Replace the 20% test split used for **best-layer selection/early stopping** with a **proper validation set** carved from training (or a held-out set). Re-run retraining results accordingly and discuss any changes.
3. **Sensitivity/ablation studies:** Vary PCA variance thresholds, the **sign-flip augmentation**, number of components per class, distance metrics (principal angles vs. first-component angle), and **layer choice** strategies (e.g., earliest layer passing a threshold). Quantify robustness across **seeds** (≥5) with **mean ± 95% CI** and simple significance tests.
4. **Modern backbones:** Include **ResNet-18/50** (and, ideally, a small **ViT**) to demonstrate that the effect holds on contemporary architectures.
5. **Clarify computational complexity:** Provide **time/memory** costs for building per-class subspaces and for inference across layers; discuss scaling to high-dimensional layers.

**Important but not strictly critical (would strengthen the paper):**

6. **Theory sketch or intuition:** Provide analysis (even stylized) explaining why **class-conditioned subspaces** preserve **angle-based separability** under label noise and why later layers often help. Connect to **representational geometry** and known results on **early-learning** vs. memorization.
7. **Failure analysis:** Systematically report **where MASC fails** or brings marginal gains (datasets, layers, corruption levels); examine **when true-label subspaces** help the most.
8. **Corruption types:** Beyond uniform label shuffling, test **instance-dependent** or **class-dependent** noise; discuss domain shift vs. label noise.
9. **Code and reproducibility:** Release code and configs; document PCA implementation details, preprocessing, and exact layer taps.
10. **Fair baseline training:** Include baselines with **regularization/early stopping** typical for each dataset to ensure model test accuracies are not artificially deflated.

---

> ### Author Response · Authors · 2025-12-10
> **Author response to Reviewer mKtr**
>
> We thank the reviewer for their very detailed and constructive review. We have made multiple changes in response to the reviewer’s requests, which are detailed below.
>
> * *1. Add stronger baselines.*
>
> We thank the reviewer for the suggestion to strengthen the baselines. We have run additional baselines, as requested, and incorporated the requested comparisons in Section 5 of the main paper and Section 3 of the Supplementary Material, where we now report both layerwise test accuracy and computational cost (GFLOPs) for MASC alongside Logistic regression, nearest class mean, LDA, QDA, and k-NN probes. These results cover MLP, CNN, and AlexNet models under multiple corruption degrees.
>
> For point (e) cosine-angle to class mean subspaces vs. PCA subspaces, we would appreciate clarification regarding the intended comparison. Specifically, it is not fully clear to us what the reviewer means by *class-mean subspaces*.
>
> * *2. Eliminate test-set leakage in selection.*
>
> We thank the reviewer for raising the concern about potential test-set leakage in layer selection. We would like to clarify that all reported model results in Section 7 were evaluated on the remaining 80% of the test dataset. We have improved the exposition in Section 7 to make this clearer.
>
> Regarding the suggestion to replace this 20% split with a dedicated validation set carved from the training data: While we agree that this would be a cleaner experimental setup, implementing it would require retraining all models from scratch, including all corruption-degrees variants and evaluations. Given the scale of these experiments, we were unfortunately unable to complete this.
>
>
> * *3. Sensitivity/ablation studies.*
>
> We thank the reviewer for these suggestions. Firstly, as requested, we have conducted a more detailed evaluation by varying the PCA variance thresholds used to construct subspaces (99%, 75%, 50%, 25%, and for 1 principal component). These results are reported in Section 4 of Supplementary Material  (Figure 7 and Figure 8). Secondly, we also quantify robustness across five independent seeds (10, 20, 30, 40, 50) and present the mean ± 95% confidence interval, as recommended.
>
> However, we were not able to incorporate additional variations. While these are good points, implementing these would require substantial time, methodological extensions and additional compute budgets beyond the current review cycle.
>
> * *4. Modern backbones.*
>
> We have now included results for ResNet-18 trained on CIFAR-10. These updates appear in Table 1, Figure 3 and Figure 5 of the main paper, and the corresponding table (Table 6)  in the Supplementary Material have been revised accordingly.
>
> * *5. Clarify computational complexity.*
>
> We thank the reviewer for this suggestion. In response, we have added a new Section 5 in the Supplementary Material where we report the total time (in seconds) and computational cost (in GFLOPs) of MASC for subspace construction and inference over the layers of the networks, in addition to briefly discussing scaling of these measures as a function of dimensionality. We also include the GFLOPs required specifically for building the per-class subspaces. We have also analyzed computational cost across layers for the other probes in Section 3.3.
>
>
>
> * *9. Code and reproducibility.*
>
> Thank you for this point. We agree and are in the process of consolidating and documenting all the code, including those of the new baselines, which will take a few more days. We will do so and make the code repository publicly available in the final version of the paper.
>
> * *Broader Impact Concerns.*
>
> We thank the reviewer for the suggestion. We have now added a section on the Broader impact statement along the lines of the reviewer’s suggestion.
>
> In closing, we thank the reviewer once again for the detailed comments and suggestions. Indeed, we have re-organized parts of the manuscript, and run multiple new experiments in response, including running new probes, running multiple PCA variants, as well as running MASC on a larger model (ResNet-18). We believe these changes serve to improve the paper.
> For some of the remaining points, while these are indeed interesting and valuable questions, we have not explored them in depth at this time. Some of these would require significant additional analysis and experiments.

---

### Review · Reviewer_Fq5y · 2025-11-20

**Summary Of Contributions:**

## Summary

Consider a classification task with a data distribution $D$ and an iid training dataset $T$ of size $m$ sampled from $D$. For a fixed percentage $p$, let $\hat{T}_p$ denote a modified training, where $p$ percentage of inputs are relabeled with a uniformly random labeling function. Denote the distribution of the corrupted dataset $\hat{T}_p$ by $\hat{D}_p$.

Prior work (Zhang et. al., Arpit et. al) demonstrates the ability of neural networks to fit the noisy dataset $\hat{T}_p$. This phenomena suggests that the hypothesis class $H$ pertaining to a specific neural architecture is highly expressive and capable of classifying for learning tasks with complex distributions (e.g. $\hat{D}_p$).

Somewhat confusingly, prior work and this article refer to the above phenomena as “memorization” a term that is both ill-suited and imprecise but repeatedly misused in this thread of literature (including Arpit et. al., Stephenson et. al etc).

Naturally as the percentage of noise $p$ increases, the distribution of the noisy training data $\hat{D}_p$ is further apart from the true distribution $D$ and thus one expects that predictors such as $h$ learned to fit samples $\hat{T}_p$ from $\hat{D}_p$ are ill-suited for labeling inputs as per $D$. In this article, the authors investigate whether a network $h$ trained to fit noisy data $\hat{T}_p$ contains layer-wise representations with predictive power for the true noiseless distribution $D$.

The authors, suggest that, from the “internals of $h$” -- layer-wise representation embeddings computed by $h$-- one can identify subspaces $S_{l,c}$ for each class $c$ and layer $l$ that could inform a variant of nearest neighbour classifier $h_{MASC, l}$ where for each input $x$, the predicted label is determined as,
$$\text{prediction}(h_{MASC, l}, x) =  \min_c ~\text{angle}\left(x_l, \text{projection}(x_l, S_{l,c})\right).$$
Here $x_l$ is the l-th layer representation internal to $h(x)$.

Experiments on MNIST, Fashion-MNIST and CIFAR100 show that the modified classifier $h_{MASC,l}$ often exhibits improved performance on test data from the original noise-less data distribution $D$. This procedure requires no retraining and suggests a way for learning accurate classifiers from noisy data. The article also suggests that a version of self-training where all labels (including the noisy labels) are re-labeled using $h_{MASC,l}$ and the neural weights that are fine-tuned on this relabeled dataset $T_{p,MASC,l}$ exhibits better performance on $D$.


## Strengths
1. **Experimental evidence**

The empirical study is systematic. The authors evaluate multiple networks, corruption levels, and both corrupted-label and true-label subspace constructions. The results generally affirm the utility of MASC as a tool for recovering performance on noise-less data.

2. **Lightweight method**

MASC is straightforward to implement, computationally inexpensive, and easy to interpret. This is a valuable property compared to conventional probes that require additional training.

## Weaknesses
1. **Usage of the term “Memorization”**

The poor performance of a network $h$ fit to training dataset $T_p \sim D_p$ when evaluated on $D$ is not memorization. It is perhaps more appropriate to refer to this as “overfitting” if $h$ attains low error on $T_p$ but high error on $D_p$. Otherwise, this is simply learning under noise.


Achieving low training error on $T_p$ is an outcome of 3 factors: (1) expressivity of $H$, (2) level of noise $p$, and (3) sample size $m$.  In the extreme, when $p = 100$, for any input $x$ in the domain, all labels are equally likely and thus *all* deterministic functions $h \in H$ exhibit the same test error: $\frac{C-1}{C}$ where $C$ is the number of labels. When the sample size $m$ is small such that for a training input, there is only one observed pair $(x,c)\in T_p$ then the learning algorithm can learn a neural network that predicts the label $c$ for $x$ and this way, can attain low training error on $T_p$ while exhibit low error on $D_p$, i.e. overfit w.r.t $D_p$.  Thus, achieving low training error on $T_p$ shows that the hypothesis class $H$ contains a sufficiently complex labeling function $h$ capable of labeling as per $T_p$ and such a phenomena requires low noise levels and small sample sizes. Even for $p=10$ with sufficiently large sample size, if one observes $(x,c)$ for all $c$ in the training data then a deterministic labeler cannot achieve low training error.

The continual insistence on terming this phenomena memorization rather than simply learning under noise is puzzling. Poor learning w.r.t $D$ is not the automatically the same as memorization.. especially if the training input are not i.i.d from $D$!

2. **Ambiguity of the usage of the term “generalization”**

After multiple reads, it appears the authors consider good or bad generalization to always mean “performance on D” regardless of the training distribution. This is not the precise usage of the term! If a network is trained on $T_p \sim D_p$ then generalization should technically mean performance on $D_p$ after learning from $T_p$.
While I understand the intention of the authors: to investigate whether $h$ learned on $T_p$ can be modified to perform well on $D$, I would strongly suggest that the authors be very precise about each usage of the term generalization. Some of the ambiguity is perhaps easily resolved if the authors set down more formal notation and identify the exact training and testing setup with symbols for the datasets and distributions (as in this review) and to rewrite instances of the word generalization with appropriate precision.


3. **The two motivating hypotheses are not mutually exclusive**

The paper contrasts two explanations for poor test accuracy under label corruption: (i) degraded representations and (ii) intact representations with poor readout. In practice, both mechanisms may be at play, especially given evidence that later layers overfit while earlier layers retain structure. A revised exposition should acknowledge this and situate MASC as quantifying the extent to which useful signal persists. Once again, the framing of this article is unclear and could be more precise. This is a serious issue as it impacts readability and either prevents other researchers in the field from building on the authors’ contributions or amplifies the existence of vague terminology in the literature.

4. **The main contribution is underemphasized**

The discussion in the last page briefly mentions that the experiments “suggest that MASC enables leveraging weights of models learnt on noisy data,” but treats this as future work. In fact, this capability appears to be the real (and only) contribution of this article. Elevating this point—namely, that noisy-trained weights can be repurposed effectively via subspace decoding—would greatly strengthen the manuscript. I would strongly recommend that the authors invest time/experiments to bolster this argument.

5. **Reported percentage improvements exaggerate effect sizes**

The manuscript frequently highlights percentage improvements measured relative to somewhat small baselines (e.g., 198%). These numbers can be misleading and should be replaced with absolute accuracy differences (e.g. model learnt on noisy data attains error A on noise-less test data, while model modified with MASC prediction rule attains error B > A), which would give a more transparent sense of improvement.

6. **Stylistic and terminological issues**

The manuscript occasionally uses informal terminology (e.g. words such as agnostic, corollary, memorization) without regard to their protected meanings (e.g. agnostic in the context of generalization!). On a minor stylistic note, there is inconsistent and over-capitalization of certain words (e.g., “Deep Networks”). Revising these phrases for clarity and style-consistency would greatily improve readability.

**Additional Comments:**

I apologize to the authors and the action editors for the delay in the review process.

**Audience:**

Yes

**Audience Explanation:**

Understanding the ability of neural networks to learn under noisy data is an important and relevant direction of inquiry.

**Broader Impact Concerns:**

I don't foresee any broader impact concern.

**Claims And Evidence:**

No

**Claims Explanation:**

The empirical findings are interesting, and MASC is a clean methodological contribution. However, the article's framing of contributions in the title, abstract and main summaries are vague and imprecise translation of their empirical finding.

**Requested Changes:**

This article would greatly benefit from improved conceptual framing and clearer articulation of its central message. With revisions emphasizing the utility of representation learned under noisy settings, the paper could make a stronger contribution. Please address the weakness listed above, by rewriting sections wherever appropriate, and being formal and precise using more mathematical notation.

---

> ### Author Response · Authors · 2025-12-10
> **Part-1 of author response to Reviewer Fq5y**
>
> We thank the reviewer for their careful reading of the paper and their thoughtful comments particularly emphasizing the use of precise language and terminology. We believe that the changes we have made in response will serve to significantly improve the clarity of the paper and the rigor of the formulation. We respond individually below to the comments made by the reviewer.
>
> * *1. Usage of the term “Memorization”.*
>
> In principle, we agree with the reviewer’s view on the unsuitability and imprecise nature of the term *memorization* to characterize the phenomenon of networks being able to perfectly learn training data corrupted with label noise. However, and as the reviewer acknowledges, this has unfortunately become standard practice (e.g., Zhang et al. 2017; Arpit et al. 2017) in this sub-area. Our intent with continuing the usage of this term is merely to help with readability for readers who are already familiar with prior work in this sub-area. That said, we agree with the substance of the point made by the reviewer and we have added additional exposition on this point. Specifically, we now start the Methods section with a new subsection on Preliminaries, which characterizes the setting more formally, along the lines of the succinct formulation of the reviewer. Secondly, we have sharpened the exposition in the abstract and the introduction so that the meaning of this setting is less likely to be misconstrued. Thirdly, in the first mention of the term memorization in the introduction, we have added a footnote directing the reader to the precise formulation in the new section on Preliminaries. Ultimately, after due consideration, we have decided to keep the usage of the term memorization in the rest of the paper so that it is consistent with usage of this term in prior related work, imperfect as it may be. In addition to avoiding dissonance for those familiar with this sub-field, we feel this will also offer for readers new to this subfield, the means to read other prior related work without subsequent cognitive load that adjusting to the now conventional, if unfortunate, use of these terms would entail.
>
> * *2. Ambiguity of the usage of the term “generalization”.*
>
> Here as well, we don’t disagree with the substance of the reviewer’s point and we have made the following changes, in response.
>
> In the abstract, we have rewritten corresponding wording to make it clear that we mean generalization to true labels. We also state this more precisely in the new Preliminaries subsection. In the Introduction and throughout the paper, all occurrences of “generalization” are now replaced with “generalization to true labels” to avoid the possibility of confusion.
>
> * *3. The two motivating hypotheses are not mutually exclusive.*
>
> Thank you for this incisive point. We have sharpened and slightly modified the framing, to address it. Specifically, we now frame the first hypothesis as one where all layers might have degraded representations, i.e. those that lead to diminished generalization to true labels. The alternative hypothesis has now been framed to allow for the possibility of one or more layers to have representations that allow for better generalization to true labels than the model. In a footnote in the introduction, we now specifically emphasize that the “better” representations may only manifest at certain layers. As for the reviewer’s comment that later layers overfit while earlier layers retain structure, we want to say the following. We empirically observe that in certain models (e.g. MLP-MNIST) later layers have better MASC accuracy than earlier layers, and in certain other models (e.g. CNN-Fashion-MNIST) the opposite happens, especially for high corruption degrees. In short, later layers overfitting isn’t a given for all the models we test. In any case, we are grateful to the reviewer for pointing this out; we believe that the revised framing significantly improves clarity on this front.

---

> ### Author Response · Authors · 2025-12-10
> **Part-2 of author response to Reviewer Fq5y**
>
> * *4. The main contribution is underemphasized.*
>
> Thanks for this point. Since this phenomenon has not been previously reported, our focus in this paper was on a careful empirical study of it under various conditions, including e.g. the case of subspaces corresponding to true labels. We agree that the question of better subspace decoding is an important direction for investigation and we plan to do so for future work. We feel that it is beyond the scope of the present paper, since this direction deserves detailed examination and exposition.
>
> * *5. Reported percentage improvements exaggerate effect sizes*
>
> We have been careful in stating precisely what the numbers reported mean. Indeed, there has been no intent to exaggerate. That said, we recognize the broader point of the reviewer and have accordingly also reported accuracy differences in Table 5 and 7, in Supplementary Material and referenced the same in the main text.
>
> * *6. Stylistic and terminological issues.*
>
> Thank you for bringing this up. We have made changes to the terms *agnostic* and *corollary* to avoid this type of confusion. In the revision, we have revised the writing to avoid over-capitalization as well.

---

### Decision · Action_Editor_LKgc · 2026-01-02

**Recommendation:** Accept with minor revision

**Additional Comments:**

Suggestions for revising the paper for the camera ready version:

Per the TMLR author guide, it is acceptable to place an Appendix after the References. It may be preferable to move the material currently in the supplementary file (which is the appendix) into the main PDF as an appendix. If you do so, please also update cross-references in the main text to point directly to the corresponding figures/sections in the revised manuscript (with hyperlinks where applicable). For example, instead of "A schematic of the retraining process using MASC is shown in Figure 1 in the supplementary material", we can have "A schematic of the retraining process using MASC is shown in Figure 7", with an appropriate link to the figure.

Although there is a dedicated related work section in section 2, the relationship and novelty over related work was not so clear. For example, it would be preferable to add citations for each of the two hypotheses explained in the 2nd paragraph in the introduction section.

The early stopping experiments in the appendix (Section 6 in page 10 of the supplementary file) seem to be important. The authors can consider moving this to the main paper.

**Audience:**

Yes

**Audience Explanation:**

The paper focuses on an important topic with a lot of related work in our community.

**Claims And Evidence:**

Yes

**Claims Explanation:**

The paper presents experimental results to support the claim that neural networks trained with noisy labels still contains layer-wise representations that can predict true labels, and the paper proposes a probing method. Two reviewers recommended "Leaning Accept" and one reviewer recommended "Leaning Reject". The latter reviewer's main concerns were about imprecise framing and misleading terminology usage. The authors clarified these points in the updated paper, and also added many new experiments to address the reviewers' (including the former two reviewers') concerns.

---

> ### Author Response · Authors · 2026-02-02
>
> Dear AE,
>
> We have uploaded the camera-ready version of the paper after incorporating all the revisions suggested as detailed below, as well as some minor edits involving punctuation and minor rephrasing, which came up when we went over the paper.
>
> > Per the TMLR author guide, it is acceptable to place an Appendix after the References. It may be preferable to move the material currently in the supplementary file (which is the appendix) into the main PDF as an appendix. If you do so, please also update cross-references in the main text to point directly to the corresponding figures/sections in the revised manuscript (with hyperlinks where applicable). For example, instead of "A schematic of the retraining process using MASC is shown in Figure 1 in the supplementary material", we can have "A schematic of the retraining process using MASC is shown in Figure 7", with an appropriate link to the figure.
>
> Thank you for this suggestion. We have moved the entire Supplementary file to the Appendix in the main PDF and updated cross-references and phrasing, as suggested.
>
> > Although there is a dedicated related work section in section 2, the relationship and novelty over related work was not so clear. For example, it would be preferable to add citations for each of the two hypotheses explained in the 2nd paragraph in the introduction section.
>
> Thank you for this point. We have added more explicit phrasing in the Related Work section referencing [Alain & Bengio, 2018] making remarks that align with our null hypothesis. The alternative hypothesis was proposed by us, and we have made this more explicit.
>
> > The early stopping experiments in the appendix (Section 6 in page 10 of the supplementary file) seem to be important. The authors can consider moving this to the main paper.
>
> Thank you for this suggestion. We have moved the two corresponding figures to the two appropriate sections in the main text and also added pointers to those figures in the said sections of the main text.
>
> We thank the reviewers and you once again for your time and consideration with this paper.
>
> Best,
>
> Authors

---

> > ### Comment · Action_Editor_LKgc · 2026-02-02
> >
> > Dear Authors,
> >
> > Thank you for the update! I will proceed with this version.